# Provable Generalization of Overparameterized Meta-learning Trained with SGD

**Yu Huang**
IIIS
Tsinghua University
y-huang20@mails.tsinghua.edu.cn

**Yingbin Liang**
Department of ECE
The Ohio State University
liang.889@osu.edu

**Longbo Huang**[*]
IIIS
Tsinghua University
longbohuang@tsinghua.edu.cn

## Abstract

Despite the superior empirical success of deep meta-learning, theoretical understanding of overparameterized meta-learning is still limited. This paper studies the generalization of a widely used meta-learning approach, Model-Agnostic Meta-Learning (MAML), which aims to find a good initialization for fast adaptation to new tasks. Under a mixed linear regression model, we analyze the generalization properties of MAML trained with SGD in the overparameterized regime. We provide both upper and lower bounds for the excess risk of MAML, which captures how SGD dynamics affect these generalization bounds. With such sharp characterizations, we further explore how various learning parameters impact the generalization capability of overparameterized MAML, including explicitly identifying typical data and task distributions that can achieve diminishing generalization error with overparameterization, and characterizing the impact of adaptation learning rate on both excess risk and the early stopping time. Our theoretical findings are further validated by experiments.

## 1 Introduction

Meta-learning [22] is a learning paradigm which aims to design algorithms that are capable of gaining knowledge from many previous tasks and then using it to improve the performance on future tasks efficiently. It has exhibited great power in various machine learning applications spanning over few-shot image classification [31, 32], reinforcement learning [21] and intelligent medicine [20].

One prominent type of meta-learning approaches is an optimization-based method, Model-Agnostic Meta-Learning (MAML) [16], which achieves impressive results in different tasks [30, 4, 2]. The idea of MAML is to learn a good initialization $\omega^*$, such that for a new task we can adapt quickly to a good task parameter starting from $\omega^*$. MAML takes a bi-level implementation: the inner-level initializes at the meta parameter and takes task-specific updates using a few steps of gradient descent (GD), and the outer-level optimizes the meta parameter across all tasks.

With the superior empirical success, theoretical justifications have been provided for MAML and its variants over the past few years from both optimization [18, 38, 14, 25] and generalization perspectives [1, 11, 15, 9]. However, most existing analyses did not take overparameterization into

---

[*]Corresponding author

36th Conference on Neural Information Processing Systems (NeurIPS 2022).

consideration, which we deem as crucial to demystify the remarkable generalization ability of deep meta-learning [41, 22]. More recently, [36] studied the MAML with overparameterized deep neural nets and derived a complexity-based bound to quantify the difference between the empirical and population loss functions at their optimal solutions. However, complexity-based generalization bounds tend to be weak in the high dimensional, especially in the overparameterized regime. Recent works [6, 43] developed more precise bounds for overparameterized setting under a mixed linear regression model, and identified the effect of adaptation learning rate on the generalization. Yet, they considered only the simple isotropic covariance for data and tasks, and did not explicitly capture how the generalization performance of MAML depends on the data and task distributions. Therefore, the following important problem still remains largely open:

*Can **overparameterized** MAML generalize well to a new task, under general data and task distributions?*

In this work, we utilize the mixed linear regression, which is widely adopted in theoretical studies for meta-learning [27, 6, 12, 3], as a proxy to address the above question. In particular, we assume that each task $\tau$ is a noisy linear regression and the associated weight vector is sampled from a common distribution. Under this model, we consider one-step MAML meta-trained with stochastic gradient descent (SGD), where we minimize the loss evaluated at single GD step further ahead for each task. Such settings correspond to real-world implementations of MAML [17, 28, 22] and are extensively considered in theoretical analysis [14, 8, 15]. The focus of this work is the overparameterized regime, i.e., the data dimension $d$ is far larger than the meta-training iterations $T$ ($d \gg T$).

## 1.1 Our Contributions

Our goal is to characterize the generalization behaviours of the MAML output in the overparameterized regime, and to explore how different problem parameters, such as data and task distributions, the adaptation learning rate $\beta^{\text{tr}}$, affect the test error. The main contributions are highlighted below.

- Our first contribution is a sharp characterization (both upper and lower bounds) of the excess risk of MAML trained by SGD. The results are presented in a general manner, which depend on a new notion of effective meta weight, data spectrum, task covariance matrix, and other hyperparameters such as training and test learning rates. In particular, the **effective meta weight** captures an essential property of MAML, where the inner-loop gradient updates have distinctive effects on different dimensions of data eigenspace, i.e., the importance of "leading" space will be magnified whereas the "tail" space will be suppressed.

- We investigate the influence of data and task distributions on the excess risk of MAML. For $\log$-decay data spectrum, our upper and lower bounds establish a sharp phase transition of the generalization. Namely, the excess risk vanishes for large $T$ (where benign fitting occurs) if the data spectrum decay rate is faster than the task diversity rate, and non-vanishing risk occurs otherwise. In contrast, for polynomial or exponential data spectrum decays, excess risk always vanishes for large $T$ irrespective of the task diversity spectrum.

- We showcase the important role the adaptation learning rate $\beta^{\text{tr}}$ plays in the excess risk and the early stopping time of MAML. We provably identify a novel tradeoff between the different impacts of $\beta^{\text{tr}}$ on the "leading" and "tail" data spectrum spaces as the main reason behind the phenomena that the excess risk will first increase then decrease as $\beta^{\text{tr}}$ changes from negative to positive values under general data settings. This complements the explanation based only on the "leading" data spectrum space given in [6] for the isotropic case. We further theoretically illustrate that $\beta^{\text{tr}}$ plays a similar role in determining the early stopping time, i.e., the iteration at which MAML achieves steady generalization error.

**Notations.** We will use bold lowercase and capital letters for vectors and matrices respectively. $\mathcal{N}\left(0, \sigma^2\right)$ denotes the Gaussian distribution with mean $0$ and variance $\sigma^2$. We use $f(x) \lesssim g(x)$ to denote the case $f(x) \leq cg(x)$ for some constant $c > 0$. We use the standard big-O notation and its variants: $\mathcal{O}(\cdot), \Omega(\cdot)$, where $T$ is the problem parameter that becomes large. Occasionally, we use the symbol $\widetilde{\mathcal{O}}(\cdot)$ to hide $\text{polylog}(T)$ factors. $\mathbf{1}_{(\cdot)}$ denotes the indicator function. Let $x^+ = \max\{x, 0\}$.

## 2 Related Work

**Optimization theory for MAML-type approaches** Theoretical guarantee of MAML was initially provided in [17] by proving a universal approximation property under certain conditions. One line of theoretical works have focused on the optimization perspective. [14] established the convergence guarantee of one-step MAML for general nonconvex functions, and [25] extended such results to the multi-step setting. [18] analyzed the regret bound for online MAML. [38, 36] studied the global optimality of MAML with sufficiently wide deep neural nets (DNN). Recently, [10] studied MAML from a representation point of view, and showed that MAML can provably recover the ground-truth subspace.

**Statistical theory for MAML-type approaches.** One line of theoretical analyses lie in the statistical aspect. [15] studied the generalization of MAML on recurring and unseen tasks. Information theory-type generalization bounds for MAML were developed in [26, 9]. [8] characterized the gap of generalization error between MAML and Bayes MAML. [36] provided the statistical error bound for MAML with overparameterized DNN. Our work falls into this category, where the overparameterization has been rarely considered in previous works. Note that [36] only derived the generalization bound from the complexity-based perspective to study the difference between the empirical and population losses for the obtained optimization solutions. Such complexity bound is typically related to the data dimension [29] and may yield vacuous bound in the high dimensional regime. However, our work show that the generalization error of MAML can be small even the data dimension is sufficiently large.

**Overparamterized meta-learning.** [13, 34] studied overparameterized meta-learning from a representation learning perspective. The most relevant papers to our work are [43, 6], where they derived the population risk in overparameterized settings to show the effect of the adaptation learning rate for MAML. Our analysis differs from these works from two essential perspectives: i). we analyze the excess risk of MAML based on the optimization trajectory of SGD in non-asymptotic regime, highlighting the dependence of iterations $T$, while they directly solved the MAML objective asymptotically; ii). [43, 6] mainly focused on the simple isotropic case for data and task covariance, while we explicitly explore the role of data and task distributions under general settings.

**Overparameterized linear model.** There has been several recent progress in theoretical understanding of overparameterized linear model under different scenarios, where the main goal is to provide non-asymptotic generalization guarantees, such as studies of linear regression [5], ridge regression [35], constant-stepsize SGD [42], decaying-stepsize SGD [39], GD [40], Gaussian Mixture models [37]. This paper aims to derive the non-asymptotic excess risk bound for MAML under mixed linear model, which can be independent of data dimension $d$ and still converge as the iteration $T$ enlarges.

## 3 Preliminary

### 3.1 Meta Learning Formulation

In this work, we consider a standard meta-learning setting [15], where a number of tasks share some similarities, and the learner aims to find a good model prior by leveraging task similarities, so that the learner can quickly find a desirable model for a new task by adapting from such an initial prior.

**Learning a proper initialization.** Suppose we are given a collection of tasks $\{\tau_t\}_{t=1}^T$ sampled from some distribution $\mathcal{T}$. For each task $\tau_t$, we observe $N$ samples $\mathcal{D}_t \triangleq (\mathbf{X}_t, \mathbf{y}_t) = \left\{ (\mathbf{x}_{t,j}, y_{t,j}) \in \mathbb{R}^d \times \mathbb{R} \right\}_{j \in [N]} \overset{i.i.d.}{\sim} \mathbb{P}_{\phi_t}(y|\mathbf{x})\mathbb{P}(\mathbf{x})$, where $\phi_t$ is the model parameter for the $t$-th task. The collection of $\{\mathcal{D}_t\}_{t=1}^T$ is denoted as $\mathcal{D}$. Suppose that $\mathcal{D}_t$ is randomly split into training and validation sets, denoted respectively as $\mathcal{D}_t^{\text{in}} \triangleq (\mathbf{X}_t^{\text{in}}, \mathbf{y}_t^{\text{in}})$ and $\mathcal{D}_t^{\text{out}} \triangleq (\mathbf{X}_t^{\text{out}}, \mathbf{y}_t^{\text{out}})$, correspondingly containing $n_1$ and $n_2$ samples (i.e., $N = n_1 + n_2$). We let $\boldsymbol{\omega} \in \mathbb{R}^d$ denote the initialization variable. Each task $\tau_t$ applies an inner algorithm $\mathcal{A}$ with such an initial and obtains an output $\mathcal{A}(\boldsymbol{\omega}; \mathcal{D}_t^{\text{in}})$. Thus, the adaptation performance of $\boldsymbol{\omega}$ for task $\tau_t$ can be measured by the mean squared loss over the validation set given by $\ell(\mathcal{A}(\boldsymbol{\omega}; \mathcal{D}_t^{\text{in}}); \mathcal{D}_t^{\text{out}}) := \frac{1}{2n_2} \sum_{j=1}^{n_2} \left( \langle \mathbf{x}_{t,j}^{\text{out}}, \mathcal{A}(\boldsymbol{\omega}; \mathcal{D}_t^{\text{in}}) \rangle - y_{t,j}^{\text{out}} \right)^2$. The goal of meta-learning is to find an optimal initialization $\widehat{\boldsymbol{\omega}}^* \in \mathbb{R}^d$ by minimizing the following empirical

meta-training loss:

$$\min_{\boldsymbol{\omega} \in \mathbb{R}^d} \widehat{\mathcal{L}}(\mathcal{A}, \boldsymbol{\omega}; \mathcal{D}) \quad \text{where} \quad \widehat{\mathcal{L}}(\mathcal{A}, \boldsymbol{\omega}; \mathcal{D}) = \frac{1}{T} \sum_{t=1}^{T} \ell(\mathcal{A}(\boldsymbol{\omega}; \mathcal{D}_t^{\text{in}}); \mathcal{D}_t^{\text{out}}). \tag{1}$$

In the testing process, suppose a new task $\tau$ sampled from $\mathcal{T}$ is given, which is associated with the dataset $\mathcal{Z}$ consisting of $m$ points with the task. We apply the learned initial $\widehat{\boldsymbol{\omega}}^*$ as well as the inner algorithm $\mathcal{A}$ on $\mathcal{Z}$ to produce a task predictor. Then the test performance can be evaluated via the following population loss:

$$\mathcal{L}(\mathcal{A}, \boldsymbol{\omega}) = \mathbb{E}_{\tau \sim \mathcal{T}} \mathbb{E}_{\mathcal{Z}, (\mathbf{x}, y) \sim \mathbb{P}_\phi(y|\mathbf{x})\mathbb{P}(\mathbf{x})} \left[ \ell\left(\mathcal{A}\left(\boldsymbol{\omega}; \mathcal{Z}\right); (\mathbf{x}, y)\right) \right]. \tag{2}$$

**Inner Loop with one-step GD.** Our focus of this paper is the popular meta-learning algorithm MAML [16], where inner stage takes a few steps of GD update initialized from $\boldsymbol{\omega}$. We consider one step for simplicity, which is commonly adopted in the previous studies [6, 10, 19]. Formally, for any $\boldsymbol{\omega} \in \mathbb{R}^d$, and any dataset $(\mathbf{X}, \mathbf{y})$ with $n$ samples, the inner loop algorithm for MAML with a learning rate $\beta$ is given by

$$\mathcal{A}(\boldsymbol{\omega}; (\mathbf{X}, \mathbf{y})) := \boldsymbol{\omega} - \beta \nabla_{\boldsymbol{\omega}} \ell\left(\boldsymbol{\omega}; (\mathbf{X}, \mathbf{y})\right) = (\mathbf{I} - \frac{\beta}{n} \mathbf{X}^\top \mathbf{X}) \boldsymbol{\omega} + \frac{\beta}{n} \mathbf{X}^\top \mathbf{y}. \tag{3}$$

We allow the learning rate to differ at the meta-training and testing stages, denoted as $\beta^{\text{tr}}$ and $\beta^{\text{te}}$ respectively. Moreover, in subsequent analysis, we will include the dependence on the learning rate to the inner loop algorithm and loss functions as $\mathcal{A}(\boldsymbol{\omega}, \beta; (\mathbf{X}, \mathbf{y}))$, $\widehat{\mathcal{L}}(\mathcal{A}, \boldsymbol{\omega}, \beta; \mathcal{D})$ and $\mathcal{L}(\mathcal{A}, \boldsymbol{\omega}, \beta)$.

**Outer Loop with SGD.** We adopt SGD to iteratively update the meta initialization variable $\boldsymbol{\omega}$ based on the empirical meta-training loss eq. (1), which is how MAML is implemented in practice [17]. Specifically, we use the constant stepsize SGD with iterative averaging [15, 12, 11], and the algorithm is summarized in Algorithm 1. Note that at each iteration, we use one task for updating the meta parameter, which can be easily generalized to the case with a mini-batch tasks for each iteration.

---

**Algorithm 1** MAML with SGD

---

**Input:** Stepsize $\alpha > 0$, meta learning rate $\beta^{\text{tr}} > 0$
**Initialization:** $\boldsymbol{\omega}_0$
   **for** $t = 1$ to $T$ **do**
      Receive task $\tau_t$ with data $\mathcal{D}_t$
      Randomly divided into training and validation set: $\mathcal{D}_t^{in} = (\mathbf{X}_t^{in}, \mathbf{y}_t^{in})$, $\mathcal{D}_t^{out} = (\mathbf{X}_t^{out}, \mathbf{y}_t^{out})$
      Update $\boldsymbol{\omega}_{t+1} = \boldsymbol{\omega}_t - \alpha \nabla \ell(\mathcal{A}(\boldsymbol{\omega}, \beta^{\text{tr}}; \mathcal{D}_t^{\text{in}}); \mathcal{D}_t^{\text{out}})$
   **end for**
   **return** $\overline{\boldsymbol{\omega}}_T = \frac{1}{T} \sum_{t=0}^{T-1} \boldsymbol{\omega}_t$

---

**Meta Excess Risk of SGD.** Let $\boldsymbol{\omega}^*$ denote the optimal solution to the population meta-test error eq. (2). We define the following excess risk for the output $\overline{\boldsymbol{\omega}}_T$ of SGD:

$$R(\overline{\boldsymbol{\omega}}_T, \beta^{\text{te}}) \triangleq \mathbb{E} \left[ \mathcal{L}(\mathcal{A}, \overline{\boldsymbol{\omega}}_T, \beta^{\text{te}}) \right] - \mathcal{L}(\mathcal{A}, \boldsymbol{\omega}^*, \beta^{\text{te}}) \tag{4}$$

which identifies the difference between adapting from the SGD output $\overline{\boldsymbol{\omega}}_T$ and from the optimal initialization $\boldsymbol{\omega}^*$. Assuming that each task contains a fixed constant number of samples, the total number of samples over all tasks is $\mathcal{O}(T)$. Hence, the overparameterized regime can be identified as $d \gg T$, which is the focus of this paper, and is in contrast to the well studied underparameterized setting with finite dimension $d$ ($d \ll T$). The goal of this work is to characterize the impact of SGD dynamics, demonstrating how the iteration $T$ affects the excess risk, which has not been considered in the previous overparameterized MAML analysis [6, 43].

### 3.2 Task and Data Distributions

To gain more explicit knowledge of MAML, we specify the task and data distributions in this section.

**Mixed Linear Regression.** We consider a canonical case in which the tasks are linear regressions. This setting has been commonly adopted recently in [6, 3, 27]. Given a task $\tau$, its model parameter $\phi$ is determined by $\boldsymbol{\theta} \in \mathbb{R}^d$, and the output response is generated as follows:

$$y = \boldsymbol{\theta}^\top \mathbf{x} + z, \quad \mathbf{x} \sim \mathcal{P}_{\mathbf{x}}, \quad z \sim \mathcal{P}_z \tag{5}$$

where $\mathbf{x}$ is the input feature, which follows the same distribution $\mathcal{P}_{\mathbf{x}}$ across different tasks, and $z$ is the i.i.d. Gaussian noise sampled from $\mathcal{N}(0, \sigma^2)$. The task signal $\boldsymbol{\theta}$ has the mean $\boldsymbol{\theta}^*$ and the covariance $\Sigma_{\boldsymbol{\theta}} \triangleq \mathbb{E}[\boldsymbol{\theta}\boldsymbol{\theta}^\top]$. Denote the distribution of $\boldsymbol{\theta}$ as $\mathcal{P}_{\boldsymbol{\theta}}$. We do not make any additional assumptions on $\mathcal{P}_{\boldsymbol{\theta}}$, whereas recent studies on MAML [6, 43] assume it to be Gaussian and isotropic.

**Data distribution.** For the data distribution $\mathcal{P}_{\mathbf{x}}$, we first introduce some mild regularity conditions:

1. $\mathbf{x} \in \mathbb{R}^d$ is mean zero with covariance operator $\boldsymbol{\Sigma} = \mathbb{E}[\mathbf{x}\mathbf{x}^\top]$;
2. The spectral decomposition of $\boldsymbol{\Sigma}$ is $\boldsymbol{V}\boldsymbol{\Lambda}\boldsymbol{V}^\top = \sum_{i>0} \lambda_i \boldsymbol{v}_i \boldsymbol{v}_i^\top$, with decreasing eigenvalues $\lambda_1 \geq \cdots \geq \lambda_d > 0$, and suppose $\sum_{i>0} \lambda_i < \infty$.
3. $\boldsymbol{\Sigma}^{-\frac{1}{2}}\mathbf{x}$ is $\sigma_{\mathbf{x}}$-subGaussian.

To analyze the stochastic approximation method SGD, we take the following standard fourth moment condition [42, 24, 7].

**Assumption 1** (Fourth moment condition). *There exist positive constants $c_1, b_1 > 0$, such that for any positive semidefinite (PSD) matrix $\mathbf{A}$, it holds that*

$$b_1 \operatorname{tr}(\boldsymbol{\Sigma}\mathbf{A})\boldsymbol{\Sigma} + \boldsymbol{\Sigma}\mathbf{A}\boldsymbol{\Sigma} \preceq \mathbb{E}_{\mathbf{x} \sim \mathcal{P}_{\mathbf{x}}}\left[\mathbf{x}\mathbf{x}^\top \mathbf{A}\mathbf{x}\mathbf{x}^\top\right] \preceq c_1 \operatorname{tr}(\boldsymbol{\Sigma}\mathbf{A})\boldsymbol{\Sigma}$$

*For the Gaussian distribution, it suffices to take $c_1 = 3, b_1 = 2$.*

### 3.3 Connection to a Meta Least Square Problem.

After instantiating our study on the task and data distributions in the last section, note that $\nabla\ell(\mathcal{A}(\boldsymbol{\omega}, \beta^{\text{tr}}; \mathcal{D}_t^{\text{in}}); \mathcal{D}_t^{\text{out}})$ is linear with respect to $\boldsymbol{\omega}$. Hence, we can reformulate the problem eq. (1) as a least square (LS) problem with transformed meta inputs and output responses.

**Proposition 1** (Meta LS Problem). *Under the mixed linear regression model, the expectation of the meta-training loss eq. (1) taken over task and data distributions can be rewritten as:*

$$\mathbb{E}\left[\widehat{\mathcal{L}}(\mathcal{A}, \boldsymbol{\omega}, \beta^{\text{tr}}; \mathcal{D})\right] = \mathcal{L}(\mathcal{A}, \boldsymbol{\omega}, \beta^{\text{tr}}) = \mathbb{E}_{\mathbf{B}, \boldsymbol{\gamma}} \frac{1}{2}\left[\|\mathbf{B}\boldsymbol{\omega} - \boldsymbol{\gamma}\|^2\right]. \tag{6}$$

*The meta data are given by*

$$\mathbf{B} = \frac{1}{\sqrt{n_2}}\mathbf{X}^{out}\left(\mathbf{I} - \frac{\beta^{tr}}{n_1}\mathbf{X}^{in^T}\mathbf{X}^{in}\right)$$

$$\boldsymbol{\gamma} = \frac{1}{\sqrt{n_2}}\left(\mathbf{X}^{out}\left(\mathbf{I} - \frac{\beta^{tr}}{n_1}\mathbf{X}^{in^T}\mathbf{X}^{in}\right)\boldsymbol{\theta} + \mathbf{z}^{out} - \frac{\beta^{tr}}{n_1}\mathbf{X}^{out}\mathbf{X}^{in^\top}\mathbf{z}^{in}\right) \tag{7}$$

*where $\mathbf{X}^{in} \in \mathbb{R}^{n_1 \times d}, \mathbf{z}^{in} \in \mathbb{R}^{n_1}, \mathbf{X}^{out} \in \mathbb{R}^{n_2 \times d}$ and $\mathbf{z}^{out} \in \mathbb{R}^{n_2}$ denote the inputs and noise for training and validation. Furthermore, we have*

$$\boldsymbol{\gamma} = \mathbf{B}\boldsymbol{\theta}^* + \boldsymbol{\xi} \quad \text{with meta noise } \mathbb{E}[\boldsymbol{\xi} \mid \mathbf{B}] = 0. \tag{8}$$

Therefore, the meta-training objective is equivalent to searching for a $\boldsymbol{\omega}$, which is close to the task mean $\boldsymbol{\theta}^*$. Moreover, with the specified data and task model, the optimal solution for meta-test loss eq. (2) can be directly calculated [19], and we obtain $\boldsymbol{\omega}^* = \mathbb{E}[\boldsymbol{\theta}] = \boldsymbol{\theta}^*$. Hence, the meta excess risk eq. (4) is identical to the standard excess risk [5] for the linear model eq. (8), i.e., $R(\overline{\boldsymbol{\omega}}_T, \beta^{\text{te}}) = \mathbb{E}_{\mathbf{B}, \boldsymbol{\gamma}}\frac{1}{2}\left[\|\mathbf{B}\overline{\boldsymbol{\omega}}_T - \boldsymbol{\gamma}\|^2 - \|\mathbf{B}\boldsymbol{\theta}^* - \boldsymbol{\gamma}\|^2\right]$, but with more complicated input and output data expressions. The following analysis will focus on this transformed linear model.

Furthermore, we can calculate the statistical properties of the reformed input $\mathbf{B}$, and obtain the meta-covariance:

$$\mathbb{E}[\mathbf{B}^\top \mathbf{B}] = (\mathbf{I} - \beta^{\text{tr}}\boldsymbol{\Sigma})^2\boldsymbol{\Sigma} + \frac{\beta^{\text{tr}2}}{n_1}(\mathbf{F} - \boldsymbol{\Sigma}^3)$$

where $\mathbf{F} = \mathbb{E}[\mathbf{x}\mathbf{x}^\top \Sigma \mathbf{x}\mathbf{x}^\top]$. Let $\mathbf{X} \in \mathbb{R}^{n \times d}$ denote the collection of $n$ i.i.d. samples from $\mathcal{P}_\mathbf{x}$, and denote

$$\mathbf{H}_{n,\beta} = \mathbb{E}[(\mathbf{I} - \frac{\beta}{n}\mathbf{X}^\top\mathbf{X})\Sigma(\mathbf{I} - \frac{\beta}{n}\mathbf{X}^\top\mathbf{X})] = (\mathbf{I} - \beta\Sigma)^2\Sigma + \frac{\beta^2}{n}(\mathbf{F} - \Sigma^3).$$

We can then write $\mathbb{E}[\mathbf{B}^\top\mathbf{B}] = \mathbf{H}_{n_1,\beta^{\mathrm{tr}}}$. Regarding the form of $\mathbf{B}$ and $\mathbf{H}_{n_1,\beta^{\mathrm{tr}}}$, we need some further conditions on the higher order moments of the data distribution.

**Assumption 2** (Commutity). $\mathbf{F} = \mathbb{E}[\mathbf{x}\mathbf{x}^\top\Sigma\mathbf{x}\mathbf{x}^\top]$ *commutes with the data covariance* $\Sigma$.

Assumption 2 holds for Gaussian data. Such commutity of $\Sigma$ has also been considered in [42].

**Assumption 3** (Higher order moment condition). *Given* $|\beta| < \frac{1}{\lambda_1}$ *and* $\Sigma$, *there exists a constant* $C(\beta, \Sigma) > 0$, *for large* $n > 0$, *s.t. for any unit vector* $\mathbf{v} \in \mathbb{R}^d$, *we have:*

$$\mathbb{E}[\|\mathbf{v}^\top\mathbf{H}_{n,\beta}^{-\frac{1}{2}}(\mathbf{I} - \frac{\beta}{n}\mathbf{X}^\top\mathbf{X})\Sigma(\mathbf{I} - \frac{\beta}{n}\mathbf{X}^\top\mathbf{X})\mathbf{H}_{n,\beta}^{-\frac{1}{2}}\mathbf{v}\|^2] < C(\beta, \Sigma). \tag{9}$$

In Assumption 3, the analytical form of $C(\beta, \Sigma)$ can be derived if $\Sigma^{-\frac{1}{2}}\mathbf{x}$ is Gaussian. Moreover, if $\beta = 0$, then we obtain $C(\beta, \Sigma) = 1$. Further technical discussions are presented in Appendix.

# 4 Main Results

In this section, we present our analyses on generalization properties of MAML optimized by average SGD and derive insights on the effect of various parameters. Specifically, our results consist of three parts. First, we characterize the meta excess risk of MAML trained with SGD. Then, we establish the generalization error bound for various types of data and task distributions, to reveal which kind of overparameterization regarding data and task is essential for diminishing meta excess risk. Finally, we explore how the adaptation learning rate $\beta^{\mathrm{tr}}$ affects the excess risk and the training dynamics.

## 4.1 Performance Bounds

Before starting our results, we first introduce relevant notations and concepts. We define the following rates of interest (See Remark 3 for further discussions)

$$c(\beta, \Sigma) := c_1(1 + 8|\beta|\lambda_1\sqrt{C(\beta, \Sigma)}\sigma_x^2 + 64\sqrt{C(\beta, \Sigma)}\sigma_x^4\beta^2\operatorname{tr}(\Sigma^2))$$

$$f(\beta, n, \sigma, \Sigma, \Sigma_{\boldsymbol{\theta}}) := c(\beta, \Sigma)\operatorname{tr}(\Sigma_{\boldsymbol{\theta}}\Sigma) + 4c_1\sigma^2\sigma_x^2\beta^2\sqrt{C(\beta, \Sigma)}\operatorname{tr}(\Sigma^2) + \sigma^2/n$$

$$g(\beta, n, \sigma, \Sigma, \Sigma_{\boldsymbol{\theta}}) := \sigma^2 + b_1\operatorname{tr}(\Sigma_{\boldsymbol{\theta}}\mathbf{H}_{n,\beta}) + \beta^2\mathbf{1}_{\beta \le 0}b_1\operatorname{tr}(\Sigma^2)/n.$$

Moreover, for a positive semi-definite matrix $\mathbf{H}$, s.t. $\mathbf{H}$ and $\Sigma$ can be diagonalized simultaneously, let $\mu_i(\mathbf{H})$ denote its corresponding eigenvalues for $\mathbf{v}_i$, i.e. $\mathbf{H} = \sum_i \mu_i(\mathbf{H})\mathbf{v}_i\mathbf{v}_i^\top$ (Recall $\mathbf{v}_i$ is the $i$-th eigenvector of $\Sigma$).

We next introduce the following new notion of the *effective meta weight*, which will serve as an important quantity for capturing the generalization of MAML.

**Definition 1** (Effective Meta Weights). *For* $|\beta^{\mathrm{tr}}|, |\beta^{\mathrm{te}}| < 1/\lambda_1$, *given step size* $\alpha$ *and iteration* $T$, *define*

$$\Xi_i(\Sigma, \alpha, T) = \begin{cases} \mu_i(\mathbf{H}_{m,\beta^{\mathrm{te}}})/(T\mu_i(\mathbf{H}_{n_1,\beta^{\mathrm{tr}}})) & \mu_i(\mathbf{H}_{n_1,\beta^{\mathrm{tr}}}) \ge \frac{1}{\alpha T}; \\ T\alpha^2\mu_i(\mathbf{H}_{n_1,\beta^{\mathrm{tr}}})\mu_i(\mathbf{H}_{m,\beta^{\mathrm{te}}}) & \mu_i(\mathbf{H}_{n_1,\beta^{\mathrm{tr}}}) < \frac{1}{\alpha T}. \end{cases} \tag{10}$$

We call $\mu_i(\mathbf{H}_{m,\beta^{\mathrm{te}}})/\mu_i(\mathbf{H}_{n_1,\beta^{\mathrm{tr}}})$ and $\mu_i(\mathbf{H}_{m,\beta^{\mathrm{te}}})\mu_i(\mathbf{H}_{n_1,\beta^{\mathrm{tr}}})$ the **meta ratio** (See Remark 2).

We omit the arguments of the effective meta weight $\Xi_i$ for simplicity in the following analysis.

Our first results characterize matching upper and lower bounds on the meta excess risk of MAML in terms of the effective meta weight.

**Theorem 1** (Upper Bound). *Let* $\omega_i = \langle \boldsymbol{\omega}_0 - \boldsymbol{\theta}^*, \mathbf{v}_i \rangle$. *If* $|\beta^{tr}|, |\beta^{te}| < 1/\lambda_1$, $n_1$ *is large ensuring that* $\mu_i(\mathbf{H}_{n_1,\beta^{tr}}) > 0$, $\forall i$ *and* $\alpha < 1/(c(\beta^{tr}, \Sigma)\operatorname{tr}(\Sigma))$, *then the meta excess risk* $R(\overline{\boldsymbol{\omega}}_T, \beta^{te})$ *is bounded above as follows*

$$R(\overline{\boldsymbol{\omega}}_T, \beta^{te}) \le \text{Bias} + \text{Var}$$

*where*

$$Bias = \frac{2}{\alpha^2 T} \sum_i \Xi_i \frac{\omega_i^2}{\mu_i(\mathbf{H}_{n_1,\beta^{tr}})}$$

$$Var = \frac{2}{(1 - \alpha c(\beta^{tr}, \mathbf{\Sigma}) \operatorname{tr}(\mathbf{\Sigma}))} \left( \sum_i \Xi_i \right)$$

$$\times [\underbrace{f(\beta^{tr}, n_2, \sigma, \mathbf{\Sigma_\theta}, \mathbf{\Sigma})}_{V_1} + \underbrace{2c(\beta^{tr}, \mathbf{\Sigma}) \sum_i \left( \frac{\mathbf{1}_{\mu_i(\mathbf{H}_{n_1,\beta^{tr}}) \geq \frac{1}{\alpha T}}}{T\alpha\mu_i(\mathbf{H}_{n_1,\beta^{tr}})} + \mathbf{1}_{\mu_i(\mathbf{H}_{n_1,\beta^{tr}}) < \frac{1}{\alpha T}} \right) \lambda_i \omega_i^2}_{V_2}]$$

*Remark* 1. The primary error source of the upper bound are two folds. The bias term corresponds to the error if we directly implement GD updates towards the meta objective eq. (6). The variance error is composed of the disturbance of meta noise $\boldsymbol{\xi}$ (the $V_1$ term), and the randomness of SGD itself (the $V_2$ term). Regardless of data or task distributions, for proper stepsize $\alpha$, we can easily derive that the bias term is $\mathcal{O}(\frac{1}{T})$, and the $V_2$ term is also $\mathcal{O}(\frac{1}{T})$, which is dominated by $V_1$ term ($\Omega(1)$). Hence, to achieve the vanishing risk, we need to understand the roles of $\Xi_i$ and $f(\cdot)$

*Remark* 2 (Effective Meta Weights). By Definition 1, we separate the data eigenspace into "**leading**" ($\geq \frac{1}{\alpha T}$) and "**tail**" ($< \frac{1}{\alpha T}$) spectrum spaces with different meta weights. The meta ratios indicate the impact of one-step gradient update. For large $n$, $\mu_i(\mathbf{H}_{n,\beta}) \approx (1 - \beta\lambda_i)^2 \lambda_i$, and hence a larger $\beta^{tr}$ in training will increase the weight for "leading" space and decrease the weight for "tail" space, while a larger $\beta^{te}$ always decreases the weight.

*Remark* 3 (Role of $f(\cdot)$). $f(\cdot)$ in variance term consists of various sources of meta noise $\boldsymbol{\xi}$, including inner gradient updates ($\beta$), task diversity ($\mathbf{\Sigma_\theta}$) and noise from regression tasks ($\sigma$). As mentioned in Remark 1, understanding $f(\cdot)$ is critical in our analysis. Yet, due to the multiple randomness origins, techniques for classic linear regression [42, 24] cannot be directly applied here. Our analysis overcomes such non-trivial challenges. $g(\cdot)$ in Theorem 2 plays a similar role to $f(\cdot)$.

Therefore, Theorem 1 implies that overparameterization is crucial for diminishing risk under the following conditions:

- For $f(\cdot)$: $\operatorname{tr}(\mathbf{\Sigma\Sigma_\theta})$ and $\operatorname{tr}(\mathbf{\Sigma}^2)$ is small compared to $T$;
- For $\Xi_i$: the dimension of "leading" space is $o(T)$, and the summation of meta ratio over "tail" space is $o(\frac{1}{T})$.

We next provide a lower bound on the meta excess risk, which matches the upper bound in order.

**Theorem 2** (Lower Bound). *Following the similar notations in Theorem 1, Then*

$$R(\overline{\boldsymbol{\omega}}_T, \beta^{te}) \geq \frac{1}{100\alpha^2 T} \sum_i \Xi_i \frac{\omega_i^2}{\mu_i(\mathbf{H}_{n_1,\beta^{tr}})} + \frac{1}{n_2} \cdot \frac{1}{(1 - \alpha c(\beta^{tr}, \mathbf{\Sigma}) \operatorname{tr}(\mathbf{\Sigma}))} \sum_i \Xi_i$$

$$\times [\frac{1}{100} g(\beta^{tr}, n_1, \mathbf{\Sigma}, \mathbf{\Sigma_\theta}) + \frac{b_1}{1000} \sum_i \left( \frac{\mathbf{1}_{\mu_i(\mathbf{H}_{n_1,\beta^{tr}}) \geq \frac{1}{\alpha T}}}{T\alpha\mu_i(\mathbf{H}_{n_1,\beta^{tr}})} + \mathbf{1}_{\mu_i(\mathbf{H}_{n_1,\beta^{tr}}) < \frac{1}{\alpha T}} \right) \lambda_i \omega_i^2].$$

Our lower bound can also be decomposed into bias and variance terms as the upper bound. The bias term well matches the upper bound up to absolute constants. The variance term differs from the upper bound only by $\frac{1}{n_2}$, where $n_2$ is the batch size of each task, and is treated as a constant (i.e., does not scale with $T$) [23, 33] in practice. Hence, in the overparameterized regime where $d \gg T$ and $T$ tends to be sufficiently large, the variance term also matches that in the upper bound w.r.t. $T$.

## 4.2 The Effects of Task Diversity

From Theorem 1 and Theorem 2, we observe that the task diversity $\mathbf{\Sigma_\theta}$ in $f(\cdot)$ and $g(\cdot)$ plays a crucial role in the performance guarantees for MAML. In this section, we explore several types of data distributions to further characterize the effects of the task diversity.

We take the single task setting as a comparison with meta-learning, where the task diversity diminishes (tentatively say $\mathbf{\Sigma_\theta} \to \mathbf{0}$), i.e., each task parameter $\boldsymbol{\theta} = \boldsymbol{\theta}^*$. In such a case, it is unnecessary to do one-step gradient in the inner loop and we set $\beta^{tr} = 0$, which is equivalent to directly running

SGD. Formally, the **single task setting** can be described as outputting $\overline{\boldsymbol{\omega}}_T^{\text{sin}}$ with iterative SGD that minimizes $\widehat{\mathcal{L}}(\mathcal{A}, \boldsymbol{\omega}, 0; \mathcal{D})$ with meta linear model as $\boldsymbol{\gamma} = \frac{1}{\sqrt{n_2}}(\mathbf{X}^{out}\boldsymbol{\theta}^* + \mathbf{z}^{\text{out}})$.

Theorem 1 implies that the data spectrum should decay fast, which leads to a small dimension of "leading" space and small meta ratio summation over "tail" space. Let us first consider a relatively slow decaying case: $\lambda_k = k^{-1}\log^{-p}(k+1)$ for some $p > 1$. Applying Theorem 1, we immediately derive the theoretical guarantees for single task:

**Lemma 1** (Single Task). *If $|\beta^{te}| < \frac{1}{\lambda_1}$ and if the spectrum of $\Sigma$ satisfies $\lambda_k = k^{-1}\log^{-p}(k+1)$, then $R(\overline{\boldsymbol{\omega}}_T^{sin}, \beta^{te}) = \mathcal{O}(\frac{1}{\log^p(T)})$*

At the test stage, if we set $\beta^{\text{te}} = 0$, then the meta excess risk for the single task setting, i.e., $R(\overline{\boldsymbol{\omega}}_T^{\text{sin}}, 0)$, is exactly the excess risk in classical linear regression [42]. Lemma 1 can be regarded as a generalized version of Corollary 2.3 in [42], where they provide the upper bound for $R(\overline{\boldsymbol{\omega}}_T^{\text{sin}}, 0)$, while we allow a one-step fine-tuning for testing.

Lemma 1 suggests that the $\log$-decay is sufficient to assure that $R(\overline{\boldsymbol{\omega}}_T^{\text{sin}}, 0)$ is diminishing when $d \gg T$. However, in meta-learning with multi-tasks, the task diversity captured by the task spectral distribution can highly affect the meta excess risk. In the following, our Theorem 1 and Theorem 2 (i.e., upper and lower bounds) establish a sharp phase transition of the generalization for MAML for the same data spectrum considered in Lemma 1, which is in contrast to the single task setting (see Lemma 1), where $\log$-decay data spectrum always yields vanishing excess risk.

**Proposition 2** (MAML, $\log$-Decay Data Spectrum). *Given $|\beta^{tr}|, |\beta^{te}| < \frac{1}{\lambda_1}$, under the same data distribution as in Lemma 1, and the spectrum of $\Sigma_\theta$, denoted as $\nu_i$, satisfies $\nu_k = \log^r(k+1)$ for some $r > 0$, then*

$$R(\overline{\boldsymbol{\omega}}_T, \beta^{te}) = \begin{cases} \Omega(\log^{r-2p+1}(T)) & r \geq 2p-1 \\ \mathcal{O}(\frac{1}{\log^{p-(r-p+1)^+}(T)}) & r < 2p-1 \end{cases}$$

Proposition 2 implies that under $\log$-decay data spectrum parameterized by $p$, the meta excess risk of MAML experiences a phase transition determined by the spectrum parameter $r$. Since large $r$ implies large eigenvalues and high variations for task vectors, we adopt $r$ to measure the diversity of task distributions, and call $r$ as the task diversity in the sequel. While slower task diversity rate $r < 2p-1$ guarantees vanishing excess risk, faster task diversity rate $r \geq 2p-1$ necessarily results in non-vanishing excess risk. Proposition 2 and Lemma 1 together indicate that while $\log$-decay data spectrum always yields benign fitting (vanishing risk) in the single task setting, it can yield non-vanishing risk in meta learning due to fast task diversity rate.

We further validate our theoretical results in Proposition 2 by experiments. We consider the case $p = 2$. As shown in Figure 1a, when $r < 2p-1$, the test error quickly converges to the Bayes error. When $r > 2p-1$, Figure 1b illustrates that MAML already converges on the training samples, but the test error (which is further zoomed in Figure 1c) levels off and does not vanish, showing MAML generalizes poorly when $r > 2p-1$.

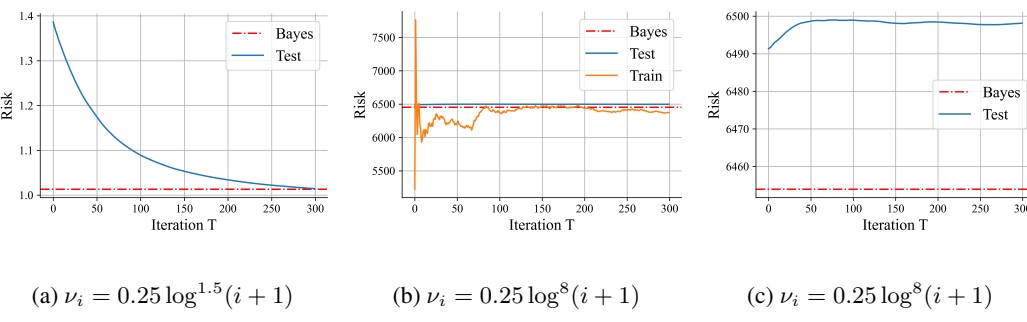

(a) $\nu_i = 0.25\log^{1.5}(i+1)$      (b) $\nu_i = 0.25\log^8(i+1)$      (c) $\nu_i = 0.25\log^8(i+1)$

Figure 1: The effects of task diversity. $d = 500$, $T = 300$, $\lambda_i = \frac{1}{i\log(i+1)^2}$, $\beta^{\text{tr}} = 0.02$, $\beta^{\text{te}} = 0.2$.

Furthermore, we show that the above phase transition that occurs for $\log$-decay data distributions no longer exists for data distributions with faster decaying spectrum.

**Proposition 3** (MAML, Fast-Decay Data Spectrum). *Under the same task distribution as in Proposition 2, i.e., the spectrum of $\Sigma_\theta$, denoted as $\nu_i$, satisfies $\nu_k = \log^r(k+1) = \widetilde{O}(1)$ for some $r > 0$, and the data distribution satisfies:*

1. *$\lambda_k = k^{-q}$ for some $q > 1$, $R(\overline{\omega}_T^{sin}, \beta^{te}) = \mathcal{O}\left(\frac{1}{T^{\frac{q-1}{q}}}\right)$ and $R(\overline{\omega}_T, \beta^{te}) = \widetilde{\mathcal{O}}\left(\frac{1}{T^{\frac{q-1}{q}}}\right)$;*

2. *$\lambda_k = e^{-k}$, $R(\overline{\omega}_T^{sin}, \beta^{te}) = \widetilde{\mathcal{O}}(\frac{1}{T})$ and $R(\overline{\omega}_T, \beta^{te}) = \widetilde{\mathcal{O}}(\frac{1}{T})$.*

### 4.3 On the Role of Adaptation Learning Rate

The analysis in [6] suggests a surprising observation that a negative learning rate (i.e., when $\beta^{tr}$ takes a negative value) optimizes the generalization for MAML under mixed linear regression models. Their results indicate that the testing risk initially increases and then decreases as $\beta^{tr}$ varies from negative to positive values around zero for Gaussian isotropic input data and tasks. Our following proposition supports such a trend, but with a novel tradeoff in SGD dynamics as a new reason for the trend, under more general data distributions. Denote $\overline{\omega}_T^\beta$ as the average SGD solution of MAML after $T$ iterations that uses $\beta$ as the inner loop learning rate.

**Proposition 4.** *Let $s = T \log^{-p}(T)$ and $d = T \log^q(T)$, where $p, q > 0$. Suppose $\mathcal{P}_\mathbf{x}$ is Gaussian and the spectrum of $\Sigma$ satisfies*

$$\lambda_k = \begin{cases} 1/s, & k \leq s \\ 1/(d-s), & s+1 \leq k \leq d. \end{cases}$$

*Suppose the spectral parameter $\nu_i$ of $\Sigma_\theta$ is $O(1)$, and let the step size $\alpha = \frac{1}{2c(\beta^{tr}, \Sigma) \operatorname{tr}(\Sigma)}$. Then for large $n_1$, $|\beta^{tr}|, |\beta^{te}| < \frac{1}{\lambda_1}$, we have*

$$R(\overline{\omega}_T^{\beta^{tr}}, \beta^{te}) \lesssim \mathcal{O}\left(\frac{1}{\log^p(T)}\right)\frac{1}{(1-\beta^{tr}\lambda_1)^2} + \mathcal{O}\left(\frac{1}{\log^q(T)}\right)\left(1 - \beta^{tr}\lambda_d\right)^2 + \widetilde{\mathcal{O}}(\frac{1}{T}). \qquad (11)$$

The first two terms in the bound of eq. (11) correspond to the impact of effective meta weights $\Xi_i$ on the "leading" and "tail" spaces, respectively, as we discuss in Remark 2. Clearly, the learning rate $\beta^{tr}$ plays a tradeoff role in these two terms, particularly when $p$ is close to $q$. This explains the fact that the test error first increases and then decreases as $\beta^{tr}$ varies from negative to positive values around zero. Such a tradeoff also serves as the reason for the first-increase-then-decrease trend of the test error under more general data distributions as we demonstrate in Figure 2. This complements the reason suggested in [6], which captures only the quadratic form $\frac{1}{(1-\beta^{tr}\lambda_1)^2}$ of $\beta^{tr}$ for isotropic $\Sigma$, where there exists only the "leading" space without "tail" space.

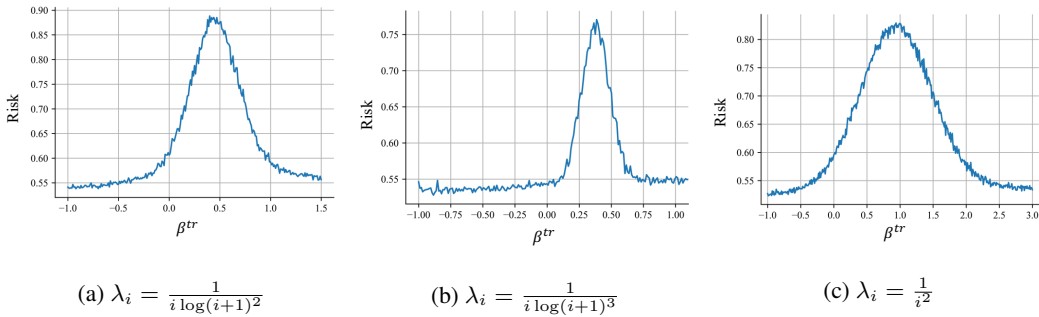

(a) $\lambda_i = \frac{1}{i \log(i+1)^2}$      (b) $\lambda_i = \frac{1}{i \log(i+1)^3}$      (c) $\lambda_i = \frac{1}{i^2}$

Figure 2: $R(\overline{\omega}_T^{\beta^{tr}}, \beta^{te})$ as a function of $\beta^{tr}$. $d = 200$, $T = 100$, $\Sigma_\theta = \frac{0.8^2}{d}\mathbf{I}$, $\beta^{te} = 0.2$.

Based on the above results, incorporating with our dynamics analysis, we surprisingly find that $\beta^{tr}$ not only affects the final risk, but also plays a pivot role towards the early iteration that the testing error tends to be steady. To formally study such a property, we define the stopping time as follows.

**Definition 2** (Stopping time). Given $\beta^{tr}, \beta^{te}$, for any $\epsilon > 0$, the corresponding stopping time $t_\epsilon(\beta^{tr}, \beta^{te})$ is defined as:

$$t_\epsilon(\beta^{tr}, \beta^{te}) = \min t \quad \text{s.t.} \quad R(\overline{\omega}_t^{\beta^{tr}}; \beta^{te}) < \epsilon.$$

In the sequel, we may omit the arguments in $t_\epsilon$ for simplicity. We consider the similar data distribution in Proposition 4 but parameterized by $K$, i.e., $s = K \log^{-p}(K)$ and $d = K \log^q(K)$, where $p, q > 0$. Then we can derive the following characterization for $t_\epsilon$.

**Corollary 1.** *If the assumptions in Proposition 4 hold and $p = q$. Further, let $\Sigma_\theta = \eta^2 \mathbf{I}$, and $|\beta^{tr}| < \frac{1}{\lambda_1}$. Then for $t_\epsilon(\beta^{tr}, \beta^{te}) \in (s, K]$, we have:*

$$\exp\left(\epsilon^{-\frac{1}{p}}\left[\frac{L_l}{(1 - \beta^{tr}\lambda_1)^2} + L_t(1 - \beta^{tr}\lambda_d)^2\right]^{\frac{1}{p}}\right) \leq t_\epsilon$$

$$\leq \exp\left(\epsilon^{-\frac{1}{p}}\left[\frac{U_l}{(1 - \beta^{tr}\lambda_1)^2} + U_t(1 - \beta^{tr}\lambda_d)^2\right]^{\frac{1}{p}}\right) \qquad (12)$$

*where $L_l$, $L_t$, $U_l$, $U_t > 0$ are factors for "leading" and "tail" spaces that are independent of $K$[2].*

Equation (12) suggests that the early stopping time $t_\epsilon$ is also controlled by the tradeoff role that $\beta^{tr}$ plays in the "leading" ($U_l, L_l$) and "tail" spaces ($U_t, L_t$), which takes a similar form as the bound in Proposition 4. Therefore, the trend for $t_\epsilon$ in terms of $\beta^{tr}$ will exhibit similar behaviours as the final excess risk, and hence the optimal $\beta^{tr}$ for the final excess risk will lead to an earliest stopping time. We plot the training and test errors for different $\beta^{tr}$ in Figure 3, under the same data distributions as Figure 2a to validate our theoretical findings. As shown in Figure 3a, $\beta^{tr}$ does not make much difference in the training stage (the process converges for all $\beta^{tr}$ when $T$ is larger than 100). However, in Figure 3b at test stage, $\beta^{tr}$ significantly affects the iteration when the test error starts to become relatively flat. Such an early stopping time first increases then decreases as $\beta^{tr}$ varies from $-0.5$ to $0.7$, which resembles the change of final excess risk in Figure 2a.

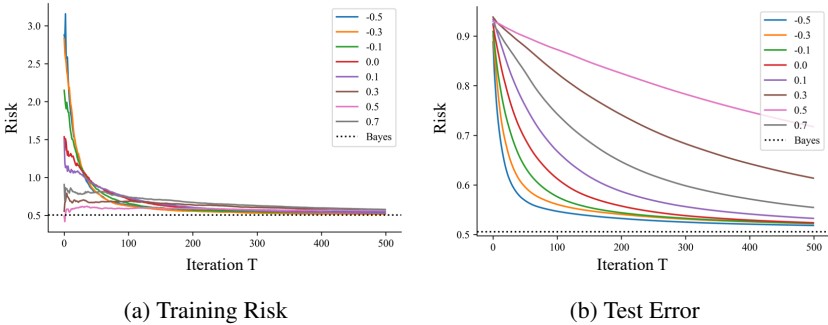

(a) Training Risk  (b) Test Error

Figure 3: Training and test curves for different $\beta^{tr}$. $d = 500$, $\lambda_i = \frac{1}{i \log^2(i+1)}$, $\Sigma_\theta = \frac{0.8^2}{d}\mathbf{I}$, $\beta^{te} = 0.2$.

## 5   Conclusions

In this work, we give the theoretical treatment towards the generalization property of MAML based on their optimization trajectory in non-asymptotic and overparameterized regime. We provide both upper and lower bounds on the excess risk of MAML trained with average SGD. Furthermore, we explore which type of data and task distributions are crucial for diminishing error with overparameterization, and discover the influence of adaption learning rate both on the generalization error and the dynamics, which brings novel insights towards the distinct effects of MAML's one-step gradient updates on "leading" and "tail" parts of data eigenspace.

## Acknowledgments and Disclosure of Funding

The work of Yu Huang and Longbo Huang is supported in part by the Technology and Innovation Major Project of the Ministry of Science and Technology of China under Grant 2020AAA0108400 and 2020AAA0108403, the Tsinghua University Initiative Scientific Research Program, and Tsinghua Precision Medicine Foundation 10001020109. The work of Y. Liang was supported in part by the U.S. National Science Foundation under the grant DMS-2134145.

---

[2]Such terms have been suppressed for clarity. Details are presented in the appendix.

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
