# Provable Generalization of Overparameterized Meta-learning Trained with SGD

**Yu Huang**
IIIS
Tsinghua University
y-huang20@mails.tsinghua.edu.cn

**Yingbin Liang**
Department of ECE
The Ohio State University
liang.889@osu.edu

**Longbo Huang**[*]
IIIS
Tsinghua University
longbohuang@tsinghua.edu.cn

# Appendices

---

[*]Corresponding author

36th Conference on Neural Information Processing Systems (NeurIPS 2022).

## A   Proof of Proposition 1

We first show how to connect the loss function associated with MAML to a Meta Least Square Problem.

**Proposition A.1** (Proposition 1 Restated)**.** *Under the mixed linear regression model, the expectation of the meta-training loss taken over task and data distributions can be rewritten as:*

$$\mathbb{E}\left[\widehat{\mathcal{L}}(\mathcal{A}, \boldsymbol{\omega}, \beta^{tr}; \mathcal{D})\right] = \mathcal{L}(\mathcal{A}, \boldsymbol{\omega}, \beta^{tr}) = \mathbb{E}_{\mathbf{B},\boldsymbol{\gamma}} \frac{1}{2}\left[\|\mathbf{B}\boldsymbol{\omega} - \boldsymbol{\gamma}\|^2\right]. \tag{1}$$

*The meta data are given by*

$$\mathbf{B} = \frac{1}{\sqrt{n_2}}\mathbf{X}^{out}\left(\mathbf{I} - \frac{\beta^{tr}}{n_1}\mathbf{X}^{in^T}\mathbf{X}^{in}\right) \tag{2}$$

$$\boldsymbol{\gamma} = \frac{1}{\sqrt{n_2}}\left(\mathbf{X}^{out}\left(\mathbf{I} - \frac{\beta^{tr}}{n_1}\mathbf{X}^{in^T}\mathbf{X}^{in}\right)\boldsymbol{\theta} + \mathbf{z}^{out} - \frac{\beta^{tr}}{n_1}\mathbf{X}^{out}\mathbf{X}^{in^\top}\mathbf{z}^{in}\right) \tag{3}$$

*where $\mathbf{X}^{in} \in \mathbb{R}^{n_1 \times d}, \mathbf{z}^{in} \in \mathbb{R}^{n_1}, \mathbf{X}^{out} \in \mathbb{R}^{n_2 \times d}$ and $\mathbf{z}^{out} \in \mathbb{R}^{n_2}$ denote the inputs and noise for training and validation. Furthermore, we have*

$$\boldsymbol{\gamma} = \mathbf{B}\boldsymbol{\theta}^* + \boldsymbol{\xi} \quad \text{with meta noise } \mathbb{E}[\boldsymbol{\xi} \mid \mathbf{B}] = 0. \tag{4}$$

*Proof.* We first rewrite $\mathcal{L}(\mathcal{A}, \boldsymbol{\omega}, \beta^{\mathrm{tr}})$ as follows:

$$
\begin{aligned}
\mathcal{L}(\mathcal{A}, \boldsymbol{\omega}, \beta^{\mathrm{tr}}) &= \mathbb{E}\left[\ell(\mathcal{A}(\boldsymbol{\omega}, \beta^{\mathrm{tr}}; \mathcal{D}^{\mathrm{in}}); \mathcal{D}^{\mathrm{out}})\right] \\
&= \mathbb{E}\left[\frac{1}{2n_2}\sum_{j=1}^{n_2}\left(\langle \mathbf{x}_j^{\mathrm{out}}, \mathcal{A}(\boldsymbol{\omega}, \beta^{\mathrm{tr}}; \mathcal{D}^{\mathrm{in}})\rangle - y_j^{\mathrm{out}}\right)^2\right] \\
&= \mathbb{E}\left[\frac{1}{2n_2}\|\mathbf{X}^{\mathrm{out}}\left(\mathbf{I} - \frac{\beta^{\mathrm{tr}}}{n_1}\mathbf{X}^{\mathrm{in}^T}\mathbf{X}^{\mathrm{in}}\right)\boldsymbol{\omega} + \frac{\beta^{\mathrm{tr}}}{n_1}\mathbf{X}^{\mathrm{in}^T}\mathbf{y}^{\mathrm{in}} - \mathbf{y}^{\mathrm{out}}\|^2\right].
\end{aligned}
$$

Using the mixed linear model:

$$\mathbf{y}^{\mathrm{in}} = \mathbf{X}^{\mathrm{in}}\boldsymbol{\theta} + \mathbf{z}^{\mathrm{in}}, \quad \mathbf{y}^{\mathrm{out}} = \mathbf{X}^{\mathrm{out}}\boldsymbol{\theta} + \mathbf{z}^{\mathrm{out}}, \tag{5}$$

we have

$$\mathcal{L}(\mathcal{A}, \boldsymbol{\omega}, \beta^{\text{tr}})$$
$$= \mathbb{E}\left[ \frac{1}{2n_2} \| \mathbf{X}^{\text{out}} \left( \mathbf{I} - \frac{\beta^{\text{tr}}}{n_1} \mathbf{X}^{\text{in}^T} \mathbf{X}^{\text{in}} \right) \boldsymbol{\omega} \right.$$
$$\left. - \left( \mathbf{X}^{\text{out}} \left( \mathbf{I} - \frac{\beta^{\text{tr}}}{n_1} \mathbf{X}^{\text{in}^T} \mathbf{X}^{\text{in}} \right) \boldsymbol{\theta} + \mathbf{z}^{out} - \frac{\beta^{\text{tr}}}{n_1} \mathbf{X}^{\text{out}} \mathbf{X}^{\text{in}^\top} \mathbf{z}^{in} \right) \|^2 \right]$$
$$= \mathbb{E}_{\mathbf{B}, \boldsymbol{\gamma}} \frac{1}{2} \left[ \| \mathbf{B}\boldsymbol{\omega} - \boldsymbol{\gamma} \|^2 \right].$$

Moreover, note that $\boldsymbol{\theta} - \boldsymbol{\theta}^*$ has mean zero and is independent of data and noise, and define

$$\boldsymbol{\xi} = \frac{1}{\sqrt{n_2}} \left( \mathbf{X}^{\text{out}} \left( \mathbf{I} - \frac{\beta^{\text{tr}}}{n_1} \mathbf{X}^{\text{in}^T} \mathbf{X}^{\text{in}} \right) (\boldsymbol{\theta} - \boldsymbol{\theta}^*) + \mathbf{z}^{\text{out}} - \frac{\beta^{\text{tr}}}{n_1} \mathbf{X}^{\text{out}} \mathbf{X}^{\text{in}^\top} \mathbf{z}^{in} \right). \tag{6}$$

We call $\boldsymbol{\xi}$ as meta noise, and then we have

$$\boldsymbol{\gamma} = \mathbf{B}\boldsymbol{\theta}^* + \boldsymbol{\xi} \quad \text{and} \quad \mathbb{E}[\boldsymbol{\xi} \mid \mathbf{B}] = 0.$$

$\square$

**Lemma A.1** (Meta Excess Risk). *Under the mixed linear regression model, the meta excess risk can be rewritten as follows:*

$$R(\overline{\boldsymbol{\omega}}_T, \beta^{te}) = \frac{1}{2} \mathbb{E} \| \overline{\boldsymbol{\omega}}_T - \boldsymbol{\theta}^* \|^2_{\mathbf{H}_{m, \beta^{te}}}$$

*where $\|\mathbf{a}\|^2_{\mathbf{A}} = \mathbf{a}^T \mathbf{A} \mathbf{a}$. Moreover, the Bayes error is given by*

$$\mathcal{L}(\mathcal{A}, \boldsymbol{\omega}^*, \beta^{te}) = \frac{1}{2} \operatorname{tr}(\boldsymbol{\Sigma}_{\boldsymbol{\theta}} \mathbf{H}_{m, \beta^{te}}) + \frac{\sigma^2 \beta^{te 2}}{2m} + \frac{\sigma^2}{2}.$$

*Proof.* Recall that

$$R(\overline{\boldsymbol{\omega}}_T, \beta^{te}) \triangleq \mathbb{E}\left[ \mathcal{L}(\mathcal{A}, \overline{\boldsymbol{\omega}}_T, \beta^{te}) \right] - \mathcal{L}(\mathcal{A}, \boldsymbol{\omega}^*, \beta^{te})$$

where $\boldsymbol{\omega}^*$ denotes the optimal solution to the population meta-test error. Under the mixed linear model, such a solution can be directly calculated [11], and we obtain $\boldsymbol{\omega}^* = \mathbb{E}[\boldsymbol{\theta}] = \boldsymbol{\theta}^*$. Hence,

$$R(\overline{\boldsymbol{\omega}}_T, \beta^{te}) = \mathbb{E}_{\mathbf{B}, \boldsymbol{\gamma}} \frac{1}{2} \left[ \| \mathbf{B}\overline{\boldsymbol{\omega}}_T - \boldsymbol{\gamma} \|^2 - \| \mathbf{B}\boldsymbol{\theta}^* - \boldsymbol{\gamma} \|^2 \right],$$

where

$$\mathbf{B} = \mathbf{x}^{\text{out}^\top} \left( \mathbf{I} - \frac{\beta^{\text{te}}}{m} \mathbf{X}^{\text{in}^T} \mathbf{X}^{\text{in}} \right)$$
$$\boldsymbol{\gamma} = \mathbf{x}^{\text{out}^\top} \left( \mathbf{I} - \frac{\beta^{\text{te}}}{m} \mathbf{X}^{\text{in}^T} \mathbf{X}^{\text{in}} \right) \boldsymbol{\theta} + \mathbf{z}^{\text{out}} - \frac{\beta^{\text{te}}}{m} \mathbf{x}^{\text{out}^\top} \mathbf{X}^{\text{in}^\top} \mathbf{z}^{in}, \tag{7}$$

and $\mathbf{x}^{\text{out}} \in \mathbb{R}^d$, $\mathbf{z}^{\text{out}} \in \mathbb{R}^d$, $\mathbf{X}^{\text{in}} \in \mathbb{R}^{m \times d}$ and $\mathbf{z}^{\text{in}} \in \mathbb{R}^m$. The forms of $\mathbf{B}$ and $\boldsymbol{\gamma}$ are slightly different since we allow a new adaptation rate $\beta^{\text{te}}$ and the inner loop has $m$ samples at test stage. Similarly

$$\boldsymbol{\xi} = \left( \underbrace{\mathbf{x}^{\text{out}^\top} \left( \mathbf{I} - \frac{\beta^{\text{te}}}{m} \mathbf{X}^{\text{in}^T} \mathbf{X}^{\text{in}} \right) (\boldsymbol{\theta} - \boldsymbol{\theta}^*)}_{\xi_1} + \underbrace{\mathbf{z}^{\text{out}}}_{\xi_2} \underbrace{- \frac{\beta^{\text{tr}}}{m} \mathbf{x}^{\text{out}^\top} \mathbf{X}^{\text{in}^\top} \mathbf{z}^{in}}_{\xi_3} \right). \tag{8}$$

Then we have

$$R(\overline{\boldsymbol{\omega}}_T, \beta^{\text{te}}) = \mathbb{E}_{\mathbf{B}, \boldsymbol{\gamma}} \frac{1}{2} \left[ \| \mathbf{B}\overline{\boldsymbol{\omega}}_T - \boldsymbol{\gamma} \|^2 - \| \mathbf{B}\boldsymbol{\theta}^* - \boldsymbol{\gamma} \|^2 \right]$$
$$= \mathbb{E}_{\mathbf{B}, \boldsymbol{\gamma}} \frac{1}{2} \left[ \| \mathbf{B}(\overline{\boldsymbol{\omega}}_T - \boldsymbol{\theta}^*) \|^2 \right]$$
$$= \frac{1}{2} \mathbb{E} \| \overline{\boldsymbol{\omega}}_T - \boldsymbol{\theta}^* \|^2_{\mathbf{H}_{m, \beta^{\text{te}}}}$$

where the last equality follows because $\mathbb{E}\left[\mathbf{B}^\top\mathbf{B}\right] = \mathbf{H}_{m,\beta^{\text{te}}}$ at the test stage.

The Bayes error can be calculated as follows:

$$\mathcal{L}(\mathcal{A}, \boldsymbol{\omega}^*, \beta^{\text{te}}) = \mathbb{E}_{\mathbf{B},\boldsymbol{\gamma}} \frac{1}{2}\left[\|\mathbf{B}\boldsymbol{\theta}^* - \boldsymbol{\gamma}\|^2\right] = \mathbb{E}_{\mathbf{B},\boldsymbol{\gamma}}\frac{1}{2}\left[\xi^2\right]$$
$$\overset{(a)}{=} \frac{1}{2}\left(\mathbb{E}\left[\xi_1^2\right] + \mathbb{E}\left[\xi_2^2\right] + \mathbb{E}\left[\xi_3^2\right]\right)$$
$$= \frac{1}{2}(\text{tr}(\boldsymbol{\Sigma_\theta}\mathbf{H}_{m,\beta^{\text{te}}}) + \frac{\beta^{\text{te}\,2}\sigma^2}{m} + \sigma^2)$$

where $(a)$ follows because $\xi_1, \xi_2, \xi_3$ are independent and have zero mean conditioned on $\mathbf{X}^{\text{in}}$ and $\mathbf{x}^{\text{out}}$. $\qquad\square$

# B  Analysis for Upper Bound (Theorem 1)

## B.1  Preliminaries

We first introduce some additional notations.

**Definition 1** (Inner product of matrices). For any two matrices $\mathbf{C}, \mathbf{D}$, the inner product of them is defined as
$$\langle\mathbf{C}, \mathbf{D}\rangle = \text{tr}(\mathbf{C}^\top\mathbf{D}).$$

We will use the following property about the inner product of matrices throughout our proof.

**Property B.1.** *If $\mathbf{C} \succeq 0$ and $\mathbf{D} \succeq \mathbf{D}'$, then we have $\langle\mathbf{C}, \mathbf{D}\rangle \geq \langle\mathbf{C}, \mathbf{D}'\rangle$.*

**Definition 2** (Linear operator). Let $\otimes$ denote the tensor product. Define the following linear operators on symmetric matrices:

$$\mathcal{M} = \mathbb{E}\left[\mathbf{B}^\top \otimes \mathbf{B}^\top \otimes \mathbf{B} \otimes \mathbf{B}\right] \quad \widetilde{\mathcal{M}} := \mathbf{H}_{n_1,\beta^{\text{tr}}} \otimes \mathbf{H}_{n_1,\beta^{\text{tr}}} \quad \mathcal{I} := \mathbf{I} \otimes \mathbf{I}$$
$$\mathcal{T} := \mathbf{H}_{n_1,\beta^{\text{tr}}} \otimes \mathbf{I} + \mathbf{I} \otimes \mathbf{H}_{n_1,\beta^{\text{tr}}} - \alpha\mathcal{M}, \quad \widetilde{\mathcal{T}} = \mathbf{H}_{n_1,\beta^{\text{tr}}} \otimes \mathbf{I} + \mathbf{I} \otimes \mathbf{H}_{n_1,\beta^{\text{tr}}} - \alpha\mathbf{H}_{n_1,\beta^{\text{tr}}} \otimes \mathbf{H}_{n_1,\beta^{\text{tr}}}.$$

We next define the operation of the above linear operators on a symmetric matrix $\mathbf{A}$ as follows.

$$\mathcal{M} \circ \mathbf{A} = \mathbb{E}\left[\mathbf{B}^\top\mathbf{B}\mathbf{A}\mathbf{B}^\top\mathbf{B}\right], \quad \widetilde{\mathcal{M}} \circ \mathbf{A} = \mathbf{H}_{n_1,\beta^{\text{tr}}}\mathbf{A}\mathbf{H}_{n_1,\beta^{\text{tr}}}, \quad \mathcal{I} \circ \mathbf{A} = \mathbf{A},$$
$$\mathcal{T} \circ \mathbf{A} = \mathbf{H}_{n_1,\beta^{\text{tr}}}\mathbf{A} + \mathbf{A}\mathbf{H}_{n_1,\beta^{\text{tr}}} - \alpha\mathbb{E}\left[\mathbf{B}^\top\mathbf{B}\mathbf{A}\mathbf{B}^\top\mathbf{B}\right]$$
$$\widetilde{\mathcal{T}} \circ \mathbf{A} = \mathbf{H}_{n_1,\beta^{\text{tr}}}\mathbf{A} + \mathbf{A}\mathbf{H}_{n_1,\beta^{\text{tr}}} - \alpha\mathbf{H}_{n_1,\beta^{\text{tr}}}\mathbf{A}\mathbf{H}_{n_1,\beta^{\text{tr}}}.$$

Based on the above definitions, we have the following equations hold.

$$(\mathcal{I} - \alpha\mathcal{T}) \circ \mathbf{A} = \mathbb{E}\left[\left(\mathbf{I} - \alpha\mathbf{B}^\top\mathbf{B}\right)\mathbf{A}\left(\mathbf{I} - \alpha\mathbf{B}^\top\mathbf{B}\right)\right]$$
$$(\mathcal{I} - \alpha\widetilde{\mathcal{T}}) \circ \mathbf{A} = (\mathbf{I} - \alpha\mathbf{H}_{n_1,\beta^{\text{tr}}})\mathbf{A}(\mathbf{I} - \alpha\mathbf{H}_{n_1,\beta^{\text{tr}}}).$$

For the linear operators, we have the following technical lemma.

**Lemma B.1.** *We call the linear operator $\mathcal{O}$ a PSD mapping, if for every symmetric PSD matrix $\mathbf{A}$, $\mathcal{O} \circ \mathbf{A}$ is also PSD matrix. Then we have:*

  *(i) $\mathcal{M}, \widetilde{\mathcal{M}}$ and $(\mathcal{M} - \widetilde{\mathcal{M}})$ are all PSD mappings.*

  *(ii) $\widetilde{\mathcal{T}} - \mathcal{T}, \mathcal{I} - \alpha\mathcal{T}$ and $\mathcal{I} - \alpha\widetilde{\mathcal{T}}$ are all PSD mappings.*

  *(iii) If $0 < \alpha < \frac{1}{\max_i\{\mu_i(\mathbf{H}_{n_1,\beta^{tr}})\}}$, then $\widetilde{\mathcal{T}}^{-1}$ exists, and is a PSD mapping.*

  *(iv) If $0 < \alpha < \frac{1}{\max_i\{\mu_i(\mathbf{H}_{n_1,\beta^{tr}})\}}$, $\widetilde{\mathcal{T}}^{-1} \circ \mathbf{H}_{n_1,\beta^{tr}} \preceq \mathbf{I}$.*

  *(v) If $0 < \alpha < \frac{1}{c(\beta^{tr},\boldsymbol{\Sigma})\,\text{tr}(\boldsymbol{\Sigma})}$, then $\mathcal{T}^{-1} \circ \mathbf{A}$ exists for PSD matrix $\mathbf{A}$, and $\mathcal{T}^{-1}$ is a PSD mapping.*

*Proof.* Items (i) and (iii) directly follow from the proofs in [14, 22]. For $(iv)$, by the existence of $\widetilde{\mathcal{T}}^{-1}$, we have

$$\widetilde{\mathcal{T}}^{-1} \circ \mathbf{H}_{n_1,\beta^{\text{tr}}} = \sum_{t=0}^{\infty} \alpha (\mathcal{I} - \alpha\widetilde{\mathcal{T}})^t \circ \mathbf{H}_{n_1,\beta^{\text{tr}}}$$

$$= \sum_{t=0}^{\infty} \alpha (\mathbf{I} - \alpha\mathbf{H}_{n_1,\beta^{\text{tr}}})^t \mathbf{H}_{n_1,\beta^{\text{tr}}} (\mathbf{I} - \alpha\mathbf{H}_{n_1,\beta^{\text{tr}}})^t$$

$$\preceq \sum_{t=0}^{\infty} \alpha (\mathbf{I} - \alpha\mathbf{H}_{n_1,\beta^{\text{tr}}})^t \mathbf{H}_{n_1,\beta^{\text{tr}}} = \mathbf{I}.$$

For $(v)$, for any PSD matrix $\mathbf{A}$, consider

$$\mathcal{T}^{-1} \circ \mathbf{A} = \alpha \sum_{k=0}^{\infty} (\mathcal{I} - \alpha\mathcal{T})^k \circ \mathbf{A}.$$

We first show that $\sum_{k=0}^{\infty} (\mathcal{I} - \alpha\mathcal{T})^k \circ \mathbf{A}$ is finite, and then it suffices to show that the trace is finite, i.e.,

$$\sum_{k=0}^{\infty} \text{tr} \left( (\mathcal{I} - \alpha\mathcal{T})^k \circ \mathbf{A} \right) < \infty. \tag{9}$$

Let $\mathbf{A}_k = (\mathcal{I} - \gamma\mathcal{T})^k \circ \mathbf{A}$. Combining with the definition of $\mathcal{T}$, we obtain

$$\text{tr}(\mathbf{A}_k) = \text{tr}(\mathbf{A}_{k-1}) - 2\alpha \, \text{tr}(\mathbf{H}_{n_1,\beta^{\text{tr}}} \mathbf{A}_{k-1}) + \alpha^2 \, \text{tr}\left(\mathbf{A}\mathbb{E}\left[\mathbf{B}^\top\mathbf{B}\mathbf{B}^\top\mathbf{B}\right]\right).$$

Letting $\mathbf{A} = \mathbf{I}$ in Proposition B.1, we have $\mathbb{E}\left[\mathbf{B}^\top\mathbf{B}\mathbf{B}^\top\mathbf{B}\right] \preceq c(\beta^{\text{tr}}, \boldsymbol{\Sigma}) \text{tr}(\boldsymbol{\Sigma}) \mathbf{H}_{n_1,\beta^{\text{tr}}}$. Hence

$$\text{tr}(\mathbf{A}_k) \leq \text{tr}(\mathbf{A}_{k-1}) - \left(2\alpha - \alpha^2 c(\beta^{\text{tr}}, \boldsymbol{\Sigma}) \text{tr}(\boldsymbol{\Sigma})\right) \text{tr}(\mathbf{H}_{n_1,\beta^{\text{tr}}} \mathbf{A}_{k-1})$$

$$\leq \text{tr}\left((\mathbf{I} - \alpha\mathbf{H}_{n_1,\beta^{\text{tr}}})\mathbf{A}_{k-1}\right) \quad \text{by } \alpha < \frac{1}{c(\beta^{\text{tr}}, \boldsymbol{\Sigma}) \text{tr}(\boldsymbol{\Sigma})}$$

$$\leq \left(1 - \alpha \min_i \{\mu_i(\mathbf{H}_{n_1,\beta^{\text{tr}}})\}\right) \text{tr}(\mathbf{A}_{k-1}).$$

If $\alpha < \frac{1}{\min_i \{\mu_i(\mathbf{H}_{n_1,\beta^{\text{tr}}})\}}$, then we substitute it into eq. (9) and obtain

$$\sum_{k=0}^{\infty} \text{tr}\left((\mathcal{I} - \alpha\mathcal{T})^k \circ \mathbf{A}\right) = \sum_{k=0}^{\infty} \text{tr}(\mathbf{A}_k) \leq \frac{\text{tr}(\mathbf{A})}{\alpha \min_i \{\mu_i(\mathbf{H}_{n_1,\beta^{\text{tr}}})\}} < \infty$$

which guarantees the existence of $\mathcal{T}^{-1}$. Moreover, $\mathbf{A}_k$ is a PSD matrix for every $k$ since $\mathcal{I} - \alpha\mathcal{T}$ is a PSD mapping. The $\mathcal{T}^{-1} \circ \mathbf{A} = \alpha \sum_{k=0}^{\infty} \mathbf{A}_k$ must be a PSD matrix, which implies that $\mathcal{T}^{-1}$ is PSD mapping. $\square$

**Property B.2** (Commutity)**.** *Suppose Assumption 2 holds, then for all $n > 0$, $|\beta| < 1/\lambda_1$, $\mathbf{H}_{n,\beta}$ with different $n$ and $\beta$ commute with each other.*

## B.2 Fourth Moment Upper Bound for Meta Data

In this section, we provide a technical result for the fourth moment of meta data $\mathbf{B}$, which is essential throughout the proof of our upper bound.

**Proposition B.1.** *Suppose Assumptions 1-3 hold. Given $|\beta| < \frac{1}{\lambda_1}$, for any PSD matrix $\mathbf{A}$, we have*

$$\mathbb{E}\left[\mathbf{B}^\top\mathbf{B}\mathbf{A}\mathbf{B}^\top\mathbf{B}\right] \preceq c(\beta^{tr}, \boldsymbol{\Sigma})\mathbb{E}\left[\text{tr}(\mathbf{A}\boldsymbol{\Sigma})\right] \mathbf{H}_{n_1,\beta^{tr}}$$

*where $c(\beta, \boldsymbol{\Sigma}) := c_1 \left(1 + 8|\beta|\lambda_1\sqrt{C(\beta, \boldsymbol{\Sigma})}\sigma_x^2 + 64\sqrt{C(\beta, \boldsymbol{\Sigma})}\sigma_x^4\beta^2 \text{tr}(\boldsymbol{\Sigma}^2)\right).$*

*Proof.* Recall that $\mathbf{B} = \frac{1}{\sqrt{n_2}}\mathbf{X}^{\text{out}}(\mathbf{I} - \frac{\beta}{n_1}\mathbf{X}^{\text{in}\top}\mathbf{X}^{\text{in}})$. With a slight abuse of notations, we write $\beta^{\text{tr}}$ as $\beta$, $\mathbf{X}^{\text{in}}$ as $\mathbf{X}$ in this proof. First consider the case $\beta \geq 0$. By the definition of $\mathbf{B}$, we have

$$\mathbb{E}\left[\mathbf{B}^\top\mathbf{B}\mathbf{A}\mathbf{B}^\top\mathbf{B}\right]$$

$$= \mathbb{E}\left[(\mathbf{I} - \frac{\beta}{n_1}\mathbf{X}^\top\mathbf{X})\frac{1}{n_2}\mathbf{X}^{\text{out}\top}\mathbf{X}^{\text{out}}(\mathbf{I} - \frac{\beta}{n_1}\mathbf{X}^\top\mathbf{X})\mathbf{A}(\mathbf{I} - \frac{\beta}{n_1}\mathbf{X}^\top\mathbf{X})\frac{1}{n_2}\mathbf{X}^{\text{out}\top}\mathbf{X}^{\text{out}}(\mathbf{I} - \frac{\beta}{n_1}\mathbf{X}\mathbf{X})\right]$$

$$\preceq c_1\mathbb{E}\left[\text{tr}\left((\mathbf{I} - \frac{\beta}{n_1}\mathbf{X}^\top\mathbf{X})\mathbf{A}(\mathbf{I} - \frac{\beta}{n_1}\mathbf{X}^\top\mathbf{X})\boldsymbol{\Sigma}\right)(\mathbf{I} - \frac{\beta}{n_1}\mathbf{X}^\top\mathbf{X})\boldsymbol{\Sigma}(\mathbf{I} - \frac{\beta}{n_1}\mathbf{X}^\top\mathbf{X})\right]$$

$$\preceq c_1\mathbb{E}\left[\text{tr}\left(\mathbf{A}(\boldsymbol{\Sigma} + \frac{\beta^2}{n_1^2}\mathbf{X}^\top\mathbf{X}\boldsymbol{\Sigma}\mathbf{X}^\top\mathbf{X})\right)(\mathbf{I} - \frac{\beta}{n_1}\mathbf{X}^\top\mathbf{X})\boldsymbol{\Sigma}(\mathbf{I} - \frac{\beta}{n_1}\mathbf{X}^\top\mathbf{X})\right]$$

where the second inequality follows from Assumption 1. Let $\mathbf{x}_i$ denote the $i$-th row of $\mathbf{X}$. Note that $\mathbf{x}_i = \boldsymbol{\Sigma}^{\frac{1}{2}}\mathbf{z}_i$, where $\mathbf{z}_i$ is independent $\sigma_x$-sub-gaussian vector. For any $\mathbf{x}_{i_1}, \mathbf{x}_{i_2}, \mathbf{x}_{i_3}, \mathbf{x}_{i_4}$, where $1 \leq i_1, i_2, i_3, i_4 \leq n_1$, we have:

$$\mathbb{E}\left[\text{tr}(\mathbf{A}\mathbf{x}_{i_1}\mathbf{x}_{i_2}^\top\boldsymbol{\Sigma}\mathbf{x}_{i_3}\mathbf{x}_{i_4}^\top)(\mathbf{I} - \frac{\beta}{n_1}\mathbf{X}^\top\mathbf{X})\boldsymbol{\Sigma}(\mathbf{I} - \frac{\beta}{n_1}\mathbf{X}^\top\mathbf{X})\right]$$

$$= \mathbb{E}\left[\text{tr}(\boldsymbol{\Sigma}^{\frac{1}{2}}\mathbf{A}\boldsymbol{\Sigma}^{\frac{1}{2}}\mathbf{z}_{i_1}\mathbf{z}_{i_2}^\top\boldsymbol{\Sigma}^2\mathbf{z}_{i_3}\mathbf{z}_{i_4}^\top)(\mathbf{I} - \frac{\beta}{n_1}\mathbf{X}^\top\mathbf{X})\boldsymbol{\Sigma}(\mathbf{I} - \frac{\beta}{n_1}\mathbf{X}^\top\mathbf{X})\right]$$

$$= \sum_{k,j}\mu_k\lambda_j^2\mathbb{E}\left[(\mathbf{z}_{i_4}^\top\mathbf{u}_k)(\mathbf{z}_{i_1}^\top\mathbf{u}_k)(\mathbf{z}_{i_4}^\top\mathbf{v}_j)(\mathbf{z}_{i_1}^\top\mathbf{v}_j)(\mathbf{I} - \frac{\beta}{n_1}\mathbf{X}^\top\mathbf{X})\boldsymbol{\Sigma}(\mathbf{I} - \frac{\beta}{n_1}\mathbf{X}^\top\mathbf{X})\right]$$

where the SVD of $\boldsymbol{\Sigma}^{\frac{1}{2}}\mathbf{A}\boldsymbol{\Sigma}^{\frac{1}{2}}$ is $\sum_j \mu_j\mathbf{u}_j\mathbf{u}_j^\top$, the SVD of $\boldsymbol{\Sigma}$ is $\sum_j \lambda_j\mathbf{v}_j\mathbf{v}_j^\top$. For any unit vector $\mathbf{w} \in \mathbb{R}^d$, we have:

$$\mathbf{w}^\top\mathbb{E}\left[\mathbf{H}_{n_1,\beta}^{-\frac{1}{2}}\text{tr}(\mathbf{A}\mathbf{x}_{i_1}\mathbf{x}_{i_2}^\top\boldsymbol{\Sigma}\mathbf{x}_{i_3}\mathbf{x}_{i_4}^\top)(\mathbf{I} - \frac{\beta}{n}\mathbf{X}^\top\mathbf{X})\boldsymbol{\Sigma}(\mathbf{I} - \frac{\beta}{n}\mathbf{X}^\top\mathbf{X})\mathbf{H}_{n,\beta}^{-\frac{1}{2}}\right]\mathbf{w}$$

$$\leq \sum_{k,j}\mu_k\lambda_j^2\sqrt{\mathbb{E}\left[\left((\mathbf{z}_{i_4}^\top\mathbf{u}_k)(\mathbf{z}_{i_1}^\top\mathbf{u}_k)(\mathbf{z}_{i_4}^\top\mathbf{v}_j)(\mathbf{z}_{i_1}^\top\mathbf{v}_j)^2\right)\right]}$$

$$\times \sqrt{\mathbb{E}\left[\|\mathbf{w}^\top\mathbf{H}_{n_1,\beta}^{-\frac{1}{2}}(\mathbf{I} - \frac{\beta}{n_1}\mathbf{X}^\top\mathbf{X})\boldsymbol{\Sigma}(\mathbf{I} - \frac{\beta}{n_1}\mathbf{X}^\top\mathbf{X})\mathbf{H}_{n_1,\beta}^{-\frac{1}{2}}\mathbf{w}\|^2\right]}$$

$$\leq 64\sqrt{C(\beta,\boldsymbol{\Sigma})}\sigma_x^4\text{tr}(A\boldsymbol{\Sigma})\text{tr}(\boldsymbol{\Sigma}^2)$$

where the first inequality follows from the Cauchy Schwarz inequality; the last inequality is due to Assumption 3 and the property of sub-Gaussian distributions [19]. Therefore,

$$\mathbb{E}\left[\mathbf{H}_{n_1,\beta}^{-\frac{1}{2}}\text{tr}(\mathbf{A}\mathbf{x}_{i_1}\mathbf{x}_{i_2}^\top\boldsymbol{\Sigma}\mathbf{x}_{i_3}\mathbf{x}_{i_4}^\top)(\mathbf{I} - \frac{\beta}{n_1}\mathbf{X}^\top\mathbf{X})\boldsymbol{\Sigma}(\mathbf{I} - \frac{\beta}{n_1}\mathbf{X}^\top\mathbf{X})\mathbf{H}_{n_1,\beta}^{-\frac{1}{2}}\right]$$

$$\preceq 64\sqrt{C(\beta,\boldsymbol{\Sigma})}\sigma_x^4\text{tr}(A\boldsymbol{\Sigma}^2)\mathbf{I}$$

which implies

$$\mathbb{E}\left[\text{tr}(\mathbf{A}\mathbf{x}_{i_1}\mathbf{x}_{i_2}^\top\boldsymbol{\Sigma}\mathbf{x}_{i_3}\mathbf{x}_{i_4}^\top)(\mathbf{I} - \frac{\beta}{n_1}\mathbf{X}^\top\mathbf{X})\boldsymbol{\Sigma}(\mathbf{I} - \frac{\beta}{n_1}\mathbf{X}^\top\mathbf{X})\right] \preceq 64\sqrt{C(\beta,\boldsymbol{\Sigma})}\sigma_x^4\text{tr}(A\boldsymbol{\Sigma}^2)\mathbf{H}_{n_1,\beta}.$$

Hence,

$$\mathbb{E}\left[\mathbf{B}^\top\mathbf{B}\mathbf{A}\mathbf{B}^\top\mathbf{B}\right]$$

$$\preceq c_1\mathbb{E}\left[\text{tr}\left(\mathbf{A}(\boldsymbol{\Sigma} + 64\sqrt{C}\sigma_x^4\beta^2\boldsymbol{\Sigma}\text{tr}(\boldsymbol{\Sigma}^2))\right)(\mathbf{I} - \frac{\beta}{n}\mathbf{X}^\top\mathbf{X})\boldsymbol{\Sigma}(\mathbf{I} - \frac{\beta}{n}\mathbf{X}^\top\mathbf{X})\right]$$

$$\preceq c_1(1 + 64\sqrt{C(\beta,\boldsymbol{\Sigma})}\sigma_x^4\beta^2\text{tr}(\boldsymbol{\Sigma}^2))\mathbb{E}\left[\text{tr}(\mathbf{A}\boldsymbol{\Sigma})\right]\mathbf{H}_{n_1,\beta}.$$

Now we turn to $\beta < 0$, and derive

$$\mathbb{E}\left[\mathbf{B}^\top \mathbf{B} \mathbf{A} \mathbf{B}^\top \mathbf{B}\right]$$

$$\preceq c_1 \mathbb{E}\left[\mathrm{tr}\left((\mathbf{I} - \frac{\beta}{n_1}\mathbf{X}^\top\mathbf{X})\mathbf{A}(\mathbf{I} - \frac{\beta}{n_1}\mathbf{X}^\top\mathbf{X})\mathbf{\Sigma}\right)(\mathbf{I} - \frac{\beta}{n_1}\mathbf{X}^\top\mathbf{X})\mathbf{\Sigma}(\mathbf{I} - \frac{\beta}{n_1}\mathbf{X}^\top\mathbf{X})\right]$$

$$= c_1 \mathbb{E}\left[\mathrm{tr}\left(\mathbf{A}(\underbrace{\mathbf{\Sigma} - \frac{\beta}{n_1}(\mathbf{X}^\top\mathbf{X}\mathbf{\Sigma} + \mathbf{\Sigma}\mathbf{X}^\top\mathbf{X})}_{\mathbf{J}_1} + \frac{\beta^2}{n_1^2}\mathbf{X}^\top\mathbf{X}\mathbf{\Sigma}\mathbf{X}^\top\mathbf{X})\right) \cdot (\mathbf{I} - \frac{\beta}{n_1}\mathbf{X}^\top\mathbf{X})\mathbf{\Sigma}(\mathbf{I} - \frac{\beta}{n_1}\mathbf{X}^\top\mathbf{X})\right].$$

We can bound the extra term $\mathbf{J}_1$ in the similar way as $\beta > 0$. For any $\mathbf{x}_i$, $1 \le i \le n_1$, we have

$$\mathbb{E}\left[\mathrm{tr}\left(\mathbf{A}\mathbf{x}_i\mathbf{x}_i^\top\mathbf{\Sigma}\right)(\mathbf{I} - \frac{\beta}{n_1}\mathbf{X}^\top\mathbf{X})\mathbf{\Sigma}(\mathbf{I} - \frac{\beta}{n_1}\mathbf{X}^\top\mathbf{X})\right]$$

$$= \mathbb{E}\left[\mathrm{tr}\left(\mathbf{z}_i^\top\mathbf{\Sigma}^{\frac{3}{2}}\mathbf{A}\mathbf{\Sigma}^{\frac{1}{2}}\mathbf{z}_i\right)(\mathbf{I} - \frac{\beta}{n_1}\mathbf{X}^\top\mathbf{X})\mathbf{\Sigma}(\mathbf{I} - \frac{\beta}{n_1}\mathbf{X}^\top\mathbf{X})\right]$$

$$= \sum_k \iota_k \mathbb{E}\left[(\mathbf{z}_i^\top\boldsymbol{\kappa}_k)^2(\mathbf{I} - \frac{\beta}{n_1}\mathbf{X}^\top\mathbf{X})\mathbf{\Sigma}(\mathbf{I} - \frac{\beta}{n_1}\mathbf{X}^\top\mathbf{X})\right]$$

where the SVD of $\mathbf{\Sigma}^{\frac{3}{2}}\mathbf{A}\mathbf{\Sigma}^{\frac{1}{2}}$ is $\sum_k \iota_k \boldsymbol{\kappa}_k\boldsymbol{\kappa}_k^\top$. Similarly, for any unit vector $\mathbf{w} \in \mathbb{R}^d$, we can obtain

$$\mathbf{w}^\top \mathbb{E}\left[\mathbf{H}_{n_1,\beta}^{-\frac{1}{2}}\mathrm{tr}\left(\mathbf{A}\mathbf{x}_i\mathbf{x}_i^\top\mathbf{\Sigma}\right)(\mathbf{I} - \frac{\beta}{n_1}\mathbf{X}^\top\mathbf{X})\mathbf{\Sigma}(\mathbf{I} - \frac{\beta}{n_1}\mathbf{X}^\top\mathbf{X})\mathbf{H}_{n_1,\beta}^{-\frac{1}{2}}\right]\mathbf{w}$$

$$\le \sum_k \iota_k \sqrt{\mathbb{E}[(\mathbf{z}_i^\top\boldsymbol{\kappa}_k)^4]}\sqrt{\mathbb{E}[\|\mathbf{w}^\top\mathbf{H}_{n_1,\beta}^{-\frac{1}{2}}(\mathbf{I} - \frac{\beta}{n_1}\mathbf{X}^\top\mathbf{X})\mathbf{\Sigma}(\mathbf{I} - \frac{\beta}{n_1}\mathbf{X}^\top\mathbf{X})\mathbf{H}_{n_1,\beta}^{-\frac{1}{2}}\mathbf{w}\|^2]}$$

$$\le 4\sqrt{C(\beta, \mathbf{\Sigma})}\sigma_x^2 \mathrm{tr}(A\mathbf{\Sigma}^2)$$

which implies:

$$\mathbb{E}\left[\mathrm{tr}\left(\mathbf{A}\mathbf{x}_i\mathbf{x}_i^\top\mathbf{\Sigma}\right)(\mathbf{I} - \frac{\beta}{n_1}\mathbf{X}^\top\mathbf{X})\mathbf{\Sigma}(\mathbf{I} - \frac{\beta}{n_1}\mathbf{X}^\top\mathbf{X})\right] \preceq 4\sqrt{C(\beta, \mathbf{\Sigma})}\sigma_x^2 \mathrm{tr}(\mathbf{A}\mathbf{\Sigma}^2)\mathbf{H}_{n_1,\beta}.$$

Hence,

$$\mathbb{E}\left[\mathbf{B}^\top\mathbf{B}\mathbf{A}\mathbf{B}^\top\mathbf{B}\right]$$

$$\preceq c_1\mathbb{E}\left[\mathrm{tr}\left(\mathbf{A}(\mathbf{\Sigma} - 8\beta\sqrt{C}\sigma_x^2\mathbf{\Sigma}^2 + 64\sqrt{C}\sigma_x^4\beta^2\mathbf{\Sigma}\,\mathrm{tr}(\mathbf{\Sigma}^2))\right)(\mathbf{I} - \frac{\beta}{n_1}\mathbf{X}^\top\mathbf{X})\mathbf{\Sigma}(\mathbf{I} - \frac{\beta}{n_1}\mathbf{X}^\top\mathbf{X})\right]$$

$$\preceq c_1\left(1 - 8\beta\lambda_1\sqrt{C(\beta, \mathbf{\Sigma})}\sigma_x^2 + 64\sqrt{C(\beta, \mathbf{\Sigma})}\sigma_x^4\beta^2\,\mathrm{tr}(\mathbf{\Sigma}^2)\right)\mathbb{E}\left[\mathrm{tr}(\mathbf{A}\mathbf{\Sigma})\right]\mathbf{H}_{n_1,\beta}.$$

Together with the discussions for $\beta > 0$, we have

$$c(\beta, \mathbf{\Sigma}) = c_1(1 + 8|\beta|\lambda_1\sqrt{C(\beta, \mathbf{\Sigma})}\sigma_x^2 + 64\sqrt{C(\beta, \mathbf{\Sigma})}\sigma_x^4\beta^2\,\mathrm{tr}(\mathbf{\Sigma}^2)),$$

which completes the proof. $\qquad\square$

## B.3 Bias-Variance Decomposition

We will use the bias-variance decomposition similar to theoretical studies of classic linear regression [14, 8, 22]. Consider the error at each iteration: $\boldsymbol{\varrho}_t = \boldsymbol{\omega}_t - \boldsymbol{\theta}^*$, where $\boldsymbol{\omega}_t$ is the SGD output at each iteration $t$. Then the update rule can be written as:

$$\boldsymbol{\varrho}_t := (\mathbf{I} - \alpha\mathbf{B}_t^\top\mathbf{B}_t)\boldsymbol{\varrho}_{t-1} + \alpha\mathbf{B}_t^\top\boldsymbol{\xi}_t$$

where $\mathbf{B}_t, \boldsymbol{\xi}_t$ are the meta data and noise at iteration $t$ (see eqs. (2) and (6)). It is helpful to consider $\boldsymbol{\varrho}_t$ as the sum of the following two random processes:

- If there is no meta noise, the error comes from the bias:

$$\boldsymbol{\varrho}_t^{\text{bias}} := (\mathbf{I} - \alpha \mathbf{B}_t^\top \mathbf{B}_t)\boldsymbol{\varrho}_{t-1}^{\text{bias}} \quad \boldsymbol{\varrho}_t^{\text{bias}} = \boldsymbol{\varrho}_0.$$

- If the SGD trajectory starts from $\boldsymbol{\theta}^*$, the error originates from the variance:

$$\boldsymbol{\varrho}_t^{\text{var}} := (\mathbf{I} - \alpha \mathbf{B}_t^\top \mathbf{B}_t)\boldsymbol{\varrho}_{t-1}^{\text{var}} + \alpha \mathbf{B}_t^\top \boldsymbol{\xi}_t \quad \boldsymbol{\varrho}^{\text{var}} = \mathbf{0}$$

and $\mathbb{E}[\boldsymbol{\varrho}_t^{\text{var}}] = 0$.

With slightly abused notations, we have:

$$\boldsymbol{\varrho}_t = \boldsymbol{\varrho}_t^{\text{bias}} + \boldsymbol{\varrho}_t^{\text{var}}.$$

Define the averaged output of $\boldsymbol{\varrho}_t^{\text{bias}}$, $\boldsymbol{\varrho}_t^{\text{var}}$ and $\boldsymbol{\varrho}_t$ after $T$ iterations as:

$$\overline{\boldsymbol{\varrho}}_T^{\text{bias}} = \frac{1}{T}\sum_{t=1}^{T} \boldsymbol{\varrho}_t^{\text{bias}}, \quad \overline{\boldsymbol{\varrho}}_T^{\text{var}} = \frac{1}{T}\sum_{t=1}^{T} \boldsymbol{\varrho}_t^{\text{var}}, \quad \overline{\boldsymbol{\varrho}}_T = \frac{1}{T}\sum_{t=1}^{T} \boldsymbol{\varrho}_t. \tag{10}$$

Similarly, we have

$$\overline{\boldsymbol{\varrho}}_T = \overline{\boldsymbol{\varrho}}_T^{\text{bias}} + \overline{\boldsymbol{\varrho}}_T^{\text{var}}.$$

Now we are ready to introduce the bias-variance decomposition for the excess risk.

**Lemma B.2** (Bias-variance decomposition). *Following the notations in eq. (10), then the excess risk can be decomposed as*

$$R(\overline{\boldsymbol{\omega}}_T, \beta^{te}) \leq 2\mathcal{E}_{bias} + 2\mathcal{E}_{var}$$

*where*

$$\mathcal{E}_{bias} = \frac{1}{2}\langle \mathbf{H}_{m,\beta^{te}}, \mathbb{E}[\overline{\boldsymbol{\varrho}}_T^{bias} \otimes \overline{\boldsymbol{\varrho}}_T^{bias}]\rangle, \quad \mathcal{E}_{var} = \frac{1}{2}\langle \mathbf{H}_{m,\beta^{te}}, \mathbb{E}[\overline{\boldsymbol{\varrho}}_T^{var} \otimes \overline{\boldsymbol{\varrho}}_T^{var}]\rangle. \tag{11}$$

*Proof.* By Lemma A.1, we have

$$\begin{aligned}
R(\overline{\boldsymbol{\omega}}_T, \beta^{te}) &= \frac{1}{2}\langle \mathbf{H}_{m,\beta^{te}}, \mathbb{E}[\overline{\boldsymbol{\varrho}}_T \otimes \overline{\boldsymbol{\varrho}}_T]\rangle \\
&= \frac{1}{2}\langle \mathbf{H}_{m,\beta^{te}}, \mathbb{E}[(\overline{\boldsymbol{\varrho}}_T^{\text{bias}} + \overline{\boldsymbol{\varrho}}_T^{\text{var}}) \otimes (\overline{\boldsymbol{\varrho}}_T^{\text{bias}} + \overline{\boldsymbol{\varrho}}_T^{\text{var}})]\rangle \\
&\leq 2\left(\frac{1}{2}\langle \mathbf{H}_{m,\beta^{te}}, \mathbb{E}[\overline{\boldsymbol{\varrho}}_T^{\text{bias}} \otimes \overline{\boldsymbol{\varrho}}_T^{\text{bias}}]\rangle + \frac{1}{2}\langle \mathbf{H}_{m,\beta^{te}}, \mathbb{E}[\overline{\boldsymbol{\varrho}}_T^{\text{var}} \otimes \overline{\boldsymbol{\varrho}}_T^{\text{var}}]\rangle\right)
\end{aligned}$$

where the last inequality follows because for vector-valued random variables $\mathbf{u}$ and $\mathbf{v}$, $\mathbb{E}\|\mathbf{u} + \mathbf{v}\|_H^2 \leq \left(\sqrt{\mathbb{E}\|\mathbf{u}\|_H^2} + \sqrt{\mathbb{E}\|\mathbf{v}\|_H^2}\right)^2$ and from Cauchy-Schwarz inequality. $\square$

For $t = 0, 1, \cdots, T-1$, consider the following bias and variance iterates:

$$\begin{aligned}
\mathbf{D}_t &= (\mathcal{I} - \alpha\mathcal{T}) \circ \mathbf{D}_{t-1} \quad \text{and} \quad \mathbf{D}_0 = (\boldsymbol{\omega}_t - \boldsymbol{\theta}^*)(\boldsymbol{\omega}_t - \boldsymbol{\theta}^*)^\top \\
\mathbf{V}_t &= (\mathcal{I} - \alpha\mathcal{T}) \circ \mathbf{V}_{t-1} + \alpha^2\Pi \quad \text{and} \quad \mathbf{V}_0 = \mathbf{0}
\end{aligned} \tag{12}$$

where $\Pi = \mathbb{E}[\mathbf{B}^\top \boldsymbol{\xi}\boldsymbol{\xi}^\top \mathbf{B}]$. One can verify that

$$\mathbf{D}_t = \mathbb{E}\left[\boldsymbol{\varrho}_t^{\text{bias}} \otimes \boldsymbol{\varrho}_t^{\text{bias}}\right], \quad \mathbf{V}_t = \mathbb{E}\left[\boldsymbol{\varrho}_t^{\text{var}} \otimes \boldsymbol{\varrho}_t^{\text{var}}\right].$$

With such notations, we can further bound the bias and variance terms.

**Lemma B.3.** *Following the notations in eq. (12), we have*

$$\mathcal{E}_{bias} \leq \frac{1}{\alpha T^2}\left\langle \left(\mathbf{I} - (\mathbf{I} - \alpha\mathbf{H}_{n_1,\beta^{tr}})^T\right)\mathbf{H}_{n_1,\beta^{tr}}^{-1}\mathbf{H}_{m,\beta^{te}}, \sum_{t=0}^{T-1}\mathbf{D}_t\right\rangle, \tag{13}$$

$$\mathcal{E}_{var} \leq \frac{1}{T^2}\sum_{t=0}^{T-1}\sum_{k=t}^{T-1}\left\langle(\mathbf{I} - \alpha\mathbf{H}_{n_1,\beta^{tr}})^{k-t}\mathbf{H}_{m,\beta^{te}}, \mathbf{V}_t\right\rangle. \tag{14}$$

*Proof.* Similar calculations have appeared in the prior works [14, 22]. However, our meta linear model contains additional terms, and hence we provide a proof here for completeness. We first have

$$
\mathbb{E}[\overline{\boldsymbol{\varrho}}_T^{\mathrm{var}} \otimes \overline{\boldsymbol{\varrho}}_T^{\mathrm{var}}] = \frac{1}{T^2} \sum_{t=0}^{T-1} \sum_{k=0}^{T-1} \mathbb{E}[\boldsymbol{\varrho}_t^{\mathrm{var}} \otimes \boldsymbol{\varrho}_k^{\mathrm{var}}]
$$

$$
\preceq \frac{1}{T^2} \sum_{t=0}^{T-1} \sum_{k=t}^{T-1} \mathbb{E}[\boldsymbol{\varrho}_t^{\mathrm{var}} \otimes \boldsymbol{\varrho}_k^{\mathrm{var}}] + \mathbb{E}[\boldsymbol{\varrho}_k^{\mathrm{var}} \otimes \boldsymbol{\varrho}_t^{\mathrm{var}}]
$$

where the last inequality follows because we double count the diagonal terms $t = k$.

For $t \le k$, $\mathbb{E}[\boldsymbol{\varrho}_k^{\mathrm{var}} | \boldsymbol{\varrho}_t^{\mathrm{var}}] = (\mathbf{I} - \alpha \mathbf{H}_{n_1, \beta^{\mathrm{tr}}})^{k-t} \boldsymbol{\varrho}_t^{\mathrm{var}}$, since $\mathbb{E}[\mathbf{B}_t^\top \boldsymbol{\xi}_t | \boldsymbol{\varrho}_{t-1}] = \mathbf{0}$. From this, we have

$$
\mathbb{E}[\overline{\boldsymbol{\varrho}}_T^{\mathrm{var}} \otimes \overline{\boldsymbol{\varrho}}_T^{\mathrm{var}}] \preceq \frac{1}{T^2} \sum_{t=0}^{T-1} \sum_{k=t}^{T-1} \mathbf{V}_t (\mathbf{I} - \alpha \mathbf{H}_{n_1, \beta^{\mathrm{tr}}})^{k-t} + \mathbf{V}_t (\mathbf{I} - \alpha \mathbf{H}_{n_1, \beta^{\mathrm{tr}}})^{k-t}.
$$

Substituting the above inequality into $\frac{1}{2} \langle \mathbf{H}_{m, \beta^{\mathrm{te}}}, \mathbb{E}[\overline{\boldsymbol{\varrho}}_T^{\mathrm{var}} \otimes \overline{\boldsymbol{\varrho}}_T^{\mathrm{var}}] \rangle$, we obtain:

$$
\mathcal{E}_{\mathrm{var}} = \frac{1}{2} \langle \mathbf{H}_{m, \beta^{\mathrm{te}}}, \mathbb{E}[\overline{\boldsymbol{\varrho}}_T^{\mathrm{var}} \otimes \overline{\boldsymbol{\varrho}}_T^{\mathrm{var}}] \rangle
$$

$$
\le \frac{1}{2T^2} \sum_{t=0}^{T-1} \sum_{k=t}^{T-1} \langle \mathbf{H}_{m, \beta^{\mathrm{te}}}, \mathbf{V}_t (\mathbf{I} - \alpha \mathbf{H}_{n_1, \beta^{\mathrm{tr}}})^{k-t} \rangle + \langle \mathbf{H}_{m, \beta^{\mathrm{te}}}, \mathbf{V}_t (\mathbf{I} - \alpha \mathbf{H}_{n_1, \beta^{\mathrm{tr}}})^{k-t} \rangle
$$

$$
= \frac{1}{T^2} \sum_{t=0}^{T-1} \sum_{k=t}^{T-1} \langle (\mathbf{I} - \alpha \mathbf{H}_{n_1, \beta^{\mathrm{tr}}})^{k-t} \mathbf{H}_{m, \beta^{\mathrm{te}}}, \mathbf{V}_t \rangle
$$

where the last inequality follows from Assumption 2 that $F$ and $\boldsymbol{\Sigma}$ commute, and hence $\mathbf{H}_{m, \beta^{\mathrm{te}}}$ and $\mathbf{I} - \alpha \mathbf{H}_{n_1, \beta^{\mathrm{tr}}}$ commute.

For the bias term, similarly we have:

$$
\mathcal{E}_{\mathrm{bias}} \le \frac{1}{T^2} \sum_{t=0}^{T-1} \sum_{k=t}^{T-1} \langle (\mathbf{I} - \alpha \mathbf{H}_{n_1, \beta^{\mathrm{tr}}})^{k-t} \mathbf{H}_{m, \beta^{\mathrm{te}}}, \mathbf{D}_t \rangle \tag{15}
$$

$$
= \frac{1}{\alpha T^2} \sum_{t=0}^{T-1} \langle \left( \mathbf{I} - (\mathbf{I} - \alpha \mathbf{H}_{n_1, \beta^{\mathrm{tr}}})^{T-t} \right) \mathbf{H}_{n_1, \beta^{\mathrm{tr}}}^{-1} \mathbf{H}_{m, \beta^{\mathrm{te}}}, \mathbf{D}_t \rangle \tag{16}
$$

$$
\le \frac{1}{\alpha T^2} \langle \left( \mathbf{I} - (\mathbf{I} - \alpha \mathbf{H}_{n_1, \beta^{\mathrm{tr}}})^T \right) \mathbf{H}_{n_1, \beta^{\mathrm{tr}}}^{-1} \mathbf{H}_{m, \beta^{\mathrm{te}}}, \sum_{t=0}^{T-1} \mathbf{D}_t \rangle \tag{17}
$$

which completes the proof. $\square$

## B.4 Bounding the Bias

Now we start to bound the bias term. By Lemma B.3, we focus on bounding the summation of $\mathbf{D}_t$, i.e. $\sum_{t=0}^{T-1} \mathbf{D}_t$. Consider $\mathbf{S}_t := \sum_{k=0}^{t-1} \mathbf{D}_k$, and the following lemma shows the properties of $\mathbf{S}_t$

**Lemma B.4.** $\mathbf{S}_t$ *satisfies the recursion form:*

$$
\mathbf{S}_t = (\mathcal{I} - \alpha \mathcal{T}) \circ \mathbf{S}_{t-1} + \mathbf{D}_0.
$$

*Moreover, if* $\alpha < \frac{1}{c(\beta^{\mathrm{tr}}, \boldsymbol{\Sigma}) \operatorname{tr}(\boldsymbol{\Sigma})}$, *then we have:*

$$
\mathbf{D}_0 = \mathbf{S}_0 \preceq \mathbf{S}_1 \preceq \cdots \preceq \mathbf{S}_\infty
$$

*where* $\mathbf{S}_\infty := \sum_{k=0}^{\infty} (\mathcal{I} - \alpha \mathcal{T})^k \circ \mathbf{D}_0 = \alpha^{-1} \mathcal{T}^{-1} \circ \mathbf{D}_0.$

*Proof.* By eq. ([12](#)), we have

$$\mathbf{S}_t = \sum_{k=0}^{t-1} \mathbf{D}_k = \sum_{k=0}^{t-1} (\mathcal{I} - \alpha\mathcal{T})^k \circ \mathbf{D}_0$$

$$= \mathbf{D}_0 + (\mathcal{I} - \alpha\mathcal{T}) \circ \left( \sum_{k=0}^{t-2} (\mathcal{I} - \alpha\mathcal{T})^k \circ \mathbf{D}_0 \right)$$

$$= \mathbf{D}_0 + (\mathcal{I} - \alpha\mathcal{T}) \circ \mathbf{S}_{t-1}.$$

By Lemma [B.1](#), $(\mathcal{I} - \alpha\mathcal{T})$ is PSD mapping, and hence $\mathbf{D}_t = (\mathcal{I} - \alpha\mathcal{T}) \circ \mathbf{D}_{t-1}$ is a PSD matirx for every $t$, which implies $\mathbf{S}_{t-1} \preceq \mathbf{S}_{t-1} + \mathbf{D}_t = \mathbf{S}_t$. The form of $\mathbf{S}_\infty$ can be directly obtained by Lemma [B.1](#). $\square$

Then we can decompose $\mathbf{S}_t$ as follows:

$$\mathbf{S}_t = \mathbf{D}_0 + (\mathcal{I} - \alpha\widetilde{\mathcal{T}}) \circ \mathbf{S}_{t-1} + \alpha(\widetilde{\mathcal{T}} - \mathcal{T}) \circ \mathbf{S}_{t-1}$$

$$= \mathbf{D}_0 + (\mathcal{I} - \alpha\widetilde{\mathcal{T}}) \circ \mathbf{S}_{t-1} + \alpha^2(\mathcal{M} - \widetilde{\mathcal{M}}) \circ \mathbf{S}_{t-1}$$

$$\preceq \mathbf{D}_0 + (\mathcal{I} - \alpha\widetilde{\mathcal{T}}) \circ \mathbf{S}_{t-1} + \alpha^2\mathcal{M} \circ \mathbf{S}_T$$

$$= \sum_{k=0}^{t-1} (\mathcal{I} - \alpha\widetilde{\mathcal{T}})^k \circ (\mathbf{D}_0 + \alpha^2\mathcal{M} \circ \mathbf{S}_T) \tag{18}$$

where the inequality follows because $\mathbf{S}_t \preceq \mathbf{S}_T$ for any $t \leq T$. Therefore, it is crucial to understand $\mathcal{M} \circ \mathbf{S}_T$.

**Lemma B.5.** *For any symmetric matrix* $\mathbf{A}$, *if* $\alpha < \frac{1}{c(\beta^{tr}, \mathbf{\Sigma}) \operatorname{tr}(\mathbf{\Sigma})}$, *it holds that*

$$\mathcal{M} \circ \mathcal{T}^{-1} \circ \mathbf{A} \preceq \frac{c(\beta^{tr}, \mathbf{\Sigma}) \operatorname{tr}\left(\mathbf{\Sigma}\mathbf{H}_{n_1,\beta^{tr}}^{-1}\mathbf{A}\right)}{1 - \alpha c(\beta^{tr}, \mathbf{\Sigma}) \operatorname{tr}(\mathbf{\Sigma})} \cdot \mathbf{H}_{n_1,\beta^{tr}}.$$

*Proof.* Denote $\mathbf{C} = \mathcal{T}^{-1} \circ \mathbf{A}$. Recalling $\widetilde{\mathcal{T}} = \mathcal{T} + \alpha\mathcal{M} - \alpha\widetilde{\mathcal{M}}$, we have

$$\widetilde{\mathcal{T}} \circ \mathbf{C} = \mathcal{T} \circ \mathbf{C} + \alpha\mathcal{M} \circ \mathbf{C} - \alpha\widetilde{\mathcal{M}} \circ \mathbf{C}$$

$$\preceq \mathbf{A} + \alpha\mathcal{M} \circ \mathbf{C}.$$

Recalling that $\widetilde{\mathcal{T}}^{-1}$ exists and is a PSD mapping, we then have

$$\mathcal{M} \circ \mathbf{C} \preceq \alpha\mathcal{M} \circ \widetilde{\mathcal{T}}^{-1} \circ \mathcal{M} \circ \mathbf{C} + \mathcal{M} \circ \widetilde{\mathcal{T}}^{-1} \circ \mathbf{A}$$

$$\preceq \sum_{k=0}^{\infty} (\alpha\mathcal{M} \circ \widetilde{\mathcal{T}}^{-1})^k \circ (\mathcal{M} \circ \widetilde{\mathcal{T}}^{-1} \circ \mathbf{A}). \tag{19}$$

By Proposition [B.1](#), we have $\mathcal{M} \circ \widetilde{\mathcal{T}}^{-1} \circ \mathbf{A} \preceq \underbrace{c(\beta^{\text{tr}}, \mathbf{\Sigma}) \operatorname{tr}(\mathbf{\Sigma}\widetilde{\mathcal{T}}^{-1} \circ \mathbf{A})}_{J_2} \mathbf{H}_{n_1,\beta^{\text{tr}}}$. Substituting back into eq. ([19](#)), we obtain:

$$\sum_{k=0}^{\infty} (\alpha\mathcal{M} \circ \widetilde{\mathcal{T}}^{-1})^k \circ (\mathcal{M} \circ \widetilde{\mathcal{T}}^{-1} \circ \mathbf{A}) \preceq \sum_{k=0}^{\infty} (\alpha\mathcal{M} \circ \widetilde{\mathcal{T}}^{-1})^k \circ (J_2 \mathbf{H}_{n_1,\beta^{\text{tr}}})$$

$$\preceq J_2 \sum_{k=0}^{\infty} (\alpha c(\beta^{\text{tr}}, \mathbf{\Sigma}) \operatorname{tr}(\mathbf{\Sigma}))^k \mathbf{H}_{n_1,\beta^{\text{tr}}} \preceq \frac{J_2}{1 - \alpha c(\beta^{\text{tr}}, \mathbf{\Sigma}) \operatorname{tr}(\mathbf{\Sigma})} \mathbf{H}_{n_1,\beta^{\text{tr}}}$$

where the second inequality follows since $\widetilde{\mathcal{T}}^{-1} \circ \mathbf{H}_{n_1,\beta^{\text{tr}}} \preceq \mathbf{I}$ (Lemma [B.1](#)) and $\mathcal{M} \circ \mathbf{I} \preceq c(\beta^{\text{tr}}, \mathbf{\Sigma}) \operatorname{tr}(\mathbf{\Sigma}) \mathbf{H}_{n_1,\beta^{\text{tr}}}$ (Proposition [B.1](#)).

Finally, we bound $J_2$ as follows:

$$\operatorname{tr}\left(\boldsymbol{\Sigma}\widetilde{\mathcal{T}}^{-1}\circ\mathbf{A}\right)=\alpha\operatorname{tr}\left(\sum_{k=0}^{\infty}\boldsymbol{\Sigma}(\mathbf{I}-\alpha\mathbf{H}_{n_1,\beta^{\mathrm{tr}}})^k\mathbf{A}(\mathbf{I}-\alpha\mathbf{H}_{n_1,\beta^{\mathrm{tr}}})^k\right)$$

$$=\alpha\operatorname{tr}\left(\sum_{k=0}^{\infty}\boldsymbol{\Sigma}(\mathbf{I}-\alpha\mathbf{H}_{n_1,\beta^{\mathrm{tr}}})^{2k}\mathbf{A}\right)$$

$$=\operatorname{tr}\left(\boldsymbol{\Sigma}\left(2\mathbf{H}_{n_1,\beta^{\mathrm{tr}}}-\alpha\mathbf{H}_{n_1,\beta^{\mathrm{tr}}}^2\right)^{-1}\mathbf{A}\right)$$

$$\leq\operatorname{tr}\left(\boldsymbol{\Sigma}\mathbf{H}_{n_1,\beta^{\mathrm{tr}}}^{-1}\mathbf{A}\right)$$

where the second equality follows because $\boldsymbol{\Sigma}$ and $\mathbf{H}_{n_1,\beta^{\mathrm{tr}}}$ commute, and the last inequality holds since $\alpha<\frac{1}{\max_i\{\mu_i(\mathbf{H}_{n_1,\beta^{\mathrm{tr}}})\}}$. Putting all these results together completes the proof. $\qquad\square$

**Lemma B.6** (Bounding $\mathcal{M}\circ\mathbf{S}_T$).

$$\mathcal{M}\circ\mathbf{S}_T\preceq\frac{c(\beta^{tr},\boldsymbol{\Sigma})\cdot\operatorname{tr}\left(\boldsymbol{\Sigma}\mathbf{H}_{n_1,\beta^{tr}}^{-1}\left[\mathcal{I}-(\mathcal{I}-\alpha\widetilde{\mathcal{T}})^T\right]\circ\mathbf{D}_0\right)}{\alpha(1-c(\beta^{tr},\boldsymbol{\Sigma})\alpha\operatorname{tr}(\boldsymbol{\Sigma}))}\cdot\mathbf{H}_{n_1,\beta^{tr}}.$$

*Proof.* $\mathbf{S}_T$ can be further derived as follows:

$$\mathbf{S}_T=\sum_{k=0}^{T-1}(\mathcal{I}-\alpha\mathcal{T})^k\circ\mathbf{D}_0=\alpha^{-1}\mathcal{T}^{-1}\circ\left[\mathcal{I}-(\mathcal{I}-\alpha\mathcal{T})^T\right]\circ\mathbf{D}_0.$$

Since $\widetilde{\mathcal{T}}-\mathcal{T}$ is a PSD mapping by Lemma B.1, we have $\mathcal{I}-\alpha\widetilde{\mathcal{T}}\leq\mathcal{I}-\alpha\mathcal{T}$. Hence $\mathcal{I}-(\mathcal{I}-\alpha\mathcal{T})^T\preceq\mathcal{I}-(\mathcal{I}-\alpha\widetilde{\mathcal{T}})^T$. Combining with the fact that $\mathcal{T}^{-1}$ is also a PSD mapping, we have:

$$\mathbf{S}_T\preceq\alpha^{-1}\mathcal{T}^{-1}\circ\left[\mathcal{I}-(\mathcal{I}-\alpha\widetilde{\mathcal{T}})^T\right]\circ\mathbf{D}_0.$$

Letting $\mathbf{A}=\left[\mathcal{I}-(\mathcal{I}-\alpha\widetilde{\mathcal{T}})^T\right]\circ\mathbf{D}_0$ in Lemma B.5, we obtain:

$$\mathcal{M}\circ\mathbf{S}_T\preceq\alpha^{-1}\mathcal{M}\circ\mathcal{T}^{-1}\circ\left[\mathcal{I}-(\mathcal{I}-\alpha\widetilde{\mathcal{T}})^T\right]\circ\mathbf{D}_0$$

$$\preceq\frac{c(\beta^{\mathrm{tr}},\boldsymbol{\Sigma})\cdot\operatorname{tr}\left(\boldsymbol{\Sigma}\mathbf{H}_{n_1,\beta^{\mathrm{tr}}}^{-1}\left[\mathcal{I}-(\mathcal{I}-\alpha\widetilde{\mathcal{T}})^T\right]\circ\mathbf{D}_0\right)}{\alpha(1-c(\beta^{\mathrm{tr}},\boldsymbol{\Sigma})\alpha\operatorname{tr}(\boldsymbol{\Sigma}))}\cdot\mathbf{H}_{n_1,\beta^{\mathrm{tr}}}.$$

$$\square$$

Now we are ready to derive the upper bound on the bias term.

**Lemma B.7** (Bounding the bias). *If $\alpha<\frac{1}{c(\beta^{tr},\boldsymbol{\Sigma})\operatorname{tr}(\boldsymbol{\Sigma})}$, for sufficiently large $n_1$, s.t. $\mu_i(\mathbf{H}_{n_1,\beta^{tr}})>0$, $\forall i$, then we have*

$$\mathcal{E}_{bias}\leq\sum_i\left(\frac{1}{\alpha^2T^2}\mathbf{1}_{\mu_i(\mathbf{H}_{n_1,\beta^{tr}})\geq\frac{1}{\alpha T}}+\mu_i^2(\mathbf{H}_{n_1,\beta^{tr}})\mathbf{1}_{\mu_i(\mathbf{H}_{n_1,\beta^{tr}})<\frac{1}{\alpha T}}\right)\frac{\omega_i^2\mu_i(\mathbf{H}_{m,\beta^{te}})}{\mu_i(\mathbf{H}_{n_1,\beta^{tr}})^2}$$

$$+\frac{2c(\beta^{tr},\boldsymbol{\Sigma})}{T\alpha\left(1-c(\beta^{tr},\boldsymbol{\Sigma})\alpha\operatorname{tr}(\boldsymbol{\Sigma})\right)}\sum_i\left(\frac{1}{\mu_i(\mathbf{H}_{n_1,\beta^{tr}})}\mathbf{1}_{\mu_i(\mathbf{H}_{n_1,\beta^{tr}})\geq\frac{1}{\alpha T}}+T\alpha\mathbf{1}_{\mu_i(\mathbf{H}_{n_1,\beta^{tr}})<\frac{1}{\alpha T}}\right)\cdot\lambda_i\omega_i^2$$

$$\times\sum_i\left(\frac{1}{T}\mathbf{1}_{\mu_i(\mathbf{H}_{n_1,\beta^{tr}})\geq\frac{1}{\alpha T}}+T\alpha^2\mu_i(\mathbf{H}_{n_1,\beta^{tr}})^2\mathbf{1}_{\mu_i(\mathbf{H}_{n_1,\beta^{tr}})<\frac{1}{\alpha T}}\right)\cdot\frac{\mu_i(\mathbf{H}_{n_1,\beta^{tr}})}{\mu_i(\mathbf{H}_{m,\beta^{te}})}.$$

*Proof.* Applying Lemma B.6 to eq. (18), we can obtain:

$$\mathbf{S}_t \preceq \sum_{k=0}^{t-1}(\mathcal{I}-\alpha\widetilde{\mathcal{T}})^k \circ \left( \frac{\alpha c(\beta^{\mathrm{tr}},\boldsymbol{\Sigma})\cdot \mathrm{tr}\left(\boldsymbol{\Sigma}\mathbf{H}_{n_1,\beta^{\mathrm{tr}}}^{-1}\left[\mathcal{I}-(\mathcal{I}-\alpha\widetilde{\mathcal{T}})^T\right]\circ\mathbf{D}_0\right)}{1-c(\beta,\boldsymbol{\Sigma})\alpha\,\mathrm{tr}(\boldsymbol{\Sigma})}\cdot\mathbf{H}_{n_1,\beta^{\mathrm{tr}}}+\mathbf{D}_0\right)$$

$$=\sum_{k=0}^{t-1}(\mathbf{I}-\alpha\mathbf{H}_{n_1,\beta^{\mathrm{tr}}})^k\cdot$$

$$\left(\underbrace{\frac{\alpha c(\beta^{\mathrm{tr}},\boldsymbol{\Sigma})\cdot\mathrm{tr}\left(\boldsymbol{\Sigma}\mathbf{H}_{n_1,\beta^{\mathrm{tr}}}^{-1}(\mathbf{D}_0-(\mathbf{I}-\alpha\mathbf{H}_{n_1,\beta^{\mathrm{tr}}})^T\mathbf{D}_0(\mathbf{I}-\alpha\mathbf{H}_n)^T)\right)}{1-c(\beta^{\mathrm{tr}},\boldsymbol{\Sigma})\alpha\,\mathrm{tr}(\boldsymbol{\Sigma})}\cdot\mathbf{H}_{n_1,\beta^{\mathrm{tr}}}}_{\mathbf{G}_1}+\underbrace{\mathbf{D}_0}_{\mathbf{G}_2}\right)$$

$$\cdot(\mathbf{I}-\alpha\mathbf{H}_{n_1,\beta^{\mathrm{tr}}})^k.$$

Letting $t=T$, and substituting the upper bound of $\mathbf{S}_T$ into the bias term in Lemma B.3, we obtain:

$$\mathcal{E}_{\mathrm{bias}}\leq\frac{1}{\alpha T^2}\sum_{k=0}^{T-1}\left\langle((\mathbf{I}-\alpha\mathbf{H}_{n_1,\beta^{\mathrm{tr}}})^{2k}-(\mathbf{I}-\alpha\mathbf{H}_{n,\beta})^{T+2k})\mathbf{H}_{n,\beta}^{-1}\mathbf{H}_{m,\beta^{\mathrm{te}}},\mathbf{G}_1+\mathbf{G}_2\right\rangle$$

$$\leq\frac{1}{\alpha T^2}\sum_{k=0}^{T-1}\left\langle((\mathbf{I}-\alpha\mathbf{H}_{n_1,\beta^{\mathrm{tr}}})^{k}-(\mathbf{I}-\alpha\mathbf{H}_{n_1,\beta^{\mathrm{tr}}})^{T+k})\mathbf{H}_{n_1,\beta^{\mathrm{tr}}}^{-1}\mathbf{H}_{m,\beta^{\mathrm{te}}},\mathbf{G}_1+\mathbf{G}_2\right\rangle.$$

We first consider

$$d_1=\frac{1}{\alpha T^2}\sum_{k=0}^{T-1}\left\langle((\mathbf{I}-\alpha\mathbf{H}_{n_1,\beta^{\mathrm{tr}}})^{k}-(\mathbf{I}-\alpha\mathbf{H}_{n_1,\beta^{\mathrm{tr}}})^{T+k})\mathbf{H}_{n_1,\beta^{\mathrm{tr}}}^{-1}\mathbf{H}_{m,\beta^{\mathrm{te}}},\mathbf{G}_1\right\rangle.$$

Since $\mathbf{H}_{n_1,\beta^{\mathrm{tr}}}$, $\mathbf{H}_{m,\beta^{\mathrm{te}}}$ and $\mathbf{I}-\alpha\mathbf{H}_{n_1,\beta^{\mathrm{tr}}}$ commute, we have

$$d_1=\frac{c(\beta^{\mathrm{tr}},\boldsymbol{\Sigma})\cdot\mathrm{tr}\left(\boldsymbol{\Sigma}\mathbf{H}_{n_1,\beta^{\mathrm{tr}}}^{-1}(\mathbf{D}_0-(\mathbf{I}-\alpha\mathbf{H}_{n_1,\beta^{\mathrm{tr}}})^T\mathbf{D}_0(\mathbf{I}-\alpha\mathbf{H}_{n_1,\beta^{\mathrm{tr}}})^T)\right)}{(1-c(\beta^{\mathrm{tr}},\boldsymbol{\Sigma})\alpha\,\mathrm{tr}(\boldsymbol{\Sigma}))T^2}$$

$$\times\sum_{k=0}^{T-1}\left\langle\left((\mathbf{I}-\alpha\mathbf{H}_{n_1,\beta^{\mathrm{tr}}})^{k}-(\mathbf{I}-\alpha\mathbf{H}_{n_1,\beta^{\mathrm{tr}}})^{T+k}\right),\mathbf{H}_{m,\beta^{\mathrm{te}}}\right\rangle.$$

For the first term, since $\boldsymbol{\Sigma}$, $\mathbf{H}_{n_1,\beta^{\mathrm{tr}}}$ and $\mathbf{I}-\alpha\mathbf{H}_{n_1,\beta^{\mathrm{tr}}}$ can be diagonalized simultaneously, considering the eigen-decompositions under the basis of $\boldsymbol{\Sigma}$ and recalling $\boldsymbol{\Sigma}=\mathbf{V}\boldsymbol{\Lambda}\mathbf{V}^\top$, we have:

$$\mathrm{tr}\left(\boldsymbol{\Sigma}\mathbf{H}_{n_1,\beta^{\mathrm{tr}}}^{-1}[\mathbf{D}_0-(\mathbf{I}-\alpha\mathbf{H}_{n_1,\beta^{\mathrm{tr}}})^T\mathbf{D}_0(\mathbf{I}-\alpha\mathbf{H}_{n_1,\beta^{\mathrm{tr}}})^T]\right)$$

$$=\sum_i\left(1-(1-\alpha\mu_i(\mathbf{H}_{n_1,\beta^{\mathrm{tr}}}))^{2T}\right)\cdot(\langle\mathbf{w}_0-\mathbf{w}^*,\mathbf{v}_i\rangle)^2\frac{\lambda_i}{\mu_i(\mathbf{H}_{n_1,\beta^{\mathrm{tr}}})}$$

$$\leq 2\sum_i\left(\mathbf{1}_{\lambda_i(\mathbf{H}_{n,\beta})\geq\frac{1}{\alpha T}}+T\alpha\mu_i(\mathbf{H}_{n,\beta})\mathbf{1}_{\mu_i(\mathbf{H}_{n_1,\beta^{\mathrm{tr}}})<\frac{1}{\alpha T}}\right)\cdot(\langle\mathbf{w}_0-\mathbf{w}^*,\mathbf{v}_i\rangle)^2\frac{\lambda_i}{\mu_i(\mathbf{H}_{n_1,\beta^{\mathrm{tr}}})}$$

where the last inequality holds since $1-(1-\alpha x)^{2T}\leq\min\{2,2T\alpha x\}$.

For the second term, similarly, $\mathbf{H}_{m,\beta^{\mathrm{te}}}$ and $\mathbf{I}-\alpha\mathbf{H}_{n_1,\beta^{\mathrm{tr}}}$ can be diagonalized simultaneously. We then have

$$\sum_{k=0}^{T-1}\left\langle\left((\mathbf{I}-\alpha\mathbf{H}_{n_1,\beta^{\mathrm{tr}}})^{k}-(\mathbf{I}-\alpha\mathbf{H}_{n_1,\beta^{\mathrm{tr}}})^{T+k}\right),\mathbf{H}_{m,\beta^{\mathrm{te}}}\right\rangle$$

$$\leq\sum_{k=0}^{T-1}\sum_i[(1-\alpha\mu_i(\mathbf{H}_{n_1,\beta^{\mathrm{tr}}}))^{k}-(1-\alpha\mu_i(\mathbf{H}_{n_1,\beta^{\mathrm{tr}}}))^{T+k}]\mu_i(\mathbf{H}_{m,\beta^{\mathrm{te}}})$$

$$=\frac{1}{\alpha}\sum_i[1-(1-\alpha\mu_i(\mathbf{H}_{n_1,\beta^{\mathrm{tr}}}))^T]^2\frac{\mu_i(\mathbf{H}_{m,\beta^{\mathrm{te}}})}{\mu_i(\mathbf{H}_{n_1,\beta^{\mathrm{tr}}})}$$

$$\leq\frac{1}{\alpha}\sum_i\left(\mathbf{1}_{\lambda_i(\mathbf{H}_{n,\beta})\geq\frac{1}{\alpha T}}+T^2\alpha^2\lambda_i(\mathbf{H}_{n,\beta})\mathbf{1}_{\lambda_i(\mathbf{H}_{n,\beta})<\frac{1}{\alpha T}}\right)\frac{\mu_i(\mathbf{H}_{m,\beta^{\mathrm{te}}})}{\mu_i(\mathbf{H}_{n_1,\beta^{\mathrm{tr}}})}.$$

Now we turn to:

$$d_2 = \frac{1}{\alpha T^2} \sum_{k=0}^{T-1} \left\langle ((\mathbf{I} - \alpha \mathbf{H}_{n_1, \beta^{\text{tr}}})^k - (\mathbf{I} - \alpha \mathbf{H}_{n_1, \beta^{\text{tr}}})^{T+k}) \mathbf{H}_{n_1, \beta^{\text{tr}}}^{-1} \mathbf{H}_{m, \beta^{\text{te}}}, \mathbf{G}_2 \right\rangle.$$

Considering the orthogonal decompositions of $\mathbf{H}_{m, \beta^{\text{te}}}$ and $\mathbf{H}_{n_1, \beta^{\text{tr}}}$ under $\mathbf{V}$, $\mathbf{H}_{n_1, \beta^{\text{tr}}} = \mathbf{V} \mathbf{\Lambda}_1 \mathbf{V}^\top$, $\mathbf{H}_{m, \beta^{\text{te}}} = \mathbf{V} \mathbf{\Lambda}_2 \mathbf{V}^\top$, where the diagonal entries of $\mathbf{\Lambda}_1$ are $\mu_i(\mathbf{H}_{n_1, \beta^{\text{tr}}})$ (and $\mu_i(\mathbf{H}_{m, \beta^{\text{te}}})$ for $\mathbf{\Lambda}_2$). Then we have:

$$d_2 = \frac{1}{\alpha T^2} \sum_{k=0}^{T-1} \left\langle \underbrace{((\mathbf{I} - \alpha \mathbf{\Lambda}_1)^k - (\mathbf{I} - \alpha \mathbf{\Lambda}_1)^{T+k}) \mathbf{\Lambda}_1^{-1} \mathbf{\Lambda}_2}_{\mathbf{J}_3}, \mathbf{V}^\top \mathbf{D}_0 \mathbf{V} \right\rangle$$

$$= \frac{1}{\alpha T^2} \sum_{k=0}^{T-1} \sum_i \left[ (1 - \alpha \mu_i(\mathbf{H}_{n_1, \beta^{\text{tr}}}))^k - (1 - \alpha \mu_i(\mathbf{H}_{n_1, \beta^{\text{tr}}}))^{T+k} \right] \frac{\omega_i^2 \mu_i(\mathbf{H}_{m, \beta^{\text{te}}})}{\mu_i(\mathbf{H}_{n_1, \beta^{\text{tr}}})}$$

$$= \frac{1}{\alpha^2 T^2} \sum_i \left[ 1 - (1 - \alpha \mu_i(\mathbf{H}_{n_1, \beta^{\text{tr}}}))^T \right]^2 \frac{\omega_i^2 \mu_i(\mathbf{H}_{m, \beta^{\text{te}}})}{\mu_i^2(\mathbf{H}_{n_1, \beta^{\text{tr}}})}$$

$$\leq \frac{1}{\alpha^2 T^2} \sum_i \left( \mathbf{1}_{\mu_i(\mathbf{H}_{n_1, \beta^{\text{tr}}}) \geq \frac{1}{\alpha T}} + \alpha^2 T^2 \mu_i^2(\mathbf{H}_{n_1, \beta^{\text{tr}}}) \mathbf{1}_{\mu_i(\mathbf{H}_{n_1, \beta^{\text{tr}}}) < \frac{1}{\alpha T}} \right) \frac{\omega_i^2 \mu_i(\mathbf{H}_{m, \beta^{\text{te}}})}{\mu_i^2(\mathbf{H}_{n_1, \beta^{\text{tr}}})}$$

where $\omega_i = \langle \boldsymbol{\omega}_0 - \boldsymbol{\theta}^*, \mathbf{v}_i \rangle$ is the diagonal entry of $\mathbf{V}^\top \mathbf{D}_0 \mathbf{V}$ and the second equality holds since $\mathbf{J}_3$ is a diagonal matrix. $\qquad \square$

### B.5  Bounding the Variance

Note that the noisy part $\Pi = \mathbb{E}[\mathbf{B}^\top \boldsymbol{\xi} \boldsymbol{\xi}^\top \mathbf{B}]$ in eq. (12) is important in the variance iterates. In order to analyze the variance term, we first understand the role of $\Pi$ by the following lemma.

**Lemma B.8** (Bounding the noise).

$$\Pi = \mathbb{E}[\mathbf{B}^\top \boldsymbol{\xi} \boldsymbol{\xi}^\top \mathbf{B}] \preceq f(\beta^{tr}, n_2, \sigma, \mathbf{\Sigma}, \mathbf{\Sigma}_{\boldsymbol{\theta}}) \mathbf{H}_{n_1, \beta^{tr}}$$

*where* $f(\beta, n, \sigma, \mathbf{\Sigma}, \mathbf{\Sigma}_{\boldsymbol{\theta}}) = [c(\beta, \mathbf{\Sigma}) \operatorname{tr}(\mathbf{\Sigma}_{\boldsymbol{\theta}} \mathbf{\Sigma}) + 4c_1 \sigma^2 \sigma_x^2 \beta^2 \sqrt{C(\beta, \mathbf{\Sigma})} \operatorname{tr}(\mathbf{\Sigma}^2) + \sigma^2/n]$.

*Proof.* With a slight abuse of notations, we write $\beta^{\text{tr}}$ as $\beta$ in this proof. By definition of meta data and noise, we have

$$\Pi = \mathbb{E}[\mathbf{B}^\top \boldsymbol{\xi} \boldsymbol{\xi}^\top \mathbf{B}]$$

$$= \frac{\sigma^2}{n_2} \mathbf{H}_{n_1, \beta} + \mathbb{E}[\mathbf{B}^\top \mathbf{B} \mathbf{\Sigma}_{\boldsymbol{\theta}} \mathbf{B}^\top \mathbf{B}] + \sigma^2 \cdot \frac{\beta^2}{n_2 n_1^2} \mathbb{E}[\mathbf{B}^\top \mathbf{X}^{\text{out}} \mathbf{X}^{\text{in}\top} \mathbf{X}^{\text{in}} \mathbf{X}^{\text{out}\top} \mathbf{B}].$$

The second term can be directly bounded by Proposition B.1:

$$\mathbb{E}[\mathbf{B}^\top \mathbf{B} \mathbf{\Sigma}_{\boldsymbol{\theta}} \mathbf{B}^\top \mathbf{B}] \preceq c(\beta, \mathbf{\Sigma}) \operatorname{tr}(\mathbf{\Sigma}_{\boldsymbol{\theta}} \mathbf{\Sigma}) \mathbf{H}_{n_1, \beta}.$$

For the third term, we utilize the technique similar to Proposition B.1, and by Assumption 1, we have:

$$\sigma^2 \cdot \frac{\beta^2}{n_2 n_1^2} \mathbb{E} \left[ \mathbf{B}^\top \mathbf{X}^{\text{out}} \mathbf{X}^{\text{in}\top} \mathbf{X}^{\text{in}} \mathbf{X}^{\text{out}\top} \mathbf{B} \right]$$

$$\preceq \sigma^2 c_1 \cdot \frac{\beta^2}{n_1^2} \mathbb{E} \left[ \operatorname{tr}(\mathbf{X}^{\text{in}\top} \mathbf{X}^{\text{in}} \mathbf{\Sigma})(\mathbf{I} - \frac{\beta}{n_1} \mathbf{X}^{\text{in}\top} \mathbf{X}^{\text{in}}) \mathbf{\Sigma} (\mathbf{I} - \frac{\beta}{n_1} \mathbf{X}^{\text{in}\top} \mathbf{X}^{\text{in}}) \right].$$

Following the analysis for $\mathbf{J}_1$ in the proof of Proposition B.1, and letting $\mathbf{A} = \mathbf{I}$, we obtain:

$$\frac{1}{n_1^2} \mathbb{E} \left[ \operatorname{tr}(\mathbf{X}^{\text{in}\top} \mathbf{X}^{\text{in}} \mathbf{\Sigma})(\mathbf{I} - \frac{\beta}{n_1} \mathbf{X}^{\text{in}\top} \mathbf{X}^{\text{in}}) \mathbf{\Sigma} (\mathbf{I} - \frac{\beta}{n_1} \mathbf{X}^{\text{in}\top} \mathbf{X}^{\text{in}}) \right] \preceq 4 \sqrt{C(\beta, \mathbf{\Sigma})} \sigma_x^2 \operatorname{tr}(\mathbf{\Sigma}^2) \mathbf{H}_{n_1, \beta}.$$

Putting all these results together completes the proof. $\qquad \square$

**Lemma B.9** (Property of $\mathbf{V}_t$). *If the stepsize satisfies* $\alpha < \frac{1}{c(\beta^{tr}, \mathbf{\Sigma}) \operatorname{tr}(\mathbf{\Sigma})}$, *it holds that*

$$\mathbf{0} = \mathbf{V}_0 \preceq \mathbf{V}_1 \preceq \cdots \preceq \mathbf{V}_\infty \preceq \frac{\alpha f(\beta^{tr}, n_2, \sigma, \mathbf{\Sigma}, \mathbf{\Sigma}_{\boldsymbol{\theta}})}{1 - \alpha c(\beta^{tr}, \mathbf{\Sigma}) \operatorname{tr}(\mathbf{\Sigma})} \mathbf{I}.$$

*Proof.* Similar calculations has appeared in prior works [14, 22]. However, our analysis of the meta linear model needs to handle the complicated meta noise, and hence we provide a proof here for completeness.

We first show that $\mathbf{V}_{t-1} \preceq \mathbf{V}_t$. By recursion:

$$\mathbf{V}_t = (\mathcal{I} - \alpha \mathcal{T}) \circ \mathbf{V}_{t-1} + \alpha^2 \Pi$$

$$\overset{(a)}{=} \alpha^2 \sum_{k=0}^{t-1} (\mathcal{I} - \alpha \mathcal{T})^k \circ \Pi$$

$$= \mathbf{V}_{t-1} + \alpha^2 (\mathcal{I} - \alpha \mathcal{T})^{t-1} \circ \Pi$$

$$\overset{(b)}{\succeq} \mathbf{V}_{t-1}$$

where $(a)$ holds by solving the recursion and $(b)$ follows because $\mathcal{I} - \alpha \mathcal{T}$ is a PSD mapping.

The existence of $\mathbf{V}_\infty$ can be shown in the way similar to the proof of Lemma B.1. We first have

$$\mathbf{V}_t = \alpha^2 \sum_{k=0}^{t-1} (\mathcal{I} - \alpha \mathcal{T})^k \circ \Pi \preceq \alpha^2 \sum_{k=0}^{\infty} \underbrace{(\mathcal{I} - \alpha \mathcal{T})^k \circ \Pi}_{\mathbf{A}_k}.$$

By previous analysis in Lemma B.1 , if $\alpha < \frac{1}{c(\beta^{\mathrm{tr}}, \mathbf{\Sigma}) \operatorname{tr}(\mathbf{\Sigma})}$, we have

$$\operatorname{tr}(\mathbf{A}_k) \leq \left( 1 - \alpha \min_i \{\mu_i(\mathbf{H}_{n_1, \beta^{\mathrm{tr}}})\} \right) \operatorname{tr}(\mathbf{A}_{t-1}).$$

Therefore,

$$\operatorname{tr}(\mathbf{V}_t) \leq \alpha^2 \sum_{k=0}^{\infty} \operatorname{tr}(\mathbf{A}_k) \leq \frac{\alpha \operatorname{tr}(\Pi)}{\min_i \{\mu_i(\mathbf{H}_{n_1, \beta^{\mathrm{tr}}})\}} < \infty.$$

The trace of $\mathbf{V}_t$ is uniformly bounded from above, which indicates that $\mathbf{V}_\infty$ exists.

Finally, we bound $\mathbf{V}_\infty$. Note that $\mathbf{V}_\infty$ is the solution to:

$$\mathbf{V}_\infty = (\mathcal{I} - \alpha \mathcal{T}) \circ \mathbf{V}_\infty + \alpha^2 \Pi.$$

Then we can write $\mathbf{V}_\infty$ as $\mathbf{V}_\infty = \mathcal{T}^{-1} \circ \alpha \Pi$. Following the analysis in the proof of Lemma B.5, we have:

$$\widetilde{\mathcal{T}} \circ \mathbf{V}_\infty = \widetilde{\mathcal{T}} \circ \mathcal{T}^{-1} \circ \alpha \Pi$$

$$\preceq \alpha \Pi + \alpha \mathcal{M} \circ \mathbf{V}_\infty$$

$$\preceq \alpha f(\beta^{\mathrm{tr}}, n_2, \sigma, \mathbf{\Sigma}, \mathbf{\Sigma}_{\boldsymbol{\theta}}) \mathbf{H}_{n_1, \beta^{\mathrm{tr}}} + \alpha \mathcal{M} \circ \mathbf{V}_\infty$$

where the last inequality follows from Lemma B.8. Applying $\widetilde{\mathcal{T}}^{-1}$, which exists and is a PSD mapping, to the both sides, we have

$$\mathbf{V}_\infty \preceq \alpha f(\beta^{\mathrm{tr}}, n_2, \sigma, \mathbf{\Sigma}, \mathbf{\Sigma}_{\boldsymbol{\theta}}) \cdot \widetilde{\mathcal{T}}^{-1} \circ \mathbf{H}_{n_1, \beta^{\mathrm{tr}}} + \alpha \widetilde{\mathcal{T}}^{-1} \circ \mathcal{M} \circ \mathbf{V}_\infty$$

$$\overset{(a)}{\preceq} \alpha f(\beta^{\mathrm{tr}}, n_2, \sigma, \mathbf{\Sigma}, \mathbf{\Sigma}_{\boldsymbol{\theta}}) \cdot \sum_{t=0}^{\infty} \left( \alpha \widetilde{\mathcal{T}}^{-1} \circ \mathcal{M} \right)^t \circ \widetilde{\mathcal{T}}^{-1} \circ \mathbf{H}_{n_1, \beta^{\mathrm{tr}}}$$

$$\overset{(b)}{\preceq} \alpha f(\beta^{\mathrm{tr}}, n_2, \sigma, \mathbf{\Sigma}, \mathbf{\Sigma}_{\boldsymbol{\theta}}) \sum_{t=0}^{\infty} (\alpha c(\beta^{\mathrm{tr}}, \mathbf{\Sigma}) \operatorname{tr}(\mathbf{\Sigma}))^t \mathbf{I}$$

$$= \frac{\alpha f(\beta^{\mathrm{tr}}, n_2, \sigma, \mathbf{\Sigma}, \mathbf{\Sigma}_{\boldsymbol{\theta}})}{1 - \alpha c(\beta^{\mathrm{tr}}, \mathbf{\Sigma}) \operatorname{tr}(\mathbf{\Sigma})} \mathbf{I}$$

where $(a)$ holds by directly solving the recursion; $(b)$ follows from the fact that $\widetilde{\mathcal{T}}^{-1} \circ \mathbf{H}_{n_1, \beta^{\mathrm{tr}}} \preceq \mathbf{I}$ from Lemma B.1 and $\mathcal{M} \circ \mathbf{I} \preceq c(\beta^{\mathrm{tr}}, \mathbf{\Sigma}) \operatorname{tr}(\mathbf{\Sigma}) \mathbf{H}_{n_1, \beta^{\mathrm{tr}}}$ by letting $\mathbf{A} = \mathbf{I}$ in Proposition B.1. $\square$

Now we are ready to provide the upper bound on the variance term.

**Lemma B.10** (Bounding the Variance). *If* $\alpha < \frac{1}{c(\beta^{tr},\boldsymbol{\Sigma})\operatorname{tr}(\boldsymbol{\Sigma})}$, *for sufficiently large* $n_1$, *s.t.* $\mu_i(\mathbf{H}_{n_1,\beta^{tr}}) > 0$, $\forall i$, *then we have*

$$\mathcal{E}_{var} \leq \frac{f(\beta^{tr}, n_2, \sigma, \boldsymbol{\Sigma}, \boldsymbol{\Sigma_\theta})}{(1 - \alpha c(\beta^{tr}, \boldsymbol{\Sigma})\operatorname{tr}(\boldsymbol{\Sigma}))}$$

$$\times \sum_i \left( \frac{1}{T} \mathbf{1}_{\mu_i(\mathbf{H}_{n_1,\beta^{tr}}) \geq \frac{1}{\alpha T}} + T\alpha^2 \mu_i^2(\mathbf{H}_{n_1,\beta^{tr}}) \mathbf{1}_{\mu_i(\mathbf{H}_{n_1,\beta^{tr}}) < \frac{1}{\alpha T}} \right) \frac{\mu_i(\mathbf{H}_{m,\beta^{te}})}{\mu_i(\mathbf{H}_{n_1,\beta^{tr}})}.$$

*Proof.* Recall

$$
\begin{aligned}
\mathbf{V}_t &= (\mathcal{I} - \alpha\mathcal{T}) \circ \mathbf{V}_{t-1} + \alpha^2 \Pi \\
&= (\mathcal{I} - \alpha\widetilde{\mathcal{T}}) \circ \mathbf{V}_{t-1} + \alpha^2(\mathcal{M} - \widetilde{\mathcal{M}}) \circ \mathbf{V}_{t-1} + \alpha^2 \Pi \\
&\preceq (\mathcal{I} - \alpha\widetilde{\mathcal{T}}) \circ \mathbf{V}_{t-1} + \alpha^2 \mathcal{M} \circ \mathbf{V}_{t-1} + \alpha^2 \Pi.
\end{aligned}
\tag{20}
$$

By the uniform bound on $\mathbf{V}_t$ and $\mathcal{M}$ is a PSD mapping, we have:

$$
\begin{aligned}
\mathcal{M} \circ \mathbf{V}_t &\preceq \mathcal{M} \circ \mathbf{V}_\infty \\
&\stackrel{(a)}{\preceq} \mathcal{M} \circ \frac{\alpha f(\beta^{tr}, n_2, \sigma, \boldsymbol{\Sigma}, \boldsymbol{\Sigma_\theta})}{1 - \alpha c(\beta^{tr}, \boldsymbol{\Sigma})\operatorname{tr}(\boldsymbol{\Sigma})} \mathbf{I} \\
&\stackrel{(b)}{\preceq} \frac{\alpha f(\beta^{tr}, n_2, \sigma, \boldsymbol{\Sigma}, \boldsymbol{\Sigma_\theta}) c(\beta^{tr}, \boldsymbol{\Sigma})\operatorname{tr}(\boldsymbol{\Sigma})}{1 - \alpha c(\beta^{tr}, \boldsymbol{\Sigma})\operatorname{tr}(\boldsymbol{\Sigma})} \cdot \mathbf{H}_{n_1,\beta^{tr}}
\end{aligned}
$$

where $(a)$ directly follows from Lemma B.9; $(b)$ holds because $\mathcal{M} \circ \mathbf{I} \preceq c(\beta^{tr}, \boldsymbol{\Sigma})\operatorname{tr}(\boldsymbol{\Sigma})\mathbf{H}_{n_1,\beta^{tr}}$ (letting $\mathbf{A} = \mathbf{I}$ in Proposition B.1). Substituting it back into eq. (20), we have:

$$
\begin{aligned}
\mathbf{V}_t &\preceq (\mathcal{I} - \alpha\widetilde{\mathcal{T}}) \circ \mathbf{V}_{t-1} + \alpha^2 \frac{\alpha f c(\beta^{tr}, \boldsymbol{\Sigma})\operatorname{tr}(\boldsymbol{\Sigma})}{1 - \alpha c(\beta^{tr}, \boldsymbol{\Sigma})\operatorname{tr}(\boldsymbol{\Sigma})} \cdot \mathbf{H}_{n_1,\beta^{tr}} + \alpha^2 f \mathbf{H}_{n_1,\beta^{tr}} \\
&= (\mathcal{I} - \alpha\widetilde{\mathcal{T}}) \circ \mathbf{V}_{t-1} + \frac{\alpha^2 f(\beta^{tr}, n_2, \sigma, \boldsymbol{\Sigma}, \boldsymbol{\Sigma_\theta})}{1 - \alpha c(\beta^{tr}, \boldsymbol{\Sigma})\operatorname{tr}(\boldsymbol{\Sigma})} \mathbf{H}_{n_1,\beta^{tr}} \\
&\stackrel{(a)}{=} \frac{\alpha^2 f(\beta^{tr}, n_2, \sigma, \boldsymbol{\Sigma}, \boldsymbol{\Sigma_\theta})}{1 - \alpha c(\beta^{tr}, \boldsymbol{\Sigma})\operatorname{tr}(\boldsymbol{\Sigma})} \sum_{k=0}^{t-1} (\mathbf{I} - \alpha\widetilde{\mathcal{T}})^k \circ \mathbf{H}_{n_1,\beta^{tr}} \\
&\stackrel{(b)}{\preceq} \frac{\alpha f(\beta^{tr}, n_2, \sigma, \boldsymbol{\Sigma}, \boldsymbol{\Sigma_\theta})}{1 - \alpha c(\beta^{tr}, \boldsymbol{\Sigma})\operatorname{tr}(\boldsymbol{\Sigma})} (\mathbf{I} - (\mathbf{I} - \alpha\mathbf{H}_{n,\beta})^t)
\end{aligned}
$$

where $(a)$ holds by solving the recursion and $(b)$ is due to the fact that

$$
\begin{aligned}
\sum_{k=0}^{t-1} (\mathbf{I} - \alpha\widetilde{\mathcal{T}})^k \circ \mathbf{H}_{n_1,\beta^{tr}} &= \sum_{k=0}^{t-1} (\mathbf{I} - \alpha\mathbf{H}_{n_1,\beta^{tr}})^k \mathbf{H}_{n_1,\beta^{tr}} (\mathbf{I} - \alpha\mathbf{H}_{n_1,\beta^{tr}})^k \\
&\preceq \sum_{k=0}^{t-1} (\mathbf{I} - \alpha\mathbf{H}_{n_1,\beta^{tr}})^k \mathbf{H}_{n_1,\beta^{tr}} \\
&= \frac{1}{\alpha}[\mathbf{I} - (\mathbf{I} - \alpha\mathbf{H}_{n_1,\beta^{tr}})^t].
\end{aligned}
$$

Substituting the bound for $\mathbf{V}_t$ back into the variance term in Lemma B.3, we have

$$
\begin{aligned}
\mathcal{E}_{var} &\leq \frac{1}{T^2} \sum_{t=0}^{T-1} \sum_{k=t}^{T-1} \left\langle (\mathbf{I} - \alpha\mathbf{H}_{n_1,\beta^{tr}})^{k-t} \mathbf{H}_{m,\beta^{te}}, \mathbf{V}_t \right\rangle \\
&= \frac{1}{\alpha T^2} \sum_{t=0}^{T-1} \left\langle (\mathbf{I} - (\mathbf{I} - \alpha\mathbf{H}_{n_1,\beta^{tr}})^{T-t}) \mathbf{H}_{n_1,\beta^{tr}}^{-1} \mathbf{H}_{m,\beta^{te}}, \mathbf{V}_t \right\rangle \\
&\leq \frac{f(\beta^{tr}, n_2, \sigma, \boldsymbol{\Sigma}, \boldsymbol{\Sigma_\theta})}{(1 - \alpha c(\beta^{tr}, \boldsymbol{\Sigma})\operatorname{tr}(\boldsymbol{\Sigma}))T^2} \sum_{t=0}^{T-1} \left\langle \mathbf{I} - (\mathbf{I} - \alpha\mathbf{H}_{n,\beta})^{T-t}, (\mathbf{I} - (\mathbf{I} - \alpha\mathbf{H}_{n,\beta})^t) \mathbf{H}_{n,\beta}^{-1} \mathbf{H}_{m,\eta} \right\rangle.
\end{aligned}
$$

Simultaneously diagonalizing $\mathbf{H}_{n_1,\beta^{\mathrm{tr}}}$ and $\mathbf{H}_{m,\beta^{\mathrm{te}}}$ as the analysis in Lemma B.7, we have

$$
\begin{aligned}
\mathcal{E}_{\mathrm{var}} \leq & \frac{f(\beta^{\mathrm{tr}}, n_2, \sigma, \boldsymbol{\Sigma}, \boldsymbol{\Sigma}_{\boldsymbol{\theta}})}{(1 - \alpha c(\beta^{\mathrm{tr}}, \boldsymbol{\Sigma}) \operatorname{tr}(\boldsymbol{\Sigma})) T^2} \\
& \cdot \sum_i \sum_{t=0}^{T-1} \left(1 - (1 - \alpha \mu_i(\mathbf{H}_{n_1,\beta^{\mathrm{tr}}}))^{T-t}\right) \left(1 - (1 - \alpha \mu_i(\mathbf{H}_{n_1,\beta^{\mathrm{tr}}}))^t\right) \frac{\mu_i(\mathbf{H}_{m,\beta^{\mathrm{te}}})}{\mu_i(\mathbf{H}_{n_1,\beta^{\mathrm{tr}}})} \\
\leq & \frac{f(\beta^{\mathrm{tr}}, n_2, \sigma, \boldsymbol{\Sigma}, \boldsymbol{\Sigma}_{\boldsymbol{\theta}})}{(1 - \alpha c(\beta^{\mathrm{tr}}, \boldsymbol{\Sigma}) \operatorname{tr}(\boldsymbol{\Sigma})) T^2} \\
& \cdot \sum_i \sum_{t=0}^{T-1} \left(1 - (1 - \alpha \mu_i(\mathbf{H}_{n_1,\beta^{\mathrm{tr}}}))^{T}\right) \left(1 - (1 - \alpha \mu_i(\mathbf{H}_{n_1,\beta^{\mathrm{tr}}}))^T\right) \frac{\mu_i(\mathbf{H}_{m,\beta^{\mathrm{te}}})}{\mu_i(\mathbf{H}_{n_1,\beta^{\mathrm{tr}}})} \\
= & \frac{f(\beta^{\mathrm{tr}}, n_2, \sigma, \boldsymbol{\Sigma}, \boldsymbol{\Sigma}_{\boldsymbol{\theta}})}{(1 - \alpha c(\beta^{\mathrm{tr}}, \boldsymbol{\Sigma}) \operatorname{tr}(\boldsymbol{\Sigma})) T} \sum_i \left(1 - (1 - \alpha \mu_i(\mathbf{H}_{n_1,\beta^{\mathrm{tr}}}))^T\right)^2 \frac{\mu_i(\mathbf{H}_{m,\beta^{\mathrm{te}}})}{\mu_i(\mathbf{H}_{n_1,\beta^{\mathrm{tr}}})} \\
\leq & \frac{f(\beta^{\mathrm{tr}}, n_2, \sigma, \boldsymbol{\Sigma}, \boldsymbol{\Sigma}_{\boldsymbol{\theta}})}{(1 - \alpha c(\beta^{\mathrm{tr}}, \boldsymbol{\Sigma}) \operatorname{tr}(\boldsymbol{\Sigma})) T} \sum_i \left(\min\{1, \alpha T \mu_i(\mathbf{H}_{n_1,\beta^{\mathrm{tr}}})\}\right)^2 \frac{\mu_i(\mathbf{H}_{m,\beta^{\mathrm{te}}})}{\mu_i(\mathbf{H}_{n_1,\beta^{\mathrm{tr}}})} \\
\leq & \frac{f(\beta^{\mathrm{tr}}, n_2, \sigma, \boldsymbol{\Sigma}, \boldsymbol{\Sigma}_{\boldsymbol{\theta}})}{(1 - \alpha c(\beta^{\mathrm{tr}}, \boldsymbol{\Sigma}) \operatorname{tr}(\boldsymbol{\Sigma}))} \\
& \cdot \sum_i \left(\frac{1}{T} \mathbf{1}_{\mu_i(\mathbf{H}_{n_1,\beta^{\mathrm{tr}}}) \geq \frac{1}{\alpha T}} + T \alpha^2 \mu_i^2(\mathbf{H}_{n_1,\beta^{\mathrm{tr}}}) \mathbf{1}_{\mu i(\mathbf{H}_{n,\beta}) < \frac{1}{\alpha T}}\right) \frac{\mu_i(\mathbf{H}_{m,\beta^{\mathrm{te}}})}{\mu_i(\mathbf{H}_{n_1,\beta^{\mathrm{tr}}})},
\end{aligned}
$$

which completes the proof. $\qquad\square$

## B.6  Proof of Theorem 1

**Theorem B.3** (Theorem 1 Restated). *Let* $\omega_i = \langle \boldsymbol{\omega}_0 - \boldsymbol{\theta}^*, \mathbf{v}_i \rangle$. *If* $|\beta^{\mathrm{tr}}|, |\beta^{\mathrm{te}}| < 1/\lambda_1$, $n_1$ *is large ensuring that* $\mu_i(\mathbf{H}_{n_1,\beta^{\mathrm{tr}}}) > 0$, $\forall i$ *and* $\alpha < 1/(c(\beta^{\mathrm{tr}}, \boldsymbol{\Sigma}) \operatorname{tr}(\boldsymbol{\Sigma}))$, *then the meta excess risk* $R(\overline{\boldsymbol{\omega}}_T, \beta^{\mathrm{te}})$ *is bounded above as follows*

$$
R(\overline{\boldsymbol{\omega}}_T, \beta^{\mathrm{te}}) \leq \mathit{Bias} + \mathit{Var}
$$

*where*

$$
\mathit{Bias} = \frac{2}{\alpha^2 T} \sum_i \Xi_i \frac{\omega_i^2}{\mu_i(\mathbf{H}_{n_1,\beta^{\mathrm{tr}}})}
$$

$$
\begin{aligned}
\mathit{Var} = & \frac{2}{(1 - \alpha c(\beta^{\mathrm{tr}}, \boldsymbol{\Sigma}) \operatorname{tr}(\boldsymbol{\Sigma}))} \left(\sum_i \Xi_i\right) \\
& \times [f(\beta^{\mathrm{tr}}, n_2, \sigma, \boldsymbol{\Sigma}_{\boldsymbol{\theta}}, \boldsymbol{\Sigma}) + 2c(\beta^{\mathrm{tr}}, \boldsymbol{\Sigma}) \sum_i \left(\frac{\mathbf{1}_{\mu_i(\mathbf{H}_{n_1,\beta^{\mathrm{tr}}}) \geq \frac{1}{\alpha T}}}{T\alpha \mu_i(\mathbf{H}_{n_1,\beta^{\mathrm{tr}}})} + \mathbf{1}_{\mu_i(\mathbf{H}_{n_1,\beta^{\mathrm{tr}}}) < \frac{1}{\alpha T}}\right) \lambda_i \omega_i^2].
\end{aligned}
$$

*Proof.* By Lemma B.2, we have

$$
R(\overline{\boldsymbol{\omega}}_T, \beta^{\mathrm{te}}) \leq 2\mathcal{E}_{\mathrm{bias}} + 2\mathcal{E}_{\mathrm{var}}.
$$

Using Lemma B.7 to bound $\mathcal{E}_{\text{bias}}$, and Lemma B.10 to bound $\mathcal{E}_{\text{var}}$, we have

$$
\begin{aligned}
&R(\overline{\boldsymbol{\omega}}_T, \beta^{\text{te}}) \\
&\leq \frac{2f(\beta^{\text{tr}}, n_2, \sigma, \boldsymbol{\Sigma}, \boldsymbol{\Sigma}_{\boldsymbol{\theta}})}{(1 - \alpha c(\beta^{\text{tr}}, \boldsymbol{\Sigma})\operatorname{tr}(\boldsymbol{\Sigma}))} \\
&\quad \times \sum_i \left( \frac{1}{T} \mathbf{1}_{\mu_i(\mathbf{H}_{n_1,\beta^{\text{tr}}}) \geq \frac{1}{\alpha T}} + T\alpha^2 \mu_i^2(\mathbf{H}_{n_1,\beta^{\text{tr}}}) \mathbf{1}_{\mu_i(\mathbf{H}_{n_1,\beta^{\text{tr}}}) < \frac{1}{\alpha T}} \right) \frac{\mu_i(\mathbf{H}_{m,\beta^{\text{te}}})}{\mu_i(\mathbf{H}_{n_1,\beta^{\text{tr}}})} \\
&\quad + \frac{4c(\beta^{\text{tr}}, \boldsymbol{\Sigma})}{T\alpha(1 - c(\beta^{\text{tr}}, \boldsymbol{\Sigma})\alpha \operatorname{tr}(\boldsymbol{\Sigma}))} \sum_i \left( \frac{1}{T} \mathbf{1}_{\mu_i(\mathbf{H}_{n_1,\beta^{\text{tr}}}) \geq \frac{1}{\alpha T}} + T\alpha^2 \mu_i(\mathbf{H}_{n_1,\beta^{\text{tr}}})^2 \mathbf{1}_{\mu_i(\mathbf{H}_{n_1,\beta^{\text{tr}}}) < \frac{1}{\alpha T}} \right) \\
&\quad \times \sum_i \left( \frac{1}{\mu_i(\mathbf{H}_{n_1,\beta^{\text{tr}}})} \mathbf{1}_{\mu_i(\mathbf{H}_{n_1,\beta^{\text{tr}}}) \geq \frac{1}{\alpha T}} + T\alpha \mathbf{1}_{\mu_i(\mathbf{H}_{n_1,\beta^{\text{tr}}}) < \frac{1}{\alpha T}} \right) \cdot \lambda_i \left( \langle \boldsymbol{\omega}_0 - \boldsymbol{\theta}^*, \mathbf{v}_i \rangle \right)^2 \\
&\quad + 2\sum_i \left( \frac{1}{\alpha^2 T^2} \mathbf{1}_{\mu_i(\mathbf{H}_{n_1,\beta^{\text{tr}}}) \geq \frac{1}{\alpha T}} + \mu_i^2(\mathbf{H}_{n_1,\beta^{\text{tr}}}) \mathbf{1}_{\mu_i(\mathbf{H}_{n_1,\beta^{\text{tr}}}) < \frac{1}{\alpha T}} \right) \frac{\omega_i^2 \mu_i(\mathbf{H}_{m,\beta^{\text{te}}})}{\mu_i(\mathbf{H}_{n_1,\beta^{\text{tr}}})^2}.
\end{aligned}
$$

Incorporating with the definition of effective meta weight

$$
\Xi_i(\boldsymbol{\Sigma}, \alpha, T) = \begin{cases} \mu_i(\mathbf{H}_{m,\beta^{\text{te}}})/(T\mu_i(\mathbf{H}_{n_1,\beta^{\text{tr}}})) & \mu_i(\mathbf{H}_{n_1,\beta^{\text{tr}}}) \geq \frac{1}{\alpha T}; \\ T\alpha^2 \mu_i(\mathbf{H}_{n_1,\beta^{\text{tr}}})\mu_i(\mathbf{H}_{m,\beta^{\text{te}}}) & \mu_i(\mathbf{H}_{n_1,\beta^{\text{tr}}}) < \frac{1}{\alpha T}, \end{cases} \tag{21}
$$

we obtain

$$
\left( \frac{1}{T} \mathbf{1}_{\mu_i(\mathbf{H}_{n_1,\beta^{\text{tr}}}) \geq \frac{1}{\alpha T}} + T\alpha^2 \mu_i^2(\mathbf{H}_{n_1,\beta^{\text{tr}}}) \mathbf{1}_{\mu_i(\mathbf{H}_{n_1,\beta^{\text{tr}}}) < \frac{1}{\alpha T}} \right) \frac{\mu_i(\mathbf{H}_{m,\beta^{\text{te}}})}{\mu_i(\mathbf{H}_{n_1,\beta^{\text{tr}}})} = \Xi_i(\boldsymbol{\Sigma}, \alpha, T).
$$

Therefore,

$$
R(\overline{\boldsymbol{\omega}}_T, \beta^{\text{te}}) \leq \text{Bias} + \text{Var}
$$

where

$$
\text{Bias} = \frac{2}{\alpha^2 T} \sum_i \Xi_i \frac{\omega_i^2}{\mu_i(\mathbf{H}_{n_1,\beta^{\text{tr}}})}
$$

$$
\text{Var} = \frac{2}{(1 - \alpha c(\beta^{\text{tr}}, \boldsymbol{\Sigma})\operatorname{tr}(\boldsymbol{\Sigma}))} \left( \sum_i \Xi_i \right)
$$

$$
\times [ f(\beta^{\text{tr}}, n_2, \sigma, \boldsymbol{\Sigma}_{\boldsymbol{\theta}}, \boldsymbol{\Sigma}) + \underbrace{2c(\beta^{\text{tr}}, \boldsymbol{\Sigma}) \sum_i \left( \frac{\mathbf{1}_{\mu_i(\mathbf{H}_{n_1,\beta^{\text{tr}}}) \geq \frac{1}{\alpha T}}}{T\alpha \mu_i(\mathbf{H}_{n_1,\beta^{\text{tr}}})} + \mathbf{1}_{\mu_i(\mathbf{H}_{n_1,\beta^{\text{tr}}}) < \frac{1}{\alpha T}} \right) \lambda_i \omega_i^2}_{V_2} ].
$$

Note that the term $V_2$ is obtained by our analysis for $\mathcal{E}_{\text{bias}}$. However, it originates from the stochasticity of SGD, and hence we treat this term as the variance in our final results. $\qquad \square$

## C Analysis for Lower Bound (Theorem 2)

### C.1 Fourth Moment Lower Bound for Meta Nosie

Similarly to upper bound, we need some technical results for the fourth moment of meta data $\mathbf{B}$ and noise $\boldsymbol{\xi}$ to proceed the lower bound analysis.

**Lemma C.1.** *Suppose Assumption 1-3 hold. Given $|\beta^{tr}| < \frac{1}{\lambda_1}$, for any PSD matrix $\mathbf{A}$, we have*

$$
\mathbb{E}[\mathbf{B}^{\top}\mathbf{B}\mathbf{A}\mathbf{B}^{\top}\mathbf{B}] \succeq \mathbf{H}_{n_1,\beta^{tr}}\mathbf{A}\mathbf{H}_{n_1,\beta^{tr}} + \frac{b_1}{n_2} \operatorname{tr}(\mathbf{H}_{n_1,\beta^{tr}}\mathbf{A})\mathbf{H}_{n_1,\beta^{tr}} \tag{22}
$$

$$
\boldsymbol{\Pi} \succeq \frac{1}{n_2} g(\beta^{tr}, n_1, \sigma, \boldsymbol{\Sigma}_{\boldsymbol{\theta}}, \boldsymbol{\Sigma})\mathbf{H}_{n_1,\beta^{tr}} \tag{23}
$$

*where $g(\beta, n, \sigma, \boldsymbol{\Sigma}, \boldsymbol{\Sigma}_{\boldsymbol{\theta}}) := \sigma^2 + b_1 \operatorname{tr}(\boldsymbol{\Sigma}_{\boldsymbol{\theta}}\mathbf{H}_{n,\beta}) + \beta^2 \mathbf{1}_{\beta \leq 0} b_1 \operatorname{tr}(\boldsymbol{\Sigma}^2)/n$.*

*Proof.* With a slight abuse of notations, we write $\beta^{\text{tr}}$ as $\beta$, $\mathbf{X}^{\text{in}}$ as $\mathbf{X}$ in this proof. Note that $\mathbf{x} \in \mathbb{R}^d \sim \mathcal{P}_{\mathbf{x}}$ is independent of $\mathbf{X}^{\text{in}}$. We first derive

$$\mathbb{E}[\mathbf{B}^\top \mathbf{B} \mathbf{A} \mathbf{B}^\top \mathbf{B}]$$

$$= \frac{1}{n_2} \mathbb{E}\left[(\mathbf{I} - \frac{\beta}{n_1}\mathbf{X}^\top\mathbf{X})\mathbf{x}\mathbf{x}^\top(\mathbf{I} - \frac{\beta}{n_1}\mathbf{X}^\top\mathbf{X})\mathbf{A}(\mathbf{I} - \frac{\beta}{n_1}\mathbf{X}^\top\mathbf{X})\mathbf{x}\mathbf{x}^\top(\mathbf{I} - \frac{\beta}{n_1}\mathbf{X}^\top\mathbf{X})\right]$$

$$+ \frac{n_2 - 1}{n_2}\mathbb{E}\left[(\mathbf{I} - \frac{\beta}{n_2}\mathbf{X}^\top\mathbf{X})\mathbf{\Sigma}(\mathbf{I} - \frac{\beta}{n_1}\mathbf{X}^\top\mathbf{X})\mathbf{A}(\mathbf{I} - \frac{\beta}{n_1}\mathbf{X}^\top\mathbf{X})\mathbf{\Sigma}(\mathbf{I} - \frac{\beta}{n_1}\mathbf{X}^\top\mathbf{X})\right]$$

$$\overset{(a)}{\succeq} \frac{b_1}{n_2}\mathbb{E}\left[\text{tr}(\mathbf{A}(\mathbf{I} - \frac{\beta}{n}\mathbf{X}^\top\mathbf{X})\mathbf{\Sigma}(\mathbf{I} - \frac{\beta}{n}\mathbf{X}^\top\mathbf{X}))(\mathbf{I} - \frac{\beta}{n}\mathbf{X}^\top\mathbf{X})\mathbf{\Sigma}(\mathbf{I} - \frac{\beta}{n}\mathbf{X}^\top\mathbf{X})\right]$$

$$+ \mathbf{H}_{n_1,\beta}\mathbf{A}\mathbf{H}_{n_1,\beta}$$

$$\succeq \frac{b_1}{n_2}\text{tr}(\mathbf{H}_{n_1,\beta}\mathbf{A})\mathbf{H}_{n_1,\beta} + \mathbf{H}_{n_1,\beta}\mathbf{A}\mathbf{H}_{n_1,\beta}$$

where $(a)$ is implied by Assumption 1.

Recall that $\Pi$ takes the following form:

$$\Pi = \frac{\sigma^2}{n_2}\mathbf{H}_{n_1,\beta} + \mathbb{E}[\mathbf{B}^\top\mathbf{B}\mathbf{\Sigma}_{\boldsymbol{\theta}}\mathbf{B}^\top\mathbf{B}] + \sigma^2 \cdot \frac{\beta^2}{n_2 n_1^2}\mathbb{E}[\mathbf{B}^\top\mathbf{X}^{\text{out}}\mathbf{X}^\top\mathbf{X}\mathbf{X}^{\text{out}\top}\mathbf{B}].$$

The second term can be directly bounded by letting $\mathbf{A} = \mathbf{\Sigma}_{\boldsymbol{\theta}}$ in eq. (22), and we have:

$$\mathbb{E}[\mathbf{B}^\top\mathbf{B}\mathbf{\Sigma}_{\boldsymbol{\theta}}\mathbf{B}^\top\mathbf{B}] \succeq \frac{b_1}{n_2}\text{tr}(\mathbf{H}_{n_1,\beta}\mathbf{\Sigma}_{\boldsymbol{\theta}})\mathbf{H}_{n_1,\beta}.$$

For the third term:

$$\frac{1}{n_2}\mathbb{E}[\mathbf{B}^\top\mathbf{X}^{\text{out}}\mathbf{X}^\top\mathbf{X}\mathbf{X}^{\text{out}\top}\mathbf{B}]$$

$$= \frac{1}{n_2}\mathbb{E}\left[(\mathbf{I} - \frac{\beta}{n_1}\mathbf{X}^\top\mathbf{X})\mathbf{x}\mathbf{x}^\top\mathbf{X}^\top\mathbf{X}\mathbf{x}\mathbf{x}^\top(\mathbf{I} - \frac{\beta}{n_1}\mathbf{X}^\top\mathbf{X})\right]$$

$$+ \frac{n_2 - 1}{n_2}\mathbb{E}\left[(\mathbf{I} - \frac{\beta}{n_1}\mathbf{X}^\top\mathbf{X})\mathbf{\Sigma}\mathbf{X}^\top\mathbf{X}\mathbf{\Sigma}(\mathbf{I} - \frac{\beta}{n_1}\mathbf{X}^\top\mathbf{X})\right]$$

$$\succeq \frac{n_1 b_1 \text{tr}(\mathbf{\Sigma}^2)}{n_2}\mathbf{H}_{n_1,\beta}\mathbf{1}_{\beta \leq 0}$$

Putting these results together completes the proof. $\qquad\square$

## C.2 Bias-Variance Decomposition

For the lower bound analysis, we also decompose the excess risk into bias and variance terms.

**Lemma C.2** (Bias-variance decomposition, lower bound)**.** *Following the notations in eq. (12), the excess risk can be decomposed as follows:*

$$R(\overline{\boldsymbol{\omega}}_T, \beta^{te}) \geq \underline{\mathcal{E}}_{bias} + \underline{\mathcal{E}}_{var}$$

*where*

$$\underline{\mathcal{E}}_{bias} = \frac{1}{2T^2} \cdot \sum_{t=0}^{T-1}\sum_{k=t}^{T-1}\left\langle(\mathbf{I} - \alpha\mathbf{H}_{n_1,\beta^{tr}})^{k-t}\mathbf{H}_{m,\beta^{te}}, \mathbf{D}_t\right\rangle,$$

$$\underline{\mathcal{E}}_{var} = \frac{1}{2T^2} \cdot \sum_{t=0}^{T-1}\sum_{k=t}^{T-1}\left\langle(\mathbf{I} - \alpha\mathbf{H}_{n_1,\beta^{tr}})^{k-t}\mathbf{H}_{m,\beta^{te}}, \mathbf{V}_t\right\rangle.$$

*Proof.* The proof is similar to that for Lemma B.3, and the inequality sign is reversed since we only calculate the half of summation. In particular,

$$\mathbb{E}[\overline{\boldsymbol{\varrho}}_T^{\text{var}} \otimes \overline{\boldsymbol{\varrho}}_T^{\text{var}}] = \frac{1}{T^2}\sum_{1 \leq t < k \leq T-1}\mathbb{E}[\boldsymbol{\varrho}_t^{\text{var}} \otimes \boldsymbol{\varrho}_k^{\text{var}}] + \frac{1}{T^2}\sum_{1 \leq k < t \leq T-1}\mathbb{E}[\boldsymbol{\varrho}_t^{\text{var}} \otimes \boldsymbol{\varrho}_k^{\text{var}}]$$

$$\succeq \frac{1}{T^2}\sum_{1 \leq t < k \leq T-1}\mathbb{E}[\boldsymbol{\varrho}_t^{\text{var}} \otimes \boldsymbol{\varrho}_k^{\text{var}}].$$

For $t \leq k$, $\mathbb{E}[\boldsymbol{\varrho}_k^{\text{var}}|\boldsymbol{\varrho}_t^{\text{var}}] = (\mathbf{I} - \alpha\mathbf{H}_{n_1,\beta^{\text{tr}}})^{k-t}\boldsymbol{\varrho}_t^{\text{var}}$, since $\mathbb{E}[\mathbf{B}_t^\top\boldsymbol{\xi}_t|\boldsymbol{\varrho}_{t-1}] = \mathbf{0}$. From this

$$\mathbb{E}[\overline{\boldsymbol{\varrho}}_T^{\text{var}} \otimes \overline{\boldsymbol{\varrho}}_T^{\text{var}}] \succeq \frac{1}{T^2} \sum_{t=0}^{T-1} \sum_{k=t}^{T-1} \mathbf{V}_t(\mathbf{I} - \alpha\mathbf{H}_{n_1,\beta^{\text{tr}}})^{k-t}.$$

Plugging this into $\frac{1}{2}\langle\mathbf{H}_{m,\beta^{\text{te}}}, \mathbb{E}[\overline{\boldsymbol{\varrho}}_T^{\text{var}} \otimes \overline{\boldsymbol{\varrho}}_T^{\text{var}}]\rangle$, we obtain:

$$\frac{1}{2}\langle\mathbf{H}_{m,\beta^{\text{te}}}, \mathbb{E}[\overline{\boldsymbol{\varrho}}_T^{\text{var}} \otimes \overline{\boldsymbol{\varrho}}_T^{\text{var}}]\rangle$$

$$\geq \frac{1}{2T^2} \sum_{t=0}^{T-1} \sum_{k=t+1}^{T-1} \langle\mathbf{H}_{m,\beta^{\text{te}}}, \mathbf{V}_t(\mathbf{I} - \alpha\mathbf{H}_{n_1,\beta^{\text{tr}}})^{k-t}\rangle$$

$$= \frac{1}{2T^2} \sum_{t=0}^{T-1} \sum_{k=t}^{T-1} \langle(\mathbf{I} - \alpha\mathbf{H}_{n_1,\beta^{\text{tr}}})^{k-t}\mathbf{H}_{m,\beta^{\text{te}}}, \mathbf{V}_t\rangle$$

$$= \underline{\mathcal{E}}_{\text{var}}.$$

The proof is the same for the term $\underline{\mathcal{E}}_{\text{bias}}$. $\qquad\square$

## C.3  Bounding the Bias

We first bound the summation of $\mathbf{D}_t$, i.e. $\mathbf{S}_k = \sum_{t=0}^{k-1}\mathbf{D}_t$.

**Lemma C.3** (Bounding $\mathbf{S}_t$). *If the stepsize satisfies $\alpha < 1/(2\max_i\{\mu_i(\mathbf{H}_{n_1,\beta^{\text{tr}}})\})$, then for any $k \geq 2$, it holds that*

$$\mathbf{S}_k \succeq \frac{b_1}{4n_2}\,\text{tr}\left(\left(\mathbf{I} - (\mathbf{I} - \alpha\mathbf{H}_{n_1,\beta^{\text{tr}}})^{k/2}\right)\mathbf{D}_0\right) \cdot \left(\mathbf{I} - (\mathbf{I} - \alpha\mathbf{H}_{n_1,\beta^{\text{tr}}})^{k/2}\right)$$

$$+ \sum_{t=0}^{k-1}(\mathbf{I} - \alpha\mathbf{H}_{n_1,\beta^{\text{tr}}})^t \cdot \mathbf{D}_0 \cdot (\mathbf{I} - \alpha\mathbf{H}_{n_1,\beta^{\text{tr}}})^t.$$

*Proof.* By eq. (18), since $\widetilde{\mathcal{M}} - \mathcal{M}$ is a PSD mapping, we have

$$\mathbf{S}_k = \mathbf{D}_0 + (\mathcal{I} - \alpha\widetilde{\mathcal{T}}) \circ \mathbf{S}_{k-1} + \alpha^2(\mathcal{M} - \widetilde{\mathcal{M}}) \circ \mathbf{S}_{k-1} \qquad (24)$$

$$\succeq \sum_{t=0}^{k-1}(\mathcal{I} - \alpha\widetilde{\mathcal{T}})^t \circ \mathbf{D}_0$$

$$= \sum_{t=0}^{k-1}(\mathbf{I} - \alpha\mathbf{H}_{n_1,\beta^{\text{tr}}})^t \cdot \mathbf{D}_0 \cdot (\mathbf{I} - \alpha\mathbf{H}_{n_1,\beta^{\text{tr}}})^t.$$

Note that for PSD $\mathbf{A}$,

$$(\mathcal{M} - \widetilde{\mathcal{M}}) \circ \mathbf{A} = \mathbb{E}[\mathbf{B}^\top\mathbf{B}\mathbf{A}\mathbf{B}^\top\mathbf{B}] - \mathbf{H}_{n_1,\beta^{\text{tr}}}\mathbf{A}\mathbf{H}_{n_1,\beta^{\text{tr}}}$$

By Lemma C.1, we have

$$(\mathcal{M} - \widetilde{\mathcal{M}}) \circ \mathbf{S}_k \succeq \frac{b_1}{n_2}\,\text{tr}\left(\mathbf{H}_{n_1,\beta^{\text{tr}}}\mathbf{S}_k\right)\mathbf{H}_{n_1,\beta^{\text{tr}}}$$

$$\succeq \frac{b_1}{n_2}\,\text{tr}\left(\sum_{t=0}^{k-1}(\mathbf{I} - \alpha\mathbf{H}_{n_1,\beta^{\text{tr}}})^{2t}\mathbf{H}_{n_1,\beta^{\text{tr}}} \cdot \mathbf{D}_0\right)\mathbf{H}_{n_1,\beta^{\text{tr}}}$$

$$\succeq \frac{b_1}{n_2}\,\text{tr}\left(\sum_{t=0}^{k-1}(\mathbf{I} - 2\alpha\mathbf{H}_{n_1,\beta^{\text{tr}}})^t\mathbf{H}_{n_1,\beta^{\text{tr}}} \cdot \mathbf{D}_0\right)\mathbf{H}_{n_1,\beta^{\text{tr}}}$$

$$\succeq \frac{b_1}{2n_2\alpha}\,\text{tr}\left(\left(\mathbf{I} - (\mathbf{I} - \alpha\mathbf{H}_{n_1,\beta^{\text{tr}}})^k\right)\mathbf{D}_0\right)\mathbf{H}_{n_1,\beta^{\text{tr}}}. \qquad (25)$$

Substituting eq. (25) back into eq. (24), and solving the recursion, we obtain

$$\mathbf{S}_k \succeq \sum_{t=0}^{k-1}(\mathcal{I} - \alpha\widetilde{\mathcal{T}})^t \circ \left\{ \frac{b_1\alpha}{2n_2}\operatorname{tr}\left(\left(\mathbf{I} - (\mathbf{I} - \alpha\mathbf{H}_{n_1,\beta^{\mathrm{tr}}})^{k-1-t}\right)\mathbf{D}_0\right)\mathbf{H} + \mathbf{D}_0 \right\}$$

$$= \frac{b_1\alpha}{2n_2}\underbrace{\sum_{t=0}^{k-1}\operatorname{tr}\left(\left(\mathbf{I} - (\mathbf{I} - \alpha\mathbf{H}_{n_1,\beta^{\mathrm{tr}}})^{k-1-t}\right)\mathbf{D}_0\right)\cdot(\mathbf{I} - \alpha\mathbf{H}_{n_1,\beta^{\mathrm{tr}}})^{2t}\mathbf{H}_{n_1,\beta^{\mathrm{tr}}}}_{\mathbf{J}_4}$$

$$+ \sum_{t=0}^{k-1}(\mathbf{I} - \alpha\mathbf{H}_{n_1,\beta^{\mathrm{tr}}})^t\cdot\mathbf{D}_0\cdot(\mathbf{I} - \alpha\mathbf{H}_{n_1,\beta^{\mathrm{tr}}})^t.$$

The term $\mathbf{J}_4$ can be further bounded by the following:

$$\mathbf{J}_4 \succeq \sum_{t=0}^{k-1}\operatorname{tr}\left(\left(\mathbf{I} - (\mathbf{I} - \alpha\mathbf{H}_{n_1,\beta^{\mathrm{tr}}})^{k-1-t}\right)\mathbf{D}_0\right)\cdot(\mathbf{I} - 2\alpha\mathbf{H}_{n_1,\beta^{\mathrm{tr}}})^t\mathbf{H}_{n_1,\beta^{\mathrm{tr}}}$$

$$\succeq \operatorname{tr}\left(\left(\mathbf{I} - (\mathbf{I} - \alpha\mathbf{H}_{n_1,\beta^{\mathrm{tr}}})^{k/2}\right)\mathbf{D}_0\right)\cdot\sum_{t=0}^{k/2-1}(\mathbf{I} - 2\alpha\mathbf{H}_{n_1,\beta^{\mathrm{tr}}})^t\mathbf{H}_{n_1,\beta^{\mathrm{tr}}}$$

$$\succeq \frac{1}{2\alpha}\operatorname{tr}\left(\left(\mathbf{I} - (\mathbf{I} - \alpha\mathbf{H}_{n_1,\beta^{\mathrm{tr}}})^{k/2}\right)\mathbf{D}_0\right)\cdot\left(\mathbf{I} - (\mathbf{I} - \alpha\mathbf{H}_{n_1,\beta^{\mathrm{tr}}})^{k/2}\right)$$

which completes the proof. $\qquad\square$

Then we can bound the bias term.

**Lemma C.4** (Bounding the bias). *Let $\omega_i = \langle\boldsymbol{\omega}_0 - \boldsymbol{\theta}^*, \mathbf{v}_i\rangle$. If $\alpha < \frac{1}{c(\beta^{\mathrm{tr}},\boldsymbol{\Sigma})\operatorname{tr}(\boldsymbol{\Sigma})}$, for sufficiently large $n_1$, s.t. $\mu_i(\mathbf{H}_{n_1,\beta^{\mathrm{tr}}}) > 0$, $\forall i$, then we have*

$$\underline{\mathcal{E}_{bias}} \geq \frac{1}{100\alpha^2 T}\sum_i \Xi_i \frac{\omega_i^2}{\mu_i(\mathbf{H}_{n_1,\beta^{\mathrm{tr}}})} + \frac{b_1}{1000n_2(1 - \alpha c(\beta^{\mathrm{tr}},\boldsymbol{\Sigma})\operatorname{tr}(\boldsymbol{\Sigma}))}\sum_i \Xi_i$$

$$\times \sum_i \left(\frac{\mathbf{1}_{\mu_i(\mathbf{H}_{n_1,\beta^{\mathrm{tr}}})\geq\frac{1}{\alpha T}}}{T\alpha\mu_i(\mathbf{H}_{n_1,\beta^{\mathrm{tr}}})} + \mathbf{1}_{\mu_i(\mathbf{H}_{n_1,\beta^{\mathrm{tr}}})<\frac{1}{\alpha T}}\right)\lambda_i\omega_i^2.$$

*Proof.* From Lemma C.2, we have

$$\underline{\mathcal{E}_{bias}} = \frac{1}{2T^2}\cdot\sum_{t=0}^{T-1}\sum_{k=t}^{T-1}\left\langle(\mathbf{I} - \alpha\mathbf{H}_{n_1,\beta^{\mathrm{tr}}})^{k-t}\mathbf{H}_{m,\beta^{\mathrm{te}}}, \mathbf{D}_t\right\rangle$$

$$= \frac{1}{2\alpha T^2}\cdot\sum_{t=0}^{T-1}\left\langle\left(\mathbf{I} - (\mathbf{I} - \alpha\mathbf{H}_{n_1,\beta^{\mathrm{tr}}})^{T-t}\right)\mathbf{H}_{n_1,\beta^{\mathrm{tr}}}^{-1}\mathbf{H}_{m,\beta^{\mathrm{te}}}, \mathbf{D}_t\right\rangle$$

$$\geq \frac{1}{2\alpha T^2}\left\langle\left(\mathbf{I} - (\mathbf{I} - \alpha\mathbf{H}_{n_1,\beta^{\mathrm{tr}}})^{T/2}\right)\mathbf{H}_{n_1,\beta^{\mathrm{tr}}}^{-1}\mathbf{H}_{m,\beta^{\mathrm{te}}}, \sum_{t=0}^{T/2}\mathbf{D}_t\right\rangle$$

$$\geq \frac{1}{2\alpha T^2}\left\langle\left(\mathbf{I} - (\mathbf{I} - \alpha\mathbf{H}_{n_1,\beta^{\mathrm{tr}}})^{T/2}\right)\mathbf{H}_{n_1,\beta^{\mathrm{tr}}}^{-1}\mathbf{H}_{m,\beta^{\mathrm{te}}}, \mathbf{S}_{\frac{T}{2}}\right\rangle.$$

Applying Lemma C.3 to $\mathbf{S}_{\frac{T}{2}}$, we obtain:

$$\underline{\mathcal{E}_{bias}} \geq \underline{d_1} + \underline{d_2}$$

where

$$\underline{d_1} = \frac{b_1}{8\alpha n_2 T^2} \operatorname{tr}\left(\left(\mathbf{I} - (\mathbf{I} - \alpha\mathbf{H}_{n_1,\beta^{\mathrm{tr}}})^{T/4}\right)\mathbf{D}_0\right)$$
$$\times \left\langle \left(\mathbf{I} - (\mathbf{I} - \alpha\mathbf{H}_{n_1,\beta^{\mathrm{tr}}})^{T/2}\right)\mathbf{H}_{n_1,\beta^{\mathrm{tr}}}^{-1}\mathbf{H}_{m,\beta^{\mathrm{te}}}, \ \left(\mathbf{I} - (\mathbf{I} - \alpha\mathbf{H}_{n_1,\beta^{\mathrm{tr}}})^{T/4}\right)\right\rangle$$

$$\underline{d_2} = \frac{1}{2\alpha T^2}\left\langle \left(\mathbf{I} - (\mathbf{I} - \alpha\mathbf{H}_{n_1,\beta^{\mathrm{tr}}})^{T/2}\right)\mathbf{H}_{n_1,\beta^{\mathrm{tr}}}^{-1}\mathbf{H}_{m,\beta^{\mathrm{te}}}, \right.$$
$$\left. \sum_{t=0}^{T/2-1}(\mathbf{I} - \alpha\mathbf{H}_{n_1,\beta^{\mathrm{tr}}})^t \cdot \mathbf{D}_0 \cdot (\mathbf{I} - \alpha\mathbf{H}_{n_1,\beta^{\mathrm{tr}}})^t\right\rangle.$$

Moreover,

$$\underline{d_2} \geq \frac{1}{2\alpha T^2}\left\langle \left(\mathbf{I} - (\mathbf{I} - \alpha\mathbf{H}_{n_1,\beta^{\mathrm{tr}}})^{T/2}\right)\mathbf{H}_{n_1,\beta^{\mathrm{tr}}}^{-1}\mathbf{H}_{m,\beta^{\mathrm{te}}}, \ \sum_{t=0}^{T/2-1}(\mathbf{I} - 2\alpha\mathbf{H}_{n_1,\beta^{\mathrm{tr}}})^t\mathbf{D}_0\right\rangle;$$
$$\geq \frac{1}{4\alpha^2 T^2}\left\langle \left(\mathbf{I} - (\mathbf{I} - \alpha\mathbf{H}_{n_1,\beta^{\mathrm{tr}}})^{T/2}\right)^2\mathbf{H}_{n_1,\beta^{\mathrm{tr}}}^{-2}\mathbf{H}_{m,\beta^{\mathrm{te}}}, \mathbf{D}_0\right\rangle.$$

Using the diagonalizing technique similar to the proof for Lemma B.7, we have

$$\underline{d_1} \geq \frac{b_1}{8\alpha n_2 T^2}\left(\sum_i \left(1 - (1 - \alpha\mu_i(\mathbf{H}_{n_1,\beta^{\mathrm{tr}}}))^{T/4}\right)\omega_i^2\right) \tag{26}$$

$$\times \left(\sum_i \left(1 - (1 - \alpha\mu_i(\mathbf{H}_{n_1,\beta^{\mathrm{tr}}}))^{T/4}\right)^2 \frac{\mu_i(\mathbf{H}_{m,\beta^{\mathrm{te}}})}{\mu_i(\mathbf{H}_{n_1,\beta^{\mathrm{tr}}})}\right), \tag{27}$$

$$\underline{d_2} \geq \frac{1}{4\alpha^2 T^2}\sum_i \left(1 - (1 - \alpha\mu_i(\mathbf{H}_{n_1,\beta^{\mathrm{tr}}}))^{T/4}\right)^2 \frac{\mu_i(\mathbf{H}_{m,\beta^{\mathrm{te}}})}{\mu_i^2(\mathbf{H}_{n_1,\beta^{\mathrm{tr}}})}\omega_i^2. \tag{28}$$

We use the following fact to bound the polynomial term. For $h_1(x) = 1 - (1 - x)^{\frac{T}{4}}$, we have

$$h_1(x) \geq \begin{cases} \frac{1}{5} & x \geq 1/T \\ \frac{T}{5}x & x < 1/T \end{cases}$$

i.e., $1 - (1 - \alpha\mu_i(\mathbf{H}_{n_1,\beta^{\mathrm{tr}}}))^{T/4} \geq \left(\frac{1}{5}\mathbf{1}_{\alpha\mu_i(\mathbf{H}_{n_1,\beta^{\mathrm{tr}}})\geq\frac{1}{T}} + \frac{\alpha\mu_i(\mathbf{H}_{n_1,\beta^{\mathrm{tr}}})}{5}\mathbf{1}_{\alpha\mu_i(\mathbf{H}_{n_1,\beta^{\mathrm{tr}}})<\frac{1}{T}}\right)$. Substituting this back into eqs. (27) and (28), and using the definition of effective meta weight $\Xi_i$ complete the proof. □

## C.4 Bounding the Variance

We first bound the term $\mathbf{V}_t$.

**Lemma C.5** (Bounding $\mathbf{V}_t$). *If the stepsize satisfies* $\alpha < 1/(\max_i\{\mu_i(\mathbf{H}_{n_1,\beta^{tr}})\})$, *it holds that*

$$\mathbf{V}_t \succeq \frac{\alpha g(\beta^{tr}, n_1, \mathbf{\Sigma}, \mathbf{\Sigma}_{\boldsymbol{\theta}})}{2} \cdot \left(\mathbf{I} - (\mathbf{I} - \alpha\mathbf{H}_{n_1,\beta^{tr}})^{2t}\right).$$

*Proof.* With a slight abuse of notations, we write $g(\beta^{\mathrm{tr}}, n_1, \boldsymbol{\Sigma}, \boldsymbol{\Sigma_\theta})$ as $g$. By definition,

$$
\begin{aligned}
\mathbf{V}_t &= (\mathcal{I} - \alpha\mathcal{T}) \circ \mathbf{V}_{t-1} + \alpha^2\Pi \\
&= (\mathcal{I} - \alpha\widetilde{\mathcal{T}}) \circ \mathbf{V}_{t-1} + (\mathcal{M} - \widetilde{\mathcal{M}}) \circ \mathbf{V}_{t-1} + \alpha^2\Pi \\
&\overset{(a)}{\succeq} (\mathcal{I} - \alpha\widetilde{\mathcal{T}}) \circ \mathbf{V}_{t-1} + \alpha^2 g\mathbf{H}_{n_1,\beta^{\mathrm{tr}}} \\
&\overset{(b)}{=} \alpha^2 g \cdot \sum_{k=0}^{t-1}(\mathcal{I} - \alpha\widetilde{\mathcal{T}})^k \circ \mathbf{H}_{n_1,\beta^{\mathrm{tr}}} \\
&= \alpha^2 g \cdot \sum_{k=0}^{t-1}(\mathbf{I} - \alpha\mathbf{H}_{n_1,\beta^{\mathrm{tr}}})^k \mathbf{H}_{n_1,\beta^{\mathrm{tr}}}(\mathbf{I} - \alpha\mathbf{H}_{n_1,\beta^{\mathrm{tr}}})^k \quad \text{(by the definition of } \mathcal{I} - \alpha\widetilde{\mathcal{T}}) \\
&= \alpha g \cdot \left(\mathbf{I} - (\mathbf{I} - \alpha\mathbf{H}_{n_1,\beta^{\mathrm{tr}}})^{2t}\right) \cdot (2\mathbf{I} - \alpha\mathbf{H}_{n_1,\beta^{\mathrm{tr}}})^{-1} \\
&\overset{(c)}{\succeq} \frac{\alpha g}{2} \cdot \left(\mathbf{I} - (\mathbf{I} - \alpha\mathbf{H}_{n_1,\beta^{\mathrm{tr}}})^{2t}\right)
\end{aligned}
$$

where $(a)$ follows from the Lemma C.1, $(b)$ follows by solving the recursion and $(c)$ holds since we directly replace $(2\mathbf{I} - \alpha\mathbf{H}_{n_1,\beta^{\mathrm{tr}}})^{-1}$ by $(2\mathbf{I})^{-1}$. $\qquad\square$

**Lemma C.6** (Bounding the variance). *Let* $\omega_i = \langle\boldsymbol{\omega}_0 - \boldsymbol{\theta}^*, \mathbf{v}_i\rangle$. *If* $\alpha < \frac{1}{c(\beta^{\mathrm{tr}},\boldsymbol{\Sigma})\operatorname{tr}(\boldsymbol{\Sigma})}$, *for sufficiently large* $n_1$, *s.t.* $\mu_i(\mathbf{H}_{n_1,\beta^{\mathrm{tr}}}) > 0$, $\forall i$, *for* $T > 10$, *then we have*

$$
\underline{\mathcal{E}_{var}} \geq \frac{g(\beta^{\mathrm{tr}}, n_1, \boldsymbol{\Sigma}, \boldsymbol{\Sigma_\theta})}{100 n_2(1 - \alpha c(\beta^{\mathrm{tr}}, \boldsymbol{\Sigma})\operatorname{tr}(\boldsymbol{\Sigma}))}\sum_i \Xi_i.
$$

*Proof.* From Lemma C.2, we have

$$
\begin{aligned}
\underline{\mathcal{E}_{var}} &= \frac{1}{2T^2} \cdot \sum_{t=0}^{T-1}\sum_{k=t}^{T-1}\left\langle(\mathbf{I} - \alpha\mathbf{H}_{n_1,\beta^{\mathrm{tr}}})^{k-t}\mathbf{H}_{m,\beta^{\mathrm{te}}}, \mathbf{V}_t\right\rangle \\
&= \frac{1}{2\alpha T^2} \cdot \sum_{t=0}^{T-1}\left\langle\left(\mathbf{I} - (\mathbf{I} - \alpha\mathbf{H}_{n_1,\beta^{\mathrm{tr}}})^{T-t}\right)\mathbf{H}_{n_1,\beta^{\mathrm{tr}}}^{-1}\mathbf{H}_{m,\beta^{\mathrm{te}}}, \mathbf{V}_t\right\rangle.
\end{aligned}
$$

Then applying Lemma C.5, and writting $g(\beta^{\mathrm{tr}}, n_1, \boldsymbol{\Sigma}, \boldsymbol{\Sigma_\theta})$ as $g$, we obtain

$$
\begin{aligned}
\underline{\mathcal{E}_{var}} &\geq \frac{g}{4T^2} \cdot \sum_{t=0}^{T-1}\left\langle\left(\mathbf{I} - (\mathbf{I} - \alpha\mathbf{H}_{n_1,\beta^{\mathrm{tr}}})^{T-t}\right)\mathbf{H}_{n_1,\beta^{\mathrm{tr}}}^{-1}\mathbf{H}_{m,\beta^{\mathrm{te}}}, \left(\mathbf{I} - (\mathbf{I} - \alpha\mathbf{H}_{n_1,\beta^{\mathrm{tr}}})^{2t}\right)\right\rangle \\
&= \frac{g}{4T^2}\sum_i\frac{\mu_i(\mathbf{H}_{m,\beta^{\mathrm{te}}})}{\mu_i(\mathbf{H}_{n_1,\beta^{\mathrm{tr}}})}\sum_{t=0}^{T-1}(1 - (1 - \alpha\mu_i(\mathbf{H}_{n_1,\beta^{\mathrm{tr}}})^{T-t}))(1 - (1 - \alpha\mu_i(\mathbf{H}_{n_1,\beta^{\mathrm{tr}}})^{2t})) \\
&\geq \frac{g}{4T^2}\sum_i\frac{\mu_i(\mathbf{H}_{m,\beta^{\mathrm{te}}})}{\mu_i(\mathbf{H}_{n_1,\beta^{\mathrm{tr}}})}\sum_{t=0}^{T-1}(1 - (1 - \alpha\mu_i(\mathbf{H}_{n_1,\beta^{\mathrm{tr}}})^{T-t-1}))(1 - (1 - \alpha\mu_i(\mathbf{H}_{n_1,\beta^{\mathrm{tr}}})^{t})) \quad (29)
\end{aligned}
$$

where the equality holds by applying the diagonalizing technique again. Following the trick similar to that in [22] to lower bound the function $h_2(x) := \sum_{t=0}^{T-1}\left(1 - (1-x)^{T-t-1}\right)\left(1 - (1-x)^t\right)$ defined on $x \in (0, 1)$, for $T > 10$, we have

$$
f(x) \geq \begin{cases} \frac{T}{20}, & \frac{1}{T} \leq x < 1 \\ \frac{3T^3}{50}x^2, & 0 < x < \frac{1}{T} \end{cases}
$$

Substituting this back into eq. (29), and using the definition of effective meta weight $\Xi_i$ completes the proof. $\qquad\square$

## C.5  Proof of Theorem 2

**Theorem C.1** (Theorem 2 Restated). *Let $\omega_i = \langle \boldsymbol{\omega}_0 - \boldsymbol{\theta}^*, \mathbf{v}_i \rangle$. If $|\beta^{tr}|, |\beta^{te}| < 1/\lambda_1$, $n_1$ is large ensuring that $\mu_i(\mathbf{H}_{n_1,\beta^{tr}}) > 0$, $\forall i$ and $\alpha < 1/\left(c(\beta^{tr}, \boldsymbol{\Sigma}) \operatorname{tr}(\boldsymbol{\Sigma})\right)$. For $T > 10$, the meta excess risk $R(\overline{\boldsymbol{\omega}}_T, \beta^{te})$ is bounded below as follows*

$$R(\overline{\boldsymbol{\omega}}_T, \beta^{te}) \geq \frac{1}{100\alpha^2 T} \sum_i \Xi_i \frac{\omega_i^2}{\mu_i(\mathbf{H}_{n_1,\beta^{tr}})} + \frac{1}{n_2} \cdot \frac{1}{(1 - \alpha c(\beta^{tr}, \boldsymbol{\Sigma}) \operatorname{tr}(\boldsymbol{\Sigma}))} \sum_i \Xi_i$$

$$\times [\frac{1}{100} g(\beta^{tr}, n_1, \boldsymbol{\Sigma}, \boldsymbol{\Sigma_\theta}) + \frac{b_1}{1000} \sum_i \Big( \frac{\mathbf{1}_{\mu_i(\mathbf{H}_{n_1,\beta^{tr}}) \geq \frac{1}{\alpha T}}}{T\alpha\mu_i(\mathbf{H}_{n_1,\beta^{tr}})} + \mathbf{1}_{\mu_i(\mathbf{H}_{n_1,\beta^{tr}}) < \frac{1}{\alpha T}} \Big) \lambda_i \omega_i^2].$$

*Proof.* The proof can be completed by combining Lemmas C.4 and C.6. $\qquad\square$

# D   Proofs for Section 4.2

## D.1   Proof of Lemma 1

*Proof of Lemma 1.* For the single task setting, we first simplify our notations in Theorem B.3 as follows.

$$c(0, \boldsymbol{\Sigma}) = c_1, \quad f(0, n_2, \sigma, \boldsymbol{\Sigma}, \mathbf{0}) = \sigma^2/n_2, \quad \mathbf{H}_{n_1,\beta^{tr}} = \boldsymbol{\Sigma}.$$

By Theorem B.3, we have

$$\text{Bias} = \frac{2}{\alpha^2 T} \sum_i \left( \frac{1}{T} \mathbf{1}_{\lambda_i \geq \frac{1}{\alpha T}} + T\alpha^2 \lambda_i^2 \mathbf{1}_{\lambda_i < \frac{1}{\alpha T}} \right) \frac{\omega_i^2 \mu_i(\mathbf{H}_{m,\beta^{te}})}{\lambda_i^2}$$

$$\leq \frac{2}{\alpha^2 T} \sum_i (\alpha\lambda_i \mathbf{1}_{\lambda_i \geq \frac{1}{\alpha T}} + \alpha\lambda_i \mathbf{1}_{\lambda_i < \frac{1}{\alpha T}}) \frac{\omega_i^2 \mu_i(\mathbf{H}_{m,\beta^{te}})}{\lambda_i^2}.$$

For large $m$, we have $\mu_i(\mathbf{H}_{m,\beta^{te}}) = (1 - \beta^{te}\lambda_i)^2 \lambda_i + o(1)$. Therefore,

$$\text{Bias} \leq \frac{2(1 - \beta^{te}\lambda_d)^2}{\alpha^2 T} \sum_i \omega_i^2 \leq \mathcal{O}(\frac{1}{T}).$$

For the variance term,

$$\text{Var} = \frac{2}{(1 - \alpha c_1 \operatorname{tr}(\boldsymbol{\Sigma}))} \underbrace{\sum_i \left( \frac{1}{T} \mathbf{1}_{\lambda_i \geq \frac{1}{\alpha T}} + T\alpha^2 \lambda_i^2 \mathbf{1}_{\lambda_i < \frac{1}{\alpha T}} \right) \frac{\mu_i(\mathbf{H}_{m,\beta^{te}})}{\lambda_i}}_{J_5}$$

$$\times [\frac{\sigma^2}{n_2} + 2c_1 \sum_i \left( \frac{\mathbf{1}_{\lambda_i \geq \frac{1}{\alpha T}}}{T\alpha\lambda_i} + \mathbf{1}_{\lambda_i < \frac{1}{\alpha T}} \right) \lambda_i \omega_i^2].$$

It is easy to check that

$$\sum_i \left( \frac{\mathbf{1}_{\lambda_i \geq \frac{1}{\alpha T}}}{T\alpha\lambda_i} + \mathbf{1}_{\lambda_i < \frac{1}{\alpha T}} \right) \lambda_i \omega_i^2 \leq \sum_i \left( \frac{\mathbf{1}_{\lambda_i \geq \frac{1}{\alpha T}}}{T\alpha} + \frac{1}{\alpha T} \mathbf{1}_{\lambda_i < \frac{1}{\alpha T}} \right) \omega_i^2 \leq \mathcal{O}(1/T).$$

Moreover,

$$J_5 \leq (1 - \beta^{te}\lambda_d)^2 \sum_i \left( \frac{1}{T} \mathbf{1}_{\lambda_i \geq \frac{1}{\alpha T}} + T\alpha^2 \lambda_i^2 \mathbf{1}_{\lambda_i < \frac{1}{\alpha T}} \right).$$

The term $\sum_i \left( \frac{1}{T} \mathbf{1}_{\lambda_i \geq \frac{1}{\alpha T}} + T\alpha^2 \lambda_i^2 \mathbf{1}_{\lambda_i < \frac{1}{\alpha T}} \right)$ has the form similar to Corollary 2.3 in [22] and we directly have $J_5 = \mathcal{O}\left(\log^{-p}(T)\right)$, which implies

$$\text{Var} = \mathcal{O}\left(\log^{-p}(T)\right).$$

Thus we complete the proof. $\qquad\square$

## D.2 Proof of Proposition 2

*Proof of Proposition 2.* We first consider the bias term in Theorems B.3 and C.1 (up to absolute constants):

$$\text{Bias} = \frac{2}{\alpha^2 T} \sum_i \left( \frac{1}{T} \mathbf{1}_{\mu_i(\mathbf{H}_{n_1,\beta^{\text{tr}}}) \geq \frac{1}{\alpha T}} + \alpha^2 T \mu_i^2(\mathbf{H}_{n_1,\beta^{\text{tr}}}) \mathbf{1}_{\mu_i(\mathbf{H}_{n_1,\beta^{\text{tr}}}) < \frac{1}{\alpha T}} \right) \frac{\omega_i^2 \mu_i(\mathbf{H}_{m,\beta^{\text{te}}})}{\mu_i(\mathbf{H}_{n_1,\beta^{\text{tr}}})^2}.$$

If $\mu_i(\mathbf{H}_{n_1,\beta^{\text{tr}}}) \geq \frac{1}{\alpha T}$, $\frac{1}{T} \leq \alpha \mu_i(\mathbf{H}_{n_1,\beta^{\text{tr}}})$; and if $\mu_i(\mathbf{H}_{n_1,\beta^{\text{tr}}}) < \frac{1}{\alpha T}$, then $\alpha^2 T \mu_i^2(\mathbf{H}_{n_1,\beta^{\text{tr}}}) < \alpha \mu_i(\mathbf{H}_{n_1,\beta^{\text{tr}}})$. Hence

$$\text{Bias} \leq \frac{1}{\alpha^2 T} \sum_i \frac{\omega_i^2 \mu_i(\mathbf{H}_{m,\beta^{\text{te}}})}{\mu_i(\mathbf{H}_{n_1,\beta^{\text{tr}}})} \leq \frac{2}{\alpha^2 T} \cdot \max_i \frac{\mu_i(\mathbf{H}_{m,\beta^{\text{te}}})}{\mu_i(\mathbf{H}_{n_1,\beta^{\text{tr}}})} \|\boldsymbol{\omega}_0 - \boldsymbol{\theta}^*\|^2 = \mathcal{O}(\frac{1}{T}).$$

Moreover,

$$\sum_i \left( \frac{\mathbf{1}_{\mu_i(\mathbf{H}_{n_1,\beta^{\text{tr}}}) \geq \frac{1}{\alpha T}}}{T \alpha \mu_i(\mathbf{H}_{n_1,\beta^{\text{tr}}})} + \mathbf{1}_{\mu_i(\mathbf{H}_{n_1,\beta^{\text{tr}}}) < \frac{1}{\alpha T}} \right) \lambda_i \omega_i^2$$

$$\overset{(a)}{\leq} \frac{1}{\alpha T} \sum_i \frac{\lambda_i}{\mu_i(\mathbf{H}_{n_1,\beta^{\text{tr}}})} \omega_i^2$$

$$\leq \frac{1}{\alpha T} \max_i \frac{\lambda_i}{\mu_i(\mathbf{H}_{n_1,\beta^{\text{tr}}})} \|\boldsymbol{\omega}_0 - \boldsymbol{\theta}^*\|^2 = \mathcal{O}(\frac{1}{T})$$

where $(a)$ holds since we directly upper bound $\mu_i(\mathbf{H}_{n_1,\beta^{\text{tr}}})$ by $\frac{1}{\alpha T}$ when $\mu_i(\mathbf{H}_{n_1,\beta^{\text{tr}}}) < \frac{1}{\alpha T}$. Therefore, it is essential to analyze $f(\beta^{\text{tr}}, n_2, \sigma, \boldsymbol{\Sigma}, \boldsymbol{\Sigma_\theta}) (\sum_i \Xi_i)$ and $g(\beta^{\text{tr}}, n_1, \sigma, \boldsymbol{\Sigma}, \boldsymbol{\Sigma_\theta}) (\sum_i \Xi_i)$ from variance term in the upper and lower bounds respectively.

Then we calculate some rates of interesting in Theorems B.3 and C.1 under the specific data and task distributions in Proposition 2.

If the spectrum of $\boldsymbol{\Sigma}$ satisfies $\lambda_k = k^{-1} \log^{-p}(k+1)$, then it is easily verified that $\text{tr}(\boldsymbol{\Sigma}^s) = O(1)$ for $s = 1, \cdots, 4$. By discussions on Assumption 3 in Appendix F, we have $C(\beta, \boldsymbol{\Sigma}) = \Theta(1)$ for given $\beta$. Hence,

$$c(\beta^{\text{tr}}, \boldsymbol{\Sigma}) = \Theta(1)$$
$$f(\beta^{\text{tr}}, n_2, \sigma, \boldsymbol{\Sigma}, \boldsymbol{\Sigma_\theta}) = c(\beta^{\text{tr}}, \boldsymbol{\Sigma}) \text{tr}(\boldsymbol{\Sigma_\theta} \boldsymbol{\Sigma}) + \Theta(1)$$
$$g(\beta^{\text{tr}}, n_1, \sigma, \boldsymbol{\Sigma}, \boldsymbol{\Sigma_\theta}) = b_1 \text{tr}(\boldsymbol{\Sigma_\theta} \mathbf{H}_{n_1,\beta^{\text{tr}}}) + \Theta(1).$$

If $r \geq 2p - 1$, then we have $g(\beta^{\text{tr}}, n_1, \sigma, \boldsymbol{\Sigma}, \boldsymbol{\Sigma_\theta}) \geq \Omega\left(\log^{r-p+1}(d)\right) \geq \Omega\left(\log^{r-p+1}(T)\right)$.

Let $k^\dagger := \text{card}\{i : \mu_i(\mathbf{H}_{n_1,\beta^{\text{tr}}}) \geq 1/\alpha T\}$. For large $n_1$, we have $\mu_i(\mathbf{H}_{n_1,\beta^{\text{tr}}}) = (1 - \beta^{\text{tr}} \lambda_i)^2 \lambda_i + o(1)$. If $k^\dagger = \mathcal{O}\left(T/\log^p(T+1)\right)$, then

$$\min_{1 \leq i \leq k^\dagger + 1} \mu_i(\mathbf{H}_{n_1,\beta^{\text{tr}}}) = \omega\left( \frac{\log^p(T)}{T[\log(T) - p\log(\log(T))]^p} \right) = \omega\left( \frac{1}{T} \right)$$

which contradicts the definition of $k^\dagger$. Hence $k^\dagger = \Omega\left(T/\log^p(T+1)\right)$. Then

$$\sum_i \Xi_i \geq \Omega\left( k^\dagger \cdot \frac{1}{T} \right) = \Omega\left( \frac{1}{\log^p(T)} \right).$$

Therefore, by Theorem C.1, $R(\overline{\boldsymbol{\omega}}, \beta^{\text{te}}) = \Omega\left(\log^{r-2p+1}(T)\right)$.

For $r < 2p - 1$, if $d = T^l$, where $l$ can be sufficiently large ($d \gg T$) but still finite, then

- If $p - 1 < r < 2p - 1$, $f(\beta^{\text{tr}}, n_2, \sigma, \boldsymbol{\Sigma}, \boldsymbol{\Sigma_\theta}) \leq \mathcal{O}(\log^{r-p+1} T)$;

- If $r \leq p - 1$, $f(\beta^{\text{tr}}, n_2, \sigma, \boldsymbol{\Sigma}, \boldsymbol{\Sigma_\theta}) \leq \mathcal{O}\left( \log(\log(T)) \right)$.

Following the analysis similar to that for Corollary 2.3 in [22], we have $\sum_i \Xi_i = \mathcal{O}(\frac{1}{\log^p(T)})$. Then by Theorem B.3

$$R(\overline{\boldsymbol{\omega}}_T, \beta^{\text{te}}) = \mathcal{O}\left(\frac{1}{\log^{p-(r-p+1)^+}(T)}\right).$$

$\square$

### D.3  Proof of Proposition 3

*Proof of Proposition 3.* Following the analysis in Appendix D.2, it is essential to analyze $f(\beta^{\text{tr}}, n_2, \sigma, \boldsymbol{\Sigma}, \boldsymbol{\Sigma_\theta})\left(\sum_i \Xi_i\right)$. If $d = T^l$, where $l$ can be sufficiently large but still finite, then

$$f(\beta^{\text{tr}}, n_2, \sigma, \boldsymbol{\Sigma}, \boldsymbol{\Sigma_\theta}) = \widetilde{\Theta}(1)$$

for $\lambda_k = k^q$ $(q > 1)$ or $\lambda_k = e^{-k}$.

Following the analysis similar to that for Corollary 2.3 in [22], we have

- If $\lambda_k = k^q$ $(q > 1)$, then $\sum_i \Xi_i = \mathcal{O}\left(\frac{1}{T^{\frac{q-1}{q}}}\right)$;

- If $\lambda_k = e^{-k}$, then $\sum_i \Xi_i = \mathcal{O}\left(\frac{\log(T)}{T}\right)$.

Substituting these results back into Theorem B.3, we obtain

- If $\lambda_k = k^q$ $(q > 1)$, then $R(\overline{\boldsymbol{\omega}}_T, \beta^{\text{te}}) = \widetilde{\mathcal{O}}\left(\frac{1}{T^{\frac{q-1}{q}}}\right)$;

- If $\lambda_k = e^{-k}$, then $R(\overline{\boldsymbol{\omega}}_T, \beta^{\text{te}}) = \widetilde{\mathcal{O}}\left(\frac{1}{T}\right)$.

$\square$

## E  Proofs for Section 4.3

### E.1  Proof of Proposition 4

*Proof of Proposition 4.* Following the analysis in Appendix D.2, it is crucial to analyze $f(\beta^{\text{tr}}, n_2, \sigma, \boldsymbol{\Sigma}, \boldsymbol{\Sigma_\theta})\left(\sum_i \Xi_i\right)$.

Then we calculate the rate of interest in Theorems B.3 and C.1 under some specific data and task distributions in Proposition 4. We have $\text{tr}(\boldsymbol{\Sigma}^2) = \frac{1}{s} + \frac{1}{d-s} = \Theta(\frac{\log^p(T)}{T})$. Moreover, by discussions on Assumption 3 in Appendix F, $C(\beta, \boldsymbol{\Sigma}) = \Theta(1)$. Hence

$$c(\beta, \boldsymbol{\Sigma}) := c_1 + \widetilde{\mathcal{O}}(\frac{1}{T});$$

$$f(\beta, n, \sigma, \boldsymbol{\Sigma}, \boldsymbol{\Sigma_\theta}) := 2c_1\mathcal{O}(1) + \frac{\sigma^2}{n} + \widetilde{\mathcal{O}}\left(\frac{1}{T}\right).$$

By the definition of $\Xi_i$, we have

$$\sum_i \Xi_i = \mathcal{O}\left(s \cdot \frac{\mu_1(\mathbf{H}_{m,\beta^{\text{te}}})}{T\mu_1(\mathbf{H}_{n_1,\beta^{\text{tr}}})} + \frac{1}{d-s} \cdot T\frac{\mu_d(\mathbf{H}_{n_1,\beta^{\text{tr}}})\mu_d(\mathbf{H}_{m,\beta^{\text{te}}})}{\lambda_d^2}\right)$$

$$= \mathcal{O}\left(\frac{1}{\log^p(T)}\right)\frac{\mu_1(\mathbf{H}_{m,\beta^{\text{te}}})}{\mu_1(\mathbf{H}_{n_1,\beta^{\text{tr}}})} + \mathcal{O}\left(\frac{1}{\log^q(T)}\right)\frac{\mu_d(\mathbf{H}_{n_1,\beta^{\text{tr}}})\mu_d(\mathbf{H}_{m,\beta^{\text{te}}})}{\lambda_d^2}$$

$$= \mathcal{O}\left(\frac{1}{\log^p(T)}\right)\frac{(1-\beta^{\text{te}}\lambda_1)^2}{(1-\beta^{\text{tr}}\lambda_1)^2} + \mathcal{O}\left(\frac{1}{\log^q(T)}\right)(1-\beta^{\text{te}}\lambda_d)^2(1-\beta^{\text{tr}}\lambda_d)^2$$

where the last equality follows from the fact that for large $n$, we have $\mu_i(\mathbf{H}_{n,\beta}) = (1-\beta\lambda_i)^2\lambda_i + o(1)$. Combining with the bias term which is $\mathcal{O}(\frac{1}{T})$, and applying Theorem B.3 completes the proof. $\square$

## E.2   Proof of Corollary 1

*Proof of Corollary 1.* For $t \in (s, K]$, by Theorem C.1, one can verify that $t = \widetilde{\Theta}(K)$ for diminishing risk. Let $t = K \log^{-l}(K)$, where $p > l > 0$. Following the analysis in Appendix E.1, we have

$$R(\overline{\boldsymbol{\omega}}_t^{\beta^{\mathrm{tr}}}, \beta^{\mathrm{te}}) \lesssim \widetilde{\mathcal{O}}(\frac{1}{K}) + (2c_1\nu^2 + \frac{\sigma^2}{n_2}) \tag{30}$$

$$\times \left[ \mathcal{O}\Big(\frac{1}{\log^{p-l}(K)}\Big) \frac{(1 - \beta^{\mathrm{te}}\lambda_1)^2}{(1 - \beta^{\mathrm{tr}}\lambda_1)^2} + \mathcal{O}\Big(\frac{1}{\log^{p+l}(K)}\Big) \Big(1 - \beta^{\mathrm{tr}}\lambda_d\Big)^2 \Big(1 - \beta^{\mathrm{te}}\lambda_d\Big)^2 \right]. \tag{31}$$

To clearly illustrate the trade-off in the stopping time, we let $l = 0$ for convenience. If $R(\overline{\boldsymbol{\omega}}_t^{\beta^{\mathrm{tr}}}, \beta^{\mathrm{te}}) < \epsilon$, we have

$$t_\epsilon \leq \exp\left( \epsilon^{-\frac{1}{p}} \Big[ \frac{U_l}{(1 - \beta^{\mathrm{tr}}\lambda_1)^2} + U_t(1 - \beta^{\mathrm{tr}}\lambda_d)^2 \Big]^{\frac{1}{p}} \right)$$

where

$$U_l = \mathcal{O}\Big( (2c_1\nu^2 + \frac{\sigma^2}{n_2})(1 - \beta^{\mathrm{te}}\lambda_1)^2 \Big) \quad and \quad U_l = \mathcal{O}\Big( (2c_1\nu^2 + \frac{\sigma^2}{n_2})(1 - \beta^{\mathrm{te}}\lambda_d)^2 \Big).$$

The arguments are similar for the lower bound, and we can obtain:

$$L_l = \mathcal{O}\Big( (2\frac{b_1\nu^2}{n_2} + \frac{\sigma^2}{n_2})(1 - \beta^{\mathrm{te}}\lambda_1)^2 \Big) \quad and \quad L_l = \mathcal{O}\Big( (2\frac{b_1\nu^2}{n_2} + \frac{\sigma^2}{n_2})(1 - \beta^{\mathrm{te}}\lambda_d)^2 \Big).$$

$\square$

## F   Discussions on Assumptions

**Discussions on Assumption 2**   If $\mathcal{P}_{\mathbf{x}}$ is Gaussian distribution, then we have

$$\mathbf{F} = \mathbb{E}[\mathbf{x}\mathbf{x}^\top \boldsymbol{\Sigma} \mathbf{x}\mathbf{x}^\top] = 2\boldsymbol{\Sigma}^3 + \boldsymbol{\Sigma} \operatorname{tr}(\boldsymbol{\Sigma}^2).$$

This implies that $\mathbf{F}$ and $\boldsymbol{\Sigma}$ commute because $\boldsymbol{\Sigma}^3$ and $\boldsymbol{\Sigma}$ commute. Moreover, in this case

$$\frac{\beta^2}{n}(\mathbf{F} - \boldsymbol{\Sigma}^3) = \frac{\beta^2}{n}(\boldsymbol{\Sigma}^3 + \boldsymbol{\Sigma} \operatorname{tr}(\boldsymbol{\Sigma}^2)).$$

Therefore, if $n \gg \lambda_1(\lambda_1^2 + \operatorname{tr}(\boldsymbol{\Sigma}^2))$, then the eigen-space of $\mathbf{H}_{n,\beta}$ will be dominated by $(\mathbf{I} - \beta\boldsymbol{\Sigma})^2\boldsymbol{\Sigma}$.

**Discussions on Assumption 3**   Assumption 3 is an eighth moment condition for $\mathbf{x} := \boldsymbol{\Sigma}^{\frac{1}{2}}\mathbf{z}$, where $\mathbf{z}$ is a $\sigma_x$ sub-Gaussian vector. Given $\beta$, for sufficiently large $n$ s.t. $\mu_i(\mathbf{H}_{n,b}) > 0, \forall i$, and if $\operatorname{tr}(\boldsymbol{\Sigma}^k)$ are all $O(1)$ for $k = 1, \cdots, 4$, then by the quadratic form and the sub-Gaussian property, which has finite higher order moments, we can conclude that $C(\beta, \boldsymbol{\Sigma}) = \Theta(1)$.

The following lemma further shows that if $\mathcal{P}_{\mathbf{x}}$ is a Gaussian distribution, we can derive the analytical form for $C(\beta, \boldsymbol{\Sigma})$.

**Lemma F.1.** *Given* $|\beta| < \frac{1}{\lambda_1}$, *for sufficiently large* $n$ *s.t.* $\mu_i(\mathbf{H}_{n,b}) > 0, \forall i$, *and if* $\mathcal{P}_{\mathbf{x}}$ *is a Gaussian distribution, assuming* $\boldsymbol{\Sigma}$ *is diagonal, we have:*

$$C(\beta, \boldsymbol{\Sigma}) = 210(1 + \frac{\beta^4 \operatorname{tr}(\boldsymbol{\Sigma}^2)^2}{(1 - \beta\lambda_1)^4}).$$

*Proof.* Let $\mathbf{e}_i \in \mathbb{R}^d$ denote the vector that the $i$-th coordinate is 1, and all other coordinates equal 0. For $\mathbf{x} \sim \mathcal{P}_{\mathbf{x}}$, denote $\mathbf{x}\mathbf{x}^\top = [x_{ij}]_{1 \leq i,j \leq d}$. Then we have:

$$\mathbb{E}[\|\mathbf{e}_i^\top \mathbf{H}_{n,\beta}^{-\frac{1}{2}}(\mathbf{I} - \frac{\beta}{n}\mathbf{X}^\top\mathbf{X})\boldsymbol{\Sigma}(\mathbf{I} - \frac{\beta}{n}\mathbf{X}^\top\mathbf{X})\mathbf{H}_{n,\beta}^{-\frac{1}{2}}\mathbf{e}_i\|^2]$$

$$\leq \mathbb{E}[\|\mathbf{e}_i^\top \mathbf{H}_{n,\beta}^{-\frac{1}{2}}(\mathbf{I} - \beta\mathbf{x}\mathbf{x}^\top)\boldsymbol{\Sigma}(\mathbf{I} - \beta\mathbf{x}\mathbf{x}^\top)\mathbf{H}_{n,\beta}^{-\frac{1}{2}}\mathbf{e}_i\|^2]$$

$$= \mathbb{E}\left[ (\mathbf{e}_i^\top \mathbf{H}_{n,\beta}^{-1}\mathbf{e}_i)^2 \left( \sum_{j \neq i} \beta^2\lambda_j x_{ij}^2 + \lambda_i(1 - \beta x_{ii})^2 \right)^2 \right]$$

For Gaussian distributions, we have

$$
\mathbb{E}[x_{ij}^2 x_{ik}^2] = \begin{cases} 9\lambda_i^2 \lambda_j^2 & j = k \text{ and } \neq i \\ 105\lambda_i^4 & i = j = k \\ 3\lambda_i^2 \lambda_j \lambda_k & i \neq j \neq k \end{cases}
$$

We can further obtain:

$$
\mathbb{E}[\|\mathbf{e}_i^\top \mathbf{H}_{n,\beta}^{-\frac{1}{2}}(\mathbf{I} - \frac{\beta}{n}\mathbf{X}^\top\mathbf{X})\mathbf{\Sigma}(\mathbf{I} - \frac{\beta}{n}\mathbf{X}^\top\mathbf{X})\mathbf{H}_{n,\beta}^{-\frac{1}{2}}\mathbf{e}_i\|^2]
$$

$$
\leq 105(\mathbf{e}_i^\top \mathbf{H}_{n,\beta}^{-1}\mathbf{e}_i)^2 \left( \sum_{j\neq i} \beta^2 \lambda_j^2 + (1 - \beta\lambda_i)^2 \right)^2
$$

$$
\overset{(a)}{\leq} 210(\mathbf{e}_i^\top \mathbf{H}_{n,\beta}^{-1}\mathbf{e}_i)^2 [\beta^4 \operatorname{tr}(\mathbf{\Sigma}^2)^2 + (1 - \beta\lambda_i)^4]
$$

$$
\overset{(b)}{\leq} 210[(\mathbf{e}_i^\top \mathbf{H}_{n,\beta}^{-1}\mathbf{e}_i)^2 \beta^4 \operatorname{tr}(\mathbf{\Sigma}^2)^2 + 1]
$$

where $(a)$ follows from the Cauchy-Schwarz inequality, and $(b)$ follows the fact that $(\mathbf{e}_i^\top \mathbf{H}_{n,\beta}^{-1}\mathbf{e}_i)^2 = \frac{1}{[(1-\beta\lambda_i)\lambda_i^2 + \frac{\beta^2}{n}(\lambda_i^2 + \operatorname{tr}(\mathbf{\Sigma}^2)\lambda_i)]^2} \leq 1/(1 - \beta\lambda_i)^4$.

Therefore, for any unit $\mathbf{v} \in \mathbb{R}^d$, we have

$$
\mathbb{E}[\|\mathbf{v}^\top \mathbf{H}_{n,\beta}^{-\frac{1}{2}}(\mathbf{I} - \frac{\beta}{n}\mathbf{X}^\top\mathbf{X})\mathbf{\Sigma}(\mathbf{I} - \frac{\beta}{n}\mathbf{X}^\top\mathbf{X})\mathbf{H}_{n,\beta}^{-\frac{1}{2}}\mathbf{v}\|^2]
$$

$$
\leq \max_{\mathbf{v}} 210[(\mathbf{v}^\top \mathbf{H}_{n,\beta}^{-1}\mathbf{v})^2 \beta^4 \operatorname{tr}(\mathbf{\Sigma}^2)^2 + 1] \leq 210 \left( 1 + \frac{\beta^4 \operatorname{tr}(\mathbf{\Sigma}^2)^2}{(1 - \beta\lambda_1)^4} \right).
$$

$\square$

# G  Further Related Work

## G.1  Underparameterized Setting

Provable guarantees of meta-learning have been extensively studied in the underparameterized regime, i.e. the number of tasks or the data size is much larger than the data dimension. Here we highlight some existing related studies and discuss their differences from ours.

[4] provides the generalization error bounds for S/Q meta-learners. Their generalization error bound is $O(\beta\sqrt{n}) + O(\frac{M}{\sqrt{n}})$, where the $\beta$ is the uniform stability parameter, $n$ is the number of task, and $M$ is the uniform bound on the loss function. Note that with the square loss as considered in our setting, $M$ can be as large as the dimension $d$ of the input. For the overparameterized regime, where $d \gg n$, the bound becomes asymptotically large and not useful.

[9] shows the generalization guarantees of MAML on recurring and unseen tasks respectively, where the excess risk bound is roughly $\widetilde{O}(\frac{G^2}{T}) + O(\frac{G^2}{mn})$, depending on the total iterations $T$, the number of tasks $m$, the available data size for each task $n$, and the uniform bound for gradient norm $G$. Notice that the gradient norm typically scales polynomially with the input dimension $d$. Therefore, in the overparameterized regime $(d \gg mn)$, the bound again becomes vacuous.

[6] aims to theoretically characterize the performance between MAML and the standard Empirical Risk Minimization (ERM). Firstly, they show that the empirical training solutions will converge to their population-optimal values with concentration bounds, which have terms $O(\frac{\sqrt{d}}{\sqrt{n}})$ or $O(\frac{\sqrt{d}}{\sqrt{\tau}})$ where $n$ is the sample size and $\tau$ is the number of training episodes per task. Such bounds will be crude in the overparameterized regime $(d \gg n, \tau)$. Then from the generalization perspective, they only give excess risks for the population-optimal solutions, whereas we analyze the generalization property of empirical training solutions based on their optimization trajectory. Moreover, the comparison between the excess risk of MAML and ERM for the population-optimal solutions requires $m = \Omega(d)$, where $m$ is the number of task. Therefore, such comparison does not apply to the overparameterized regime $(d \gg m)$.

From what has been discussed above, all of these bounds, are useful (i.e., yield small or vanishing error) only for underparameterized regime, where sample size $n$ is much larger than the dimension $d$ of input (i.e., $n \gg d$), but become vacuous (i.e., yield large error bound) for overparameterized regime, where $n \ll d$. Thus, they fail to explain why the overparameterized neural network can still generalize well.

In contrast, our work provides a much more refined data-dependent upper bound by incorporating the data and task roles. Our bound can be written in a concise form as $O(\frac{h(\Sigma, \Sigma_\theta, \alpha, T)}{T})$ (see Theorem 1 in Section 4 for exact form of the bound), where the function $h(\cdot)$ is determined by data and task covariances $\Sigma$ and $\Sigma_\theta$, step-size $\alpha$ and the total number $T$ of SGD iterations (here $T$ scales linearly with the sample size). Note that the dimension $d$ is implicitly captured in $\Sigma, \Sigma_\theta$. In the overparameterized regime with $d \gg T$, the function $h(\cdot)$ explicitly captures the effect of data and task conditions to guarantee the value of $h(\cdot)$ to be small compared to $T$, and thus the excess risk can diminish under overparameterization. In Propositions 2 and 3, we further give specific examples about data and task covariances $\Sigma, \Sigma_\theta$ that yield small excess risk and good generalization.

Besides the above key difference, our paper also provides the following important results that were not studied in [4, 9, 6]. (a) We provide the lower bound to justify the tightness of our results in the overparameterized regime, whereas [4, 9, 6] do not have such a result. (b) Our results capture how the task diversity affects the excess risk in the overparameterized regime. In particular, we give an example (in Proposition 2), for which the excess risk exhibits a phase transition with respect to task diversity. Such an interesting behavior is not captured in [4, 9, 6].

### G.2 Overparameterized Setting

Despite the overparameterization is crucial to demystify the remarkable generalization ability of deep meta-learning [21, 13], there are only a few theoretical analysis being developed to study the generalization of MAML in the overparameterized regime. [20] studied the MAML with over-parameterized deep neural nets with a generalization gap quantifying the difference between the empirical and population loss functions at their optimal solutions. However, their bound is derived by conventional complexity-based techniques and does not consider the data or task-dependency similarly as [4, 9, 6] discussed in Appendix G.1, which tends to be weak in the high dimensional, especially in the overparameterized regime. Most related to our work are recent studies [2, 24], where they develop more precise generalization bounds for overparameterized setting under a mixed linear regression model. Yet, their empirical training solutions are directly calculated by taking the closed-form of training objective's minimum, which are not obtained by trained with SGD as ours. More crucially, they consider only the simple isotropic covariance for data and tasks, which are directly scaled by $d$, i.e. $\Sigma = \frac{1}{d}\mathbf{I}$. Thus, they do not explicitly capture how the generalization performance of MAML depends on the data and task distributions.

## H    Future Directions

This work takes a step towards understanding the benefits of overparameterization for MAML from the generalization aspect. There are many important future work directions, and we elaborate some interesting directions in this section.

### H.1    Generalizing to Other Learning Methods

Our result can directly extend to the random feature(RF) model, adopting the similar analysis for nonlinear model in [12] : the data $\mathbf{x}$ is generated by $\sigma(\mathbf{Wz})$, where $\mathbf{W}$ is the random feature matrix and $\sigma(\cdot)$ is the nonlinear (activation) function. Hence, the RF model can be regarded as training a two-layer neural network where the weights in the first layer are chosen randomly and then fixed and only the output layer is optimized.

Beyond the fixed feature approach, understanding the overparameterization in neural networks is much more challenging. Recently, [3] made important progress towards this aspect. They studied the benign overfitting phenomenon in training a two-layer convolutional neural network, and provided new analysis to tackle the neural network learning process. One possible direction is to generalize our analysis to neural networks by further advancing the techniques in [3].

## H.2 Meta-batch Setting

We can consider to incorporate a practical meta-batch setting in our framework, i.e. given a set of tasks in advance, and at each iteration, the task is sampled from the set with replacement uniformly at random and will be visited multiple times during optimization. This is broadly referred as multi-pass SGD [16], while we have studied the single-pass setting, that each task is used only once. For multi-pass SGD, the iterate analysis in Appendix B.3 will be more challenging since we have to consider the dependence on history that leads to a complicated calculation of the expected error matrix. To handle such complication, we may adopt some analysis techniques recently developed for multi-pass SGD [23, 17, 16], and further advance them to analyze meta-batch MAML in meta learning.

Intuitively, we expect in such a setting, meta-batch size (of tasks) will play an important role in determining the excess risk. Specifically, how meta-batch size compares with the effect of data covariance and task diversity (coupled with the number of SGD iterations) will determine whether benign fitting in the overparameterized regime can occur.

## H.3 Longer Inner Loops

Another important direction is to study the MAML beyond the one-step gradient update in the inner loop. Two possible cases are discussed as follows.

If the inner loop continually takes gradient updates towards the per-task loss function until converges (additionally regularized by the distance between task-specific parameter and the model parameter), then the algorithm is equivalent to another well-known meta method, iMAML [18]. Under the mixed linear regression model, similarly, we can reformulate the problem as a meta least square problem with modified meta inputs and output responses [7, 1]. Therefore, our analysis can directly generalize to such a setting with modified meta least square problem.

If the inner loop only takes a few steps of gradient updates, the analysis will be more challenging, because the update of meta parameter in the outer loop will involve Hessian, and the analysis of such an algorithm will need to handle the complicated statistical correlation between the gradient and Hessian estimators [15], where new techniques are required. Recently, [5] proposed that one can treat the Hessian as identity operator in theoretical analysis, where the corresponding algorithm is called FO-MAML, and FO-MAML typically achieves a performance similar to MAML in practice [10]. Such relaxation may make the problem more tractable. Then the remaining problem is to extend our technique for overparameterized MAML to FO-MAML, which is an interesting research direction for future study.