# OpenReview forum: "Provable Generalization of Overparameterized Meta-learning Trained with SGD"
_NeurIPS.cc/2022/Conference — NeurIPS 2022 Accept_

### Official Review · Reviewer_T2R8 · 2022-07-11

**Rating:** 7
**Confidence:** 2
**Soundness:** 3 good
**Presentation:** 3 good
**Contribution:** 3 good

**Summary:**

The paper introduces a generalization bound of MAML in more complex linear regression tasks in the over-parametrized regime. Their main theoretical results upper and lower bounds the excess risk while linking it to input data as well as task distribution, outer loop length and inner loop learning rate. Theoretical results are supported by experiments.

**Questions:**

See above.

1. How exactly does the theory generalise to the meta-batch setting.
2. I might have missed this but why do you have to use the iterative averaging instead of using the final w_T?
3. Can you comment on how to extend the theory for longer inner loops.


**Limitations:**

The authors do not explicitly discuss the limitations of their work, although I agree that this is not needed here.

**Strengths And Weaknesses:**

The paper is well written, relaxes common assumptions and therefore generalizes existing bounds. I am not familiar with the theoretical literature in meta-learning but I found the paper very nice to study, although I did not validate the correctness of the theoretical results thoroughly. The math in the paper is quite dense but I think the authors do a nice job presenting the results. Overall, I think the paper is of value to the (meta-learning) community at NeurIPS and I propose acceptance.

---

> ### Author Response · Authors · 2022-08-01
> **Response to Reviewer T2R8**
>
> We thank the reviewer for the positive comments and thoughtful questions. We next provide responses to your specific questions.
>
> **Q1**: How exactly does the theory generalise to the meta-batch setting.
>
> **A1**: We assume that by ``meta-batch", you meant given a set of tasks in advance. In such a setting, at each iteration, the task is sampled from the set with replacement uniformly at random and will be visited multiple times during optimization. This is broadly referred as multi-pass SGD, whereas we consider the single-pass setting in this paper, that each task is used only once. For multi-pass SGD, the iterate analysis in our paper (see Appendix B.3) will be more challenging since we have to consider the dependence on history which leads to a
> complicated calculation of the expected error matrix. To handle such complications, we may adopt some analysis techniques recently developed for multi-pass SGD [1][2][3], and further advance them to analyze meta-batch MAML in meta-learning.
>
> Intuitively, we expect in such a setting, meta-batch size (of tasks) will play an important role in determining the excess risk. Specifically, how meta-batch size compares with the effect of data covariance and task diversity (coupled with the number of SGD iterations) will determine whether benign fitting in overparameterized regime can occur.
>
> **Q2**: I might have missed this but why do you have to use the iterative averaging instead of using the final $w_T$?
>
> **A2**: Thanks for the great question! Under stochastic gradient descent (SGD) training with constant step size, the iterative averaging is commonly adopted [4,5] (see the reference list attached to the end of this message), to stabilize the output. It is also technically convenient to establish the convergence guarantee for the iterative averaging as SGD's output. Though technically more demanding, it is still possible to establish the convergence guarantee for the last iterative of SGD, which can be a very interesting topic to explore as future study.
>
> **Q3**: Can you comment on how to extend the theory for longer inner loops.
>
> **A3**: Thanks for the great question! We explain this in the following two cases.
>
> If the inner loop continually takes gradient updates towards the  per-task loss function until convergence (additionally regularized by the distance between task-specific parameter and the model parameter), then the algorithm is equivalent to another well-known meta method, iMAML [6]. Under the mixed linear regression model, similarly, we can reformulate the problem as a meta least-square problem with modified meta inputs and output responses [1][7]. Therefore, our analysis can directly generalize to such a setting with a modified meta least square problem.
>
> If the inner loop only takes a few steps of gradient updates, the analysis will be more challenging, because the update of meta parameter in the outer loop will involve Hessian, and the analysis of such an algorithm will need to handle the complicated statistical correlation between the gradient and Hessian estimators [8], where new techniques are required. Recently, [9] proposed that one can treat the Hessian as identity operator in theoretical analysis, where the corresponding algorithm is called FO-MAML, and FO-MAML typically achieves a performance similar to MAML in practice [10]. Such relaxation may make the problem more tractable. Then the remaining problem is to extend our technique for overparameterized MAML to FO-MAML, which is an interesting research direction for future study.
>
> References.
>
> [1] Pillaud-Vivien et al,  Statistical optimality of stochastic gradient
> descent on hard learning problems through multiple passes, NeurIPS 2018
>
> [2] M\"ucke et al, Beating SGD Saturation with Tail-Averaging and Minibatching, NeurIPS 2019.
>
> [3] Zou et al, Risk Bounds of Multi-Pass SGD for Least
> Squares in the Interpolation Regime, Arxiv 2022
>
> [4] Denevi et al, Learning-to-learn stochastic gradient descent with biased regularization, ICML 2019.
>
> [5] Denevi et al,  Learning to learn around a common mean, NeurIPS 2018.
>
> [6] Rajeswaran et al, Meta-Learning with Implicit Gradients, NeurIPS 2019.
>
> [7] Bai et al, How important is the train-validation split in meta-learning? ICML 2021.
>
> [8] Ji et al, Theoretical convergence of multi-step model-agnostic meta-learning, JMLR 2021.
>
> [9] Collins et al, MAML and ANIL Provably Learn Representations, ICML 2022.
>
> [10] Finn et al, Model-Agnostic Meta-Learning for Fast Adaptation of Deep Networks, ICML 2017.

---

> > ### Comment · Reviewer_T2R8 · 2022-08-08
> > **Thank you!**
> >
> > Dear authors,
> > I will keep my score as is and discuss with the other reviewers the novelty concern.
> > Thank you for your response!

---

### Official Review · Reviewer_uDLS · 2022-07-11

**Rating:** 7
**Confidence:** 2
**Soundness:** 4 excellent
**Presentation:** 4 excellent
**Contribution:** 3 good

**Summary:**

The authors study the overparameterization behavior in the Model-Agnostic Meta-Learning (MAML) framework. They find lower and upper bounds of the risk of MAML trained with SGD. They show that for a large number of meta-training iterations the excess risk vanishes. They also show the effects of early stopping and adaptation learning rate on excess risk.

**Questions:**

None

**Limitations:**

There are no limitations with regards to negative societal impacts.

**Strengths And Weaknesses:**

Strengths:

1. The paper is very well written and organized. Following it was straightforward.
2. The authors use some settings which correspond to real world applications (L46)
3. The authors derive explicit formulas for bias and variance which bound the excess risk above. Additionally, they derive a lower bound on the excess risk.
4. The leading and tail spectrum space analysis is very interesting and well developed.

Weaknesses:
1. Focusing on mixed linear regression is the right move to do when studying these theoretical properties. However, there is a drawback: people don’t really use linear regression for meta learning in practice. The authors are thus not doing anything wrong but this work is weak in practicality.

---

> ### Author Response · Authors · 2022-08-01
> **Response to Reviewer uDLS**
>
> We thank the reviewer for the valuable feedback and we appreciate the positive comments. We next provide responses to your specific questions.
>
> **Q**: Focusing on mixed linear regression is the right move to do when studying these theoretical properties. However, there is a drawback: people don’t really use linear regression for meta learning in practice. The authors are thus not doing anything wrong but this work is weak in practicality.
>
> **A**: Thanks for the comments! Based on the techniques developed in this paper, we plan to study a meta-supervised learning problem. We will consider a multi-task binary classification problem as a starting point, where all tasks can share some common features and have certain different data distribution components. The learning model can be neural networks. We expect that our current analysis needs to be further adapted and advanced to handle SGD forms of binary classification problems as well as neural networks.

---

### Official Review · Reviewer_9gwj · 2022-07-12

**Rating:** 4
**Confidence:** 4
**Soundness:** 3 good
**Presentation:** 3 good
**Contribution:** 2 fair

**Summary:**

This paper investigated the provable generalization performance of overparameterized MAML. In particular, the authors focus on the mixed linear regression model and a one-step meta-training process with stochastic gradient descent (SGD). Theoretically, they give both upper and lower bounds on the excess risk of meta-learning trained with SGD. Furthermore, they explore the affect of task/data distributions and adaptation learning rate.

Overall, this paper provides some theoretical evidences to understand the effects of MAML trained with SGD. The main concern to me is the novelty and significance of the theoretical results. Due to the fact that most of the results have been known, I think this paper is interesting but not so surprising.

**Questions:**

I hope the authors can clarify some of my concerns. But if I am wrong, please correct me.

**Ethics Review Area:**

["I don’t know"]

**Strengths And Weaknesses:**

**Strengths**
1.  This paper is clearly written and easy to follow.
2.  The proofs of the main results are complete.
3.  MAML is a popular meta-learning method and theoretical analysis on it is also important to the ML community.

**Weaknesses**
1. The main concern to me is the novelty of this paper. The analysis of generalization performance of MAML based on SGD has been investigated by Chen et al., 2020 and Fallah et al., 2021, who also give generalization bounds for MAML which show the convergency rate of $\mathcal{O}(\frac{1}{T})$ and $\mathcal{O}(\frac{1}{Tn})$, respectively. Besides, their results are more general than this work because the previous work do not relay on a specific case of mixed linear regression. I think the author should compare and discuss in details with the previous results on MAML trained with SGD. In terms of MAML for linear regression, Collins et al., 2020 have analyzed that the performance of MAML can outperform ERM, which is a much stronger result to me.

2. Another concern to me is the definition of task diversity rate $r$ in the analysis of the affect of task diversity. I can not found its formal definition in the main pages. I wonder how it can measure or represent the diversity the task distribution.

3. For the significance of this paper, I noticed that most of the results are intuitive or have been proven in the literature. I think a good theory paper should give new insights for people or can guide some practical algorithms for improvement.

4. Can the results be generalized to other machine learning methods beyond the linear regression, such as neural networks?

Chen et al., A Closer Look at the Training Strategy for Modern Meta-Learning, NeurIPS 2020
Fallah et al., Generalization of Model-Agnostic Meta-Learning Algorithms: Recurring and Unseen Tasks, NeurIPS 2021
Collins et al., Why Does MAML Outperform ERM? An Optimization Perspective, ArXiv 2020

---

> ### Author Response · Authors · 2022-08-01
> **Response to Reviewer 9gwj  Part 1/2**
>
> We thank the reviewer for the time and thoughtful comments on our work. We address the concerns you raise below.
>
> **Q1**: The main concern to me is the novelty of this paper. The analysis of generalization performance of MAML based on SGD has been investigated by Chen et al., 2020 and Fallah et al., 2021, who also give generalization bounds for MAML which show the convergency rate of $\mathcal{O}(\frac{1}{T})$ and $\mathcal{O}(\frac{1}{Tn})$, respectively. In terms of MAML for linear regression, Collins et al., 2020 have analyzed that the performance of MAML can outperform ERM, which is a much stronger result to me.
>
> **A1**：We thank the reviewer for pointing out these studies [1] (Chen et al. 2020), [2] (Fallah et al. 2021) and [3] (Collins et al. 2020) (see our reference list towards the end of this message). There are many detailed differences between our generalization error bound with those in [1,2,3], but we will focus on the key difference below. All bounds in [1,2,3] are useful (i.e., yield small or vanishing error) only for $\textbf{underparameterized}$ regime, where sample size $n$ is much larger than the dimension $d$ of input (i.e., $n \gg d$). But the bounds in [1,2,3] become vacuous and not useful (i.e., yield large error bound) for the  $\textbf{overparameterized}$ regime, where $n\ll d$. Thus, those bounds fail to explain why $\textbf{overparameterized}$ models can still generalize well. In contrast, $\textbf{we focus on the overparameterized regime}$ (i.e., $n\ll d$), and our bound explicitly captures how the bound can diminish with overparameterization and under what data and task conditions. Putting into a bigger context, understanding the generalization error in the overparameterized regime has recently been pioneered by seminal works [4][5][6][7], and arises rapidly as an active area in learning theory community. Our work contributes to this new line of literature and develops novel techniques to deal with the overparameterization in high dimensions for meta-learning, more specifically for MAML.
>
>   Below, we further elaborate the above key difference between [1,2,3] and our work.
>
> In [1], the generalization error bound is $O(\beta\sqrt{n})+O(\frac{M}{\sqrt{n}})$ (see Theorem 2 in [1]), where $\beta$ is the uniform stability parameter, $n$ is the number of task, and $M$ is the uniform bound on the loss function. Note that under the square loss as considered in our setting, $M$ can be as large as the dimension $d$ of the input. For the overparameterized regime, where $d\gg n $, the bound become asymptotically large and not useful.
>
> In [2], the excess risk bound is roughly $\tilde{O}(\frac{G^2}{T})+O(\frac{G^2}{mn})$ (see Proposition 2 in  [2]), where $T$ is the total iteration, $m$ is the number of tasks, $n$ is the data size for each task and $G$ is the uniform bound for gradient norm. Note that the gradient norm typically scales polynomially with the input dimension $d$. Hence, in the overparameterized regime $(d\gg mn)$, the bound again becomes vacuous.
>
> In [3], the concentration bounds (see Theorems 1 and 2) have terms $O(\frac{\sqrt{d}}{\sqrt{n}})$ or $O(\frac{\sqrt{d}}{\sqrt{\tau}})$ where  $n$  is the sample size and $\tau$ is the number of training episodes per task. Clearly, the bound will become vacuous in the overparameterized regime($d\gg n,\tau$).
>
> In contrast, our work provides a much more refined data-dependent upper bound by incorporating the data and task roles. Our bound can be written in a concise form as $O(\frac{h(\Sigma,\Sigma_{\theta},\alpha,T)}{T})$ (see Theorem 1 in Sec. 4.1 for exact form of the bound), where the function $h(\cdot)$ is determined by data and task covariances $\Sigma$ and $\Sigma_{\theta}$, step-size $\alpha$ and the total number $T$ of SGD iterations (here $T$ scales linearly with the sample size). Note that the dimension $d$ is implicitly captured in $\Sigma,\Sigma_{\theta}$. In the overparameterized regime with $d\gg T$, the function $h(\cdot)$ explicitly captures the effect of data and task conditions to guarantee the value of $h(\cdot)$ to be small compared to $T$, and thus the excess risk can diminish under overparameterization. In Propositions 2 and 3, we further give specific examples about data and task covariances $\Sigma,\Sigma_{\theta}$ that yield small excess risk and good generalization.
>
> Besides the above key difference, our paper also provides the following important results that were not studied in [1,2,3]. (a) We provide the lower bound to justify the tightness of our results in the overparameterized regime, whereas [1,2,3] do not have such a result. (b) Our results capture how the task diversity affects the excess risk in the overparameterized regime. In particular, we give an example (in Proposition 2), for which the excess risk exhibits a phase transition with respect to task diversity. Such an interesting behavior is not captured in [1,2,3].

---

> > ### Author Response · Authors · 2022-08-01
> > **Response to Reviewer 9gwj  Part 2/2**
> >
> > **Q2**: Another concern to me is the definition of task diversity rate $r$ in the analysis of the affect of task diversity. I can not found its formal definition in the main pages. I wonder how it can measure or represent the diversity the task distribution.
> >
> > **A2**: We thank the reviewer for pointing it out. $r$ is the spectrum parameter for the task covariance, where the eigenvalue of task covariance satisfies $\nu_{k}=\log^{r}(k+1)$ (see the second line of Proposition 2). Since a large $r$ implies large eigenvalues and high variations for task vectors, we adopt $r$ to measure the diversity of task distributions, and call $r$  the task diversity in the remaining parts of the paper. We have clarified this definition in our revision in the paragraph below Proposition 2 in Section 4.2.
> >
> > **Q3**: For the significance of this paper, I noticed that most of the results are intuitive or have been proven in the literature. I think a good theory paper should give new insights for people or can guide some practical algorithms for improvement.
> >
> > **A3**: As we explained in the response for Q1, previous generalization results for MAML fail to capture good generalization performance in the overparameterized regime. In contrast, our contribution lies in developing new analysis tools to provide data and task-dependent bound that can vanish in the overparameterized regime. As we commented before, exploring the overparameterized model in high dimensions is a new trend and active area in learning theory community, and a series of significant and insightful works have been done [4-8]. Our work contributes to this line of work by studying the meta-learning in the overparameterized regime.
> >
> > Moreover, a direct insight of our paper is the effect of task diversity discussed in Section 4.2. Our results establish a phase transition phenomenon, i.e., slower task diversity rate $r<2p-1$ guarantees vanishing excess risk, whereas faster task diversity rate $r \geq 2p - 1$ necessarily results in non-vanishing excess risk. Such a phase transition explicitly characterizes the effects of the task diversity in the overparameterized setting. Previous results about task hardness, e.g., [3], do not apply to the overparameterized setting and they only consider the isotropic task distribution. Also, the phase-transition type of results are interesting and significant in the literature on theory of overparameterization [4][5][8], and our result is the first (to our best knowledge) to capture that task diversity can cause phase transition of generalization.
> >
> > **Q4**: Can the results be generalized to other machine learning methods beyond the linear regression, such as neural networks?
> >
> > **A4**: Great question! Our work can directly extend to the random feature (RF) model, incorporating the analysis for nonlinear model in [6]: the data $\textbf{x}$ is generated by $\sigma(\textbf{Wz})$, where $\textbf{W}$ is the random feature matrix and $\sigma(\cdot)$  is the nonlinear (activation) function.  Hence, the RF model can be regarded as training a two-layer neural network where the weights in the first layer are chosen randomly and then fixed and only the output layer is optimized. Beyond the fixed feature approach, understanding the overparameterization in neural networks is much more challenging. Recently, [8] made important progress towards this aspect. They studied the benign overfitting phenomenon in training a two-layer convolutional neural network, and provided new analysis to tackle the neural network learning process. We are currently working on generalizing our analysis to neural networks by further advancing the techniques in [8].
> >
> > Finally, we want to thank the reviewer again for the helpful comments for our work. If our response resolves your concerns to a satisfactory level, we want to kindly ask the reviewer to consider raising the rating of our work. Certainly, we are more than happy to address any further questions you may have during the discussion period.
> >
> > References.
> >
> > [1] Chen et al., A Closer Look at the Training Strategy for Modern Meta-Learning, NeurIPS 2020
> >
> > [2] Fallah et al., Generalization of Model-Agnostic Meta-Learning Algorithms: Recurring and Unseen Tasks, NeurIPS 2021
> >
> > [3] Collins et al., Why Does MAML Outperform ERM? An Optimization Perspective, ArXiv 2020
> >
> > [4]  Bartlett et al., Benign overfitting in linear regression, PNAS 2020
> >
> > [5] Tsigler et al., Benign overfitting in ridge regression, Arxiv 2020
> >
> > [6] Hastie et al., Surprises in high-dimensional ridgeless least squares interpolation, Arxiv 2019
> >
> > [7] Belkin et al, Two Models of Double Descent for Weak Features, SIAM Journal on Mathematics of Data Science 2020
> >
> > [8] Cao et al., Benign Overtting in Two-layer Convolutional Neural Networks, Arxiv 2022

---

> ### Author Response · Authors · 2022-08-05
> **Could you please check our response?**
>
> Dear Reviewer 9gwj,
>
>  Since the author-reviewer discussion period has started for a few days, we will appreciate it if you could check our response to your review comments soon. This way, if you have further questions and comments, we can still reply before the author-reviewer discussion period ends. If our response resolves your concerns, we kindly ask you to consider raising the rating of our work. Thank you very much for your time and efforts!

---

> ### Author Response · Authors · 2022-08-08
> **Your prompt response is highly appreciated**
>
> Dear Reviewer 9gwj:
>
> As the author-reviewer discussion period ends soon, we will appreciate very much if your could check our response promptly. In particular, our response has explained in detail about your concerns on the novelty of the paper beyond a few existing studies (Q1) and the new insights of our results (Q3). If our response resolves your concerns, we kindly ask you to consider raising the rating of our work. We are also more than happy to answer your further questions. Thank you very much for your time and efforts!

---

> ### Comment · Area_Chair_6ioH · 2022-08-08
> **Please respond to the authors**
>
> Dear 9gwj,
>
> has the response of the authors alleviated your worries? Since the discussion period is quickly drawing to a close, please take the time to at least acknowledge the reply of the authors, and ask any final clarifications in response to their reply if you have them.
>
> Best,
> AC

---

> ### Comment · Reviewer_9gwj · 2022-08-09
> **Post-Rebuttal**
>
> The review thanks the authors for responses and revision.
>
> I understand that the main difference between this paper and the previous paper is the overparameteried setting. Due to the fact that the papers on theoretical analysis of MAML is massive, I still think the contribution of this paper is marginal.
>
> About paper revision. Thanks for letting me know. While according to NeurIPS guideline, your original submission will serve as the basis for the reviewers' (and ACs') acceptance recommendations.”
>
> My evaluation is based on the original submission and responses. I could not have sufficient time to recheck the revised version (sorry for that). Therefore, I keep my original rating.

---

> > ### Author Response · Authors · 2022-08-09
> > **Response to reviewer's post-rebuttal**
> >
> > While we thank the reviewer for providing the post-rebuttal feedback, we found that the reviewer's comment lacks details to be convincing. Below is our response.
> >
> > Q: I understand that the main difference between this paper and the previous paper is the overparameteried setting. Due to the fact that the papers on theoretical analysis of MAML is massive, I still think the contribution of this paper is marginal.
> >
> > A: Unfortunately, we disagree with the reviewer's comments due to the following two points.
> >
> > 1. We are shocked that a professional reviewer follows the logic that "the contribution of the paper is marginal simply because the studies on the topic is massive". The reviewer should at least give us a single paper that makes our result marginal. As we clearly elaborated in our previous response, all the references pointed out previously by the reviewer (which capture only underparameterized regime) are significantly different from our study here (which captures the overparameterized regime).
> >
> > 2. In fact, although theoretical studies on MAML are many, those on overparameterized MAML are extremely sparse. In fact, no paper characterized the overparameterized MAML under SGD dynamics, and no paper studied such a setting under general data and task distributions.
> >
> > We finally acknowledge that it is sufficient that the reviewer's evaluation is based on the original submission and responses. Our revision has no more new materials than our responses.

---

### Author Response · Authors · 2022-08-01
**Revision Uploaded**

We have uploaded a revised paper and supplementary materials, incorporating the various suggestions by the reviewers. Below is a summary of the changes we have made, which are highlighted by the blue-colored texts in the revisions.

1. We clarified the role of $r$ as the task diversity. See the paragraph below Proposition 2 in Section 4.2.

2. In Appendix G.1 (in the supplementary materials) we added the technical discussions about the related studies on the generalization of MAML both in the underparameterized and overparameterized regimes.

3. In Appendix H (in the supplementary materials), we added discussions about generalization of our results along a few interesting directions in the future.

We would like to thank all the reviewers again for their valuable comments and suggestions. We are more than happy to answer any further questions that the reviewers may have during the discussion period.

---

### Author Response · Authors · 2022-08-09
**Messages to All Reviewers and ACs after Rebuttal**

We first truly thank all reviewers for their insightful and constructive suggestions. We specially thank Reviewer **T2R8**, whose questions suggested interesting possible extensions of our analysis to other settings.  We note that it is sufficient that the reviewer’s evaluation is based on the original submission and our responses. Our revision has no more new materials than our responses.

Reviewer **9gwj**'s main concern lies in the difference of our study beyond the existing work. As we clearly elaborated in our response to the reviewer, all the references pointed out by the reviewer are on **underparameterized regime**, which are significantly different from our study here, which is on **overparaemterized regime**.

Unfortunately, we  regret about the further response by Reviewer **9gwj**, which is as simple as "Due to the fact that the papers on theoretical analysis of MAML is massive, I still think the contribution of this paper is marginal". We find the argument is unreasonable. Focusing on the **theoretical generalization studies of overparameterized MAML** (which is highly relevant to our work), there are no more than 5 papers to our best knowledge, which are all cited in our related work. Within such a context, our work is the **first effort** towards characterizing the overparameterized MAML under **SGD dynamics**, and the **first effort** towards understanding such a setting under **general** data and task distributions.

We thank all reviewers and ACs again for your time and efforts. In case any questions are raised during the reviewer-AC discussion period, we will be more than happy to further answer them if this is allowed by the conference review guidelines.

---

### Meta-Review · Area_Chair_6ioH · 2022-08-27

**Recommendation:** Accept
**Confidence:** Certain

**Metareview:**

This paper explores the generalization of SGD, as used in a MAML-style algorithm for meta learning in an overparametrized setting. The generative model considered is  "mixed linear regression", in which tasks follows a linear + Gaussian noise data model (a different direction per task, with minimal assumptions on the distributions for the directions --- precisely, the mean and covariance of the distribution). Overparametrization means d >> T, where d is the dimension of the vector of parameters, T is the number of tasks. The main conceptual takeaway is that there is a "phase transition" effect depending on what the step size is, and whether the risk decays as $T \to \infty$ --- depending (in a fairly complicated way) on various quantities, perhaps most interestingly a notion of "task diversity" (captured through the covariance of the direction for the tasks). The proof techniques largely follow [Zou et al '21] which considers overparametrization for SGD in just the case of linear regression --- which involves a bias/variance decomposition and a series of concentration bounds to bound each respectively. Since data coming from different tasks has a different SVD, this makes the proofs non-trivial to extend.



**Award:**

No

---

### Decision · Program_Chairs · 2022-09-14

Accept